EMBO
*reports*

# Lifelong absence of microglia alters hippocampal glutamatergic networks but not synapse and spine density

Michael Surala [1], Luna Soso-Zdravkovic[1], David Munro [2], Ali Rifat [1,3], Koliane Ouk [4], Imre Vida[5], Josef Priller [2,4,6,7,8 ✉] & Christian Madry [1,8 ✉]

## Abstract

**Microglia sculpt developing neural circuits by eliminating excess synapses in a process called synaptic pruning, by removing apoptotic neurons, and by promoting neuronal survival. To elucidate the role of microglia during embryonic and postnatal brain development, we used a mouse model deficient in microglia throughout life by deletion of the fms-intronic regulatory element (FIRE) in the *Csf1r* locus. Surprisingly, young adult *Csf1r*$^{\Delta FIRE/\Delta FIRE}$ mice display no changes in excitatory and inhibitory synapse number and spine density of CA1 hippocampal neurons compared with *Csf1r*$^{+/+}$ littermates. However, CA1 neurons are less excitable, receive less CA3 excitatory input and show altered synaptic properties, but this does not affect novel object recognition. Cytokine profiling indicates an anti-inflammatory state along with increases in ApoE levels and reactive astrocytes containing synaptic markers in *Csf1r*$^{\Delta FIRE/\Delta FIRE}$ mice. Notably, these changes in *Csf1r*$^{\Delta FIRE/\Delta FIRE}$ mice closely resemble the effects of acute microglial depletion in adult mice after normal development. Our findings suggest that microglia are not mandatory for synaptic pruning, and that in their absence pruning can be achieved by other mechanisms.**

**Keywords** Brain Development; Electrophysiology; Hippocampus; Microglia; Synapses
**Subject Category** Neuroscience

## Introduction

Microglia are the primary innate immune cells in the central nervous system (CNS). Beyond fulfilling classical immunological roles to protect the brain from injury or invading pathogens, microglia are increasingly recognized as key players involved in shaping neural function both in the developing and adult CNS (Thion and Garel, 2017; Pósfai et al, 2019). At early embryonic stages, when other glial populations such as astrocytes have not yet been generated, yolk sac-derived erythromyeloid progenitors engraft in the brain, differentiate into microglia, proliferate and rapidly colonize all brain areas until reaching maturity and steady-state cell densities early postnatally (Prinz and Priller, 2014). Thereafter, they reside as long-lived tissue macrophages in the brain parenchyma and, under physiological conditions, maintain their population through self-renewal (Askew and Gomez-Nicola, 2018).

Since microglia arise around the same time as neurons in the developing CNS, they are in an ideal position to interact with them from early on. Indeed, microglia have been shown to promote embryonic and adult neurogenesis (Vukovic et al, 2012; Ribeiro Xavier et al, 2015), as well as neuronal survival (Morgan et al, 2004; Ueno et al, 2013). They regulate axon guidance and the positioning of neocortical interneurons (Squarzoni et al, 2014), contribute to the migration and differentiation of neural precursor cells (Aarum et al, 2003; Antony et al, 2011), and eliminate newborn neurons and astrocytes by phagocytosis (Sierra et al, 2010; Cunningham et al, 2013; VanRyzin et al, 2019; Diaz-Aparicio et al, 2020). In late embryonic and early postnatal development, newly generated neurons undergo extensive synaptogenesis, followed by a period of net synapse elimination in a process called "synaptic pruning", which is critical for the maturation and refinement of neural circuits. Microglia participate in this process by eliminating unwanted or excess synapses of both excitatory and inhibitory circuits involving microglial-specific receptor pathways such as the fractalkine receptor (CX3CR1), the triggering receptor expressed on myeloid cells 2 (TREM2), or complement receptor CR3 (Paolicelli et al, 2011; Schafer et al, 2012; Zhan et al, 2014; Filipello et al, 2018; Gunner et al, 2019; Favuzzi et al, 2021). Apart from the elimination of synapses, microglia may also induce the formation of synapses and presynaptic rearrangement (Trapp et al, 2007; Miyamoto et al,

[1]Charité—Universitätsmedizin Berlin, Corporate member of Freie Universität Berlin and Humboldt Universität zu Berlin, Institute of Neurophysiology, Charitéplatz 1, 10117 Berlin, Germany. [2]University of Edinburgh and UK Dementia Research Institute, Edinburgh EH16 4TJ, UK. [3]Berlin Institute of Health at Charité—Universitätsmedizin Berlin, Charitéplatz 1, 10117 Berlin, Germany. [4]Charité—Universitätsmedizin Berlin, Corporate member of Freie Universität Berlin and Humboldt Universität zu Berlin, Neuropsychiatry and Laboratory of Molecular Psychiatry, Charitéplatz 1, 10117 Berlin, Germany. [5]Charité—Universitätsmedizin Berlin, Corporate Member of Freie Universität Berlin and Humboldt Universität zu Berlin, Institute for Integrative Neuroanatomy, Charitéplatz 1, 10117 Berlin, Germany. [6]DZNE Berlin, 10117 Berlin, Germany. [7]Department of Psychiatry and Psychotherapy; School of Medicine and Health, Technical University of Munich and German Center for Mental Health (DZPG), 81675 Munich, Germany. [8]These authors contributed equally: Josef Priller, Christian Madry. ✉E-mail: Josef.Priller@charite.de; christian.madry@charite.de

2016; Weinhard et al, 2018). Importantly, on the functional level, microglia regulate synaptic transmission and plasticity, thereby affecting network function and consequently memory formation and behavior (Paolicelli et al, 2011; Parkhurst et al, 2013; Zhan et al, 2014; Basilico et al, 2019; Merlini et al, 2021; Wang et al, 2020; Basilico et al, 2022). Microglial interactions with neurons occur at specific cell-cell contacts (Cserép et al, 2020), and depend on neuronal activity (Tremblay M- et al, 2010; Li et al, 2012; Liu et al, 2019; Stowell et al, 2019; Cserép et al, 2020; Badimon et al, 2020). Remarkably, by reducing the discharge rate of excessively active neurons, microglia reinstate homeostasis by protecting neurons from adopting potentially harmful hyperactive states (Li et al, 2012; Cserép et al, 2020; Badimon et al, 2020).

Collectively, the existing literature ascribes to microglia a crucial role in CNS development and function, leading to the hypothesis that without the involvement of microglia these processes may be significantly impaired with deficits in neuronal maturation and function on the cellular, synaptic and network level. This view is further supported by recent studies in adult mice in which microglia have been transiently depleted, resulting in changes in both glutamatergic and GABAergic transmission (Liu et al, 2021; Ma et al, 2020; Basilico et al, 2022; Du et al, 2022).

Despite increasing knowledge of microglia as active sculptors of neural circuits, their global role in shaping brain function, including the contribution of other microglial-independent mechanisms, still remains largely unclear. This is mainly due to the fact that existing studies employed models with functionally impaired or transiently depleted microglia at postnatal stages, so that microglia were either always present, albeit functionally compromised, to maintain interactions with other cells, or they were only absent from later postnatal stages onward.

The proliferation, differentiation, and survival of cells of the mononuclear phagocyte system depends upon signals from the macrophage colony-stimulating factor 1 receptor (CSF1R) which is expressed exclusively in cells of this lineage. Dominant and recessive mutations in the *CSF1R* gene in zebrafish, mice, rats and humans are associated with the loss of microglia and most tissue macrophage populations, with complex impacts on postnatal development (Hume et al, 2020). In the rat, complete loss of microglia and brain-associated macrophages as a consequence of homozygous mutation of the *Csf1r* locus has remarkably little impact on pre- and postnatal brain development (Keshvari et al, 2021; Patkar et al, 2021), and in C57BL/6J mice, the perinatal lethality of *Csf1r* mutation can be overcome by neonatal transfer of wild-type bone marrow cells (Bennett et al, 2018). These findings suggest that the developmental functions of microglia may be partly redundant.

In this study, we sought to elucidate the influence of microglia on shaping neuronal properties during brain development by using a mouse model which is entirely deficient in microglia throughout all stages of life, including embryonal development (Rojo et al, 2019). This line was generated by germ-line deletion of the *fms*-intronic regulatory element (FIRE) in the *Csf1r* locus, which is required for expression of the CSF1R in bone marrow progenitors and blood monocytes. At least on a mixed genetic background, $Csf1r^{\Delta FIRE/\Delta FIRE}$ mice are healthy, fertile and develop normally without apparent behavioral abnormalities but lack microglia and subsets of brain-associated macrophages, as well as

macrophages in heart, skin, kidney and peritoneal cavity (Rojo et al, 2019; McNamara et al, 2023). The absence of microglia in this line had little effect on gene expression in the hippocampus, aside from the loss of the microglial gene signature (Rojo et al, 2019).

We characterized the effect of the developmental absence of microglia in $Csf1r^{\Delta FIRE/\Delta FIRE}$ mice on the glutamatergic network in the hippocampus in young adult mice, at a stage when synaptic pruning has been largely completed (Jawaid et al, 2018). Specifically, we examined the morphological and functional properties of hippocampal pyramidal neurons on the cellular, synaptic and microcircuit level by whole-cell patch-clamp electrophysiology. The lack of microglia did not affect excitatory synapse number nor spine density or morphology of CA1 neurons but resulted in a weakened glutamatergic transmission in $Csf1r^{\Delta FIRE/\Delta FIRE}$ mice compared to littermate controls. We further investigated the consequences of microglial deficiency on non-spatial memory in the novel object recognition test which involves the hippocampus, and on the immunological milieu in brains of $Csf1r^{\Delta FIRE/\Delta FIRE}$ mice by biochemical and ELISA-based analyses. Our data show that object memory was not affected and cytokine levels were only mildly altered towards an anti-inflammatory state, while ApoE levels were increased and astrocytes were reactive in $Csf1r^{\Delta FIRE/\Delta FIRE}$ mice.

## Results

### Absence of microglia has no effect on CA1 excitatory synapse number and spine density, and morphology

Morphological properties of hippocampal CA1 pyramidal neurons were determined in acute brain slices from young adult 6–10-week-old $Csf1r^{\Delta FIRE/\Delta FIRE}$ mice and controls at post-pruning stages (Jawaid et al, 2018). In line with previous findings (Rojo et al, 2019), we observed a complete absence of parenchymal Iba1 immunoreactivity in hippocampal and neocortical regions of $Csf1r^{\Delta FIRE/\Delta FIRE}$ mice compared to wild-type (WT, $Csf1r^{+/+}$) littermates (Fig. 1A). We chose the hippocampus as a particularly well-characterized brain region for microglia–neuron interactions (Paolicelli et al, 2011; Parkhurst et al, 2013; Zhan et al, 2014; Basilico et al, 2019), which would allow us to compare our findings with recent studies investigating the neuronal consequences of transient depletion of microglia in adult WT mice (Parkhurst et al, 2013; Basilico et al, 2022; Du et al, 2022).

We first analyzed the density and morphology of CA1 apical dendritic spines, assuming we would find changes in the number of glutamatergic contacts (Fig. 1B). Surprisingly, three-dimensional biocytin-based anatomical reconstructions did not reveal any changes in spine density or spine length in $Csf1r^{\Delta FIRE/\Delta FIRE}$ mice compared to WT littermates (Fig. 1C–E). In addition, morpho-metric characterization of CA1 pyramidal cells by Sholl analysis showed no changes in their dimension and dendritic arborization in $Csf1r^{\Delta FIRE/\Delta FIRE}$ mice compared to WT littermates (Fig. 1F–I). To extend our analysis to the whole synapse, we immunohistochemi-cally labeled pre- and postsynaptic puncta with VGluT1 and Homer1, respectively. Consistent with the spine data, the number and size of excitatory synapses, as defined by the colocalization of

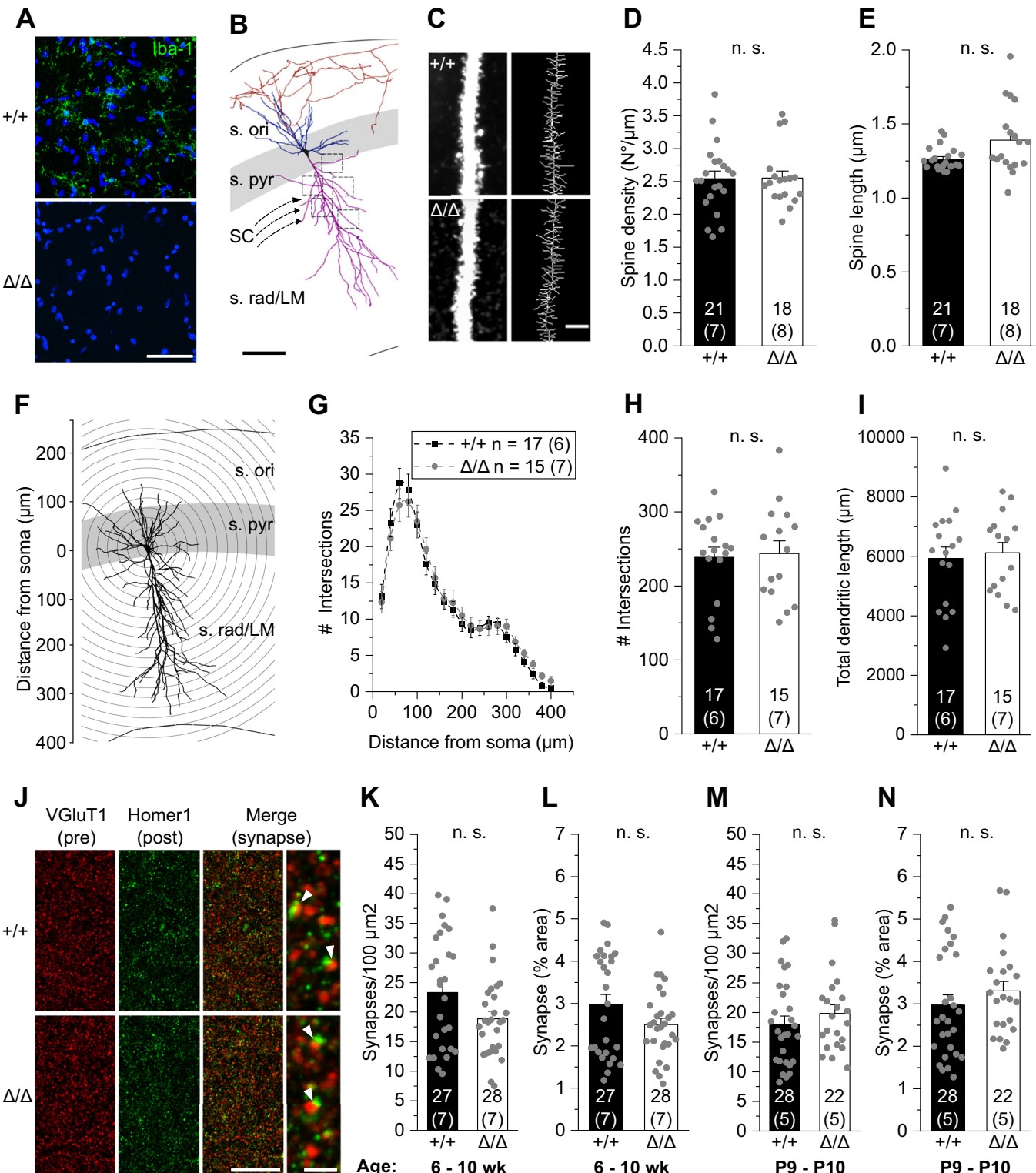

both markers, were unchanged in the CA1 region in *Csf1r*$^{\Delta FIRE/\Delta FIRE}$ mice during development at postnatal days (P)9-P10 and in young adulthood compared with WT mice (Fig. 1J–N). In summary, these findings indicate that excitatory synapses onto CA1 pyramidal cells are not changed in number by the absence of microglia.

## Absence of microglia results in reduced excitability of CA1 pyramidal cells

We next investigated whether the absence of microglia has any consequences on the physiological function of CA1 neurons.

◄ **Figure 1.  Absence of microglia has no effect on CA1 excitatory synapse number, spine density, and morphology.**

(A) Specimen confocal images illustrating the lack of microglia by Iba1 immunoreactivity (green) in acute hippocampal slices of *Csf1r*$^{\Delta FIRE/\Delta FIRE}$ (Δ/Δ) compared to WT littermates (+/+). DAPI labeling of cellular nuclei in blue. Scale bar, 50 μm. (B) 3D reconstruction of a biocytin-filled CA1 pyramidal cell (black, cell body; blue, basal dendrite; magenta, apical dendrite; red, axon), depicting the excitatory input via the Schaffer collaterals (SC) and hippocampal layers. (s. ori—*stratum oriens*, s. pyr—*stratum pyramidale*, s. rad/LM—*stratum radiatum/lacunosum-moleculare*). Squared boxes indicate five different apical regions of interest for spine analysis. Individual values were averaged to obtain a grand average per cell. Scale bar, 50 μm. (C) Representative segments of apical dendrites at high power (left) from which skeletonized branches were generated (right). Scale bar, 3 μm. (D, E) Analysis of mean apical spine density (D) and spine length (E). (F) Schematic representation of Sholl analysis of a biocytin-filled CA1 pyramidal cell. Number of intersections between dendrites and concentric spheres centered around the soma was determined at increasing distances with 20 μm increments. (G–I) Sholl analysis-derived values of the number of intersections with Sholl radii at increasing distance from the soma (G) and resulting total number of intersections (H) and total dendritic length (I) of CA1 pyramidal cells. (J) Confocal images showing VGluT1-labeled presynaptic puncta (red) and Homer1-labeled postsynaptic puncta (green) in the CA1 *stratum radiatum*. The merged image and expanded view on the right show excitatory synapses as colocalized puncta (arrowheads). Scale bars, 20 μm and 2 μm (for expanded view). (K–N) Quantification of colocalized puncta (excitatory synapses) per 100 μm$^2$ and their area covered of mice aged 6–10 weeks (K, L) and 9–10 days (M, N). Data information: Data indicate mean ± SEM. Numbers on bars show tested cells (C–I) or number of slices (K–N) and (number of animals). *P* values are from unpaired Student's *t* (H, I, K, M) or Mann–Whitney tests (D, E, L, N). Source data are available online for this figure.

Whole-cell patch-clamp recordings revealed no changes in input resistance, cell capacitance or resting potential (Fig. 2A–D). However, CA1 pyramidal cells produced a slightly lower rate of action potentials on injection of depolarizing current in *Csf1r*$^{\Delta FIRE/\Delta FIRE}$ mice compared to WT littermates (Fig. 2E,F), along with an increase in minimal current to reach action potential threshold (rheobase) (Fig. 2G). No changes were seen in the threshold voltage eliciting action potentials (Fig. 2H). This indicates lower neuronal excitability of CA1 pyramidal cells in *Csf1r*$^{\Delta FIRE/\Delta FIRE}$ mice, which plausibly reflects changes in axonal electrophysiology, contrasting the lack of changes in passive membrane parameters, mainly defined by properties of the somatodendritic domain (Fig. 2B–D).

In contrast, in heterozygous *Csf1r*$^{+/\Delta FIRE}$ mice, in which microglia were present at normal densities but were morphologically more ramified compared to WT littermates (Fig. EV1A–C), no changes were seen for excitability, input resistance, cell capacitance, and resting potential of CA1 pyramidal cells (Fig. EV1D–J).

Taken together, these results demonstrate that the absence of microglia in *Csf1r*$^{\Delta FIRE/\Delta FIRE}$ mice leads to a reduced ability of CA1 pyramidal cells to generate action potentials, thereby limiting glutamatergic transmission downstream.

## Reduced CA3–CA1 glutamatergic transmission in Csf1r$^{\Delta FIRE/\Delta FIRE}$ mice

Based on the observed changes in neuronal output, we investigated the impact of microglial deficiency on hippocampal transmission, focusing on excitatory inputs onto CA1 pyramidal cells. To this end, we electrically stimulated CA3 Schaffer collaterals at increasing stimulus strength and recorded compound excitatory postsynaptic currents (EPSCs) reflecting postsynaptic AMPA receptor (AMPAR) activation in CA1 pyramidal cells (Fig. 3A,B). As revealed by the input–output relationships, *Csf1r*$^{\Delta FIRE/\Delta FIRE}$ mice showed greatly reduced EPSC amplitudes across the entire stimulation range compared to WT littermates (Fig. 3C), indicating functional deficits in fast excitatory hippocampal transmission. Using the same recording configuration, we analyzed the amplitude ratio of two consecutive EPSCs in CA1 pyramidal cells by paired-pulse stimulation of CA3 Schaffer collaterals. However, this revealed no changes in paired-pulse ratio (PPR) in *Csf1r*$^{\Delta FIRE/\Delta FIRE}$ (Fig. 3D,E) and *Csf1r*$^{+/\Delta FIRE}$ (Fig. EV2A,B) mice compared to WT littermates.

Taken together, these results show that, although presynaptic glutamate release is preserved, the absence of microglia throughout development results in a lower excitatory input into CA1 pyramidal cells from the hippocampal CA3 region.

## Absence of microglia causes deficits in excitatory synaptic transmission

To test for changes in synaptic properties, we next analyzed spontaneous CA1 EPSCs (sEPSCs) including both action potential-dependent and -independent synaptic events, as well as action potential-independent miniature EPSCs (mEPSCs) in the presence of 0.3 μM tetrodotoxin (TTX) to block voltage-gated sodium channels (Fig. 4A). Inter-event intervals of mEPSCs and sEPSCs were not different in *Csf1r*$^{\Delta FIRE/\Delta FIRE}$ mice compared to WT littermates (Fig. 4B,C), suggesting an unchanged number of functional glutamatergic synapses in agreement with our morphological findings. No changes were further observed in mEPSC decay time reflecting postsynaptic AMPAR-mediated currents (Fig. 4D,E). Notably, analysis of current amplitudes showed impairment of multivesicular release, also referred to as synaptic multiplicity (i.e., the number of individual synaptic contacts and/or release sites between two connected neurons) in *Csf1r*$^{\Delta FIRE/\Delta FIRE}$ mice as revealed by comparable peak amplitudes of sEPSCs and mEPSCs (Fig. 4F). In contrast, these changes were not seen in *Csf1r*$^{+/\Delta FIRE}$ heterozygous mice (Fig. EV2C,D). Given the unchanged number of functional synaptic contacts (cf. Fig. 1J–N), this suggests a shift towards single-synaptic boutons with one presynapse contacting a single postsynaptic spine. In contrast, WT littermates exhibited larger amplitudes of sEPSCs compared to mEPSCs (Fig. 4F), suggesting multiple presynaptic glutamate release sites per synapse as a characteristic feature of mature excitatory circuitry (Hsia et al, 1998).

## Absence of microglia does not change inhibitory synaptic transmission

No changes were observed in GABAergic synaptic transmission since inter-event intervals and amplitudes of either spontaneous inhibitory postsynaptic currents (sIPSCs) or miniature inhibitory postsynaptic currents (mIPSCs) were unaltered in CA1 neurons of *Csf1r*$^{\Delta FIRE/\Delta FIRE}$ mice (Fig. EV3A–F). The number and size of inhibitory synapses, as defined by the colocalization of presynaptic

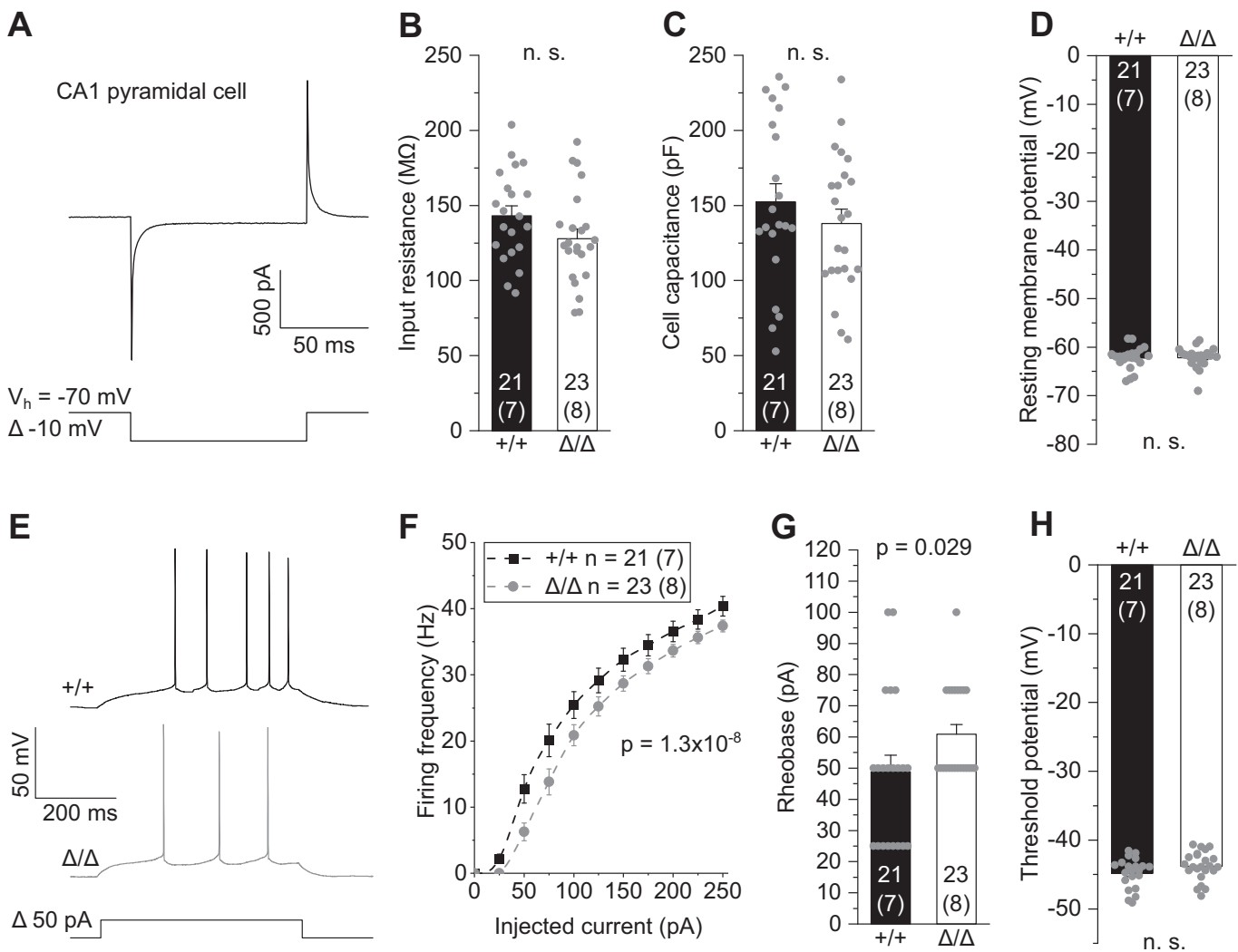

**Figure 2. Absence of microglia results in reduced excitability of CA1 pyramidal cells.**

(A) Patch-clamped membrane current of a CA1 pyramidal cell to a brief 10 mV hyperpolarization from which values for input resistance and cell capacitance were determined (see "Methods"). (B–D) Quantification of input resistance (B), cell capacitance (C) and resting membrane potential (D) of CA1 pyramidal cells in Δ/Δ and +/+ mice. (E) Specimen traces showing action potential firing patterns of CA1 pyramidal cells in response to 500 ms depolarizing current injections. (F) Corresponding course of action potential firing frequencies on increasing depolarizations. (G, H) Values for rheobase, i.e., minimal current to reach action potential threshold, (G) and action potential threshold voltage (H). Data information: Data are represented as mean ± SEM. Numbers on bars indicate tested cells and (number of animals). $P$ values are from unpaired Student's $t$ (B–D, H), or Mann–Whitney tests (G) and two-way ANOVA (F). Source data are available online for this figure.

VGAT and postsynaptic Gephyrin immunoreactivities, were unchanged in the CA1 region in 6–10-week-old $Csf1r^{\Delta FIRE/\Delta FIRE}$ mice compared with WT littermates (Fig. EV3G–I).

Taken together, our findings suggest that the number of functional synaptic contacts is unaffected in hippocampal pyramidal neurons in $Csf1r^{\Delta FIRE/\Delta FIRE}$ mice. However, the absence of microglia affects the maturation of glutamatergic synapses by impairment of multivesicular release.

## Absence of microglia leads to reduced synaptic NMDA receptor components in CA1 pyramidal cells

So far, the above experiments captured changes in fast excitatory transmission via activation of AMPARs (AMPAR), while NMDA

receptor activation (NMDAR) was largely abolished due to its blockade by $Mg^{2+}$ at negative membrane voltages. To investigate in more detail the changes in postsynaptic glutamate receptor function of both AMPAR and NMDAR, we adapted our recording conditions by perfusing slices with $Mg^{2+}$-free extracellular solution in the presence of glycine and TTX, to promote NMDAR activation and shut down network activity. This resulted in dual-component mEPSCs carried by AMPAR plus NMDAR, and NMDAR-only mEPSCs (Fig. 5A). To analyze the fraction of AMPAR versus NMDAR currents, expressed as electrical charge, we averaged >100 single events to create average mEPSCs per cell and pharmacologically isolated the (i) AMPAR component from (ii) mixed AMPAR and NMDAR events by applying the specific NMDAR blocker D-AP5. Subtracting (ii) from (i) yields the calculated NMDAR

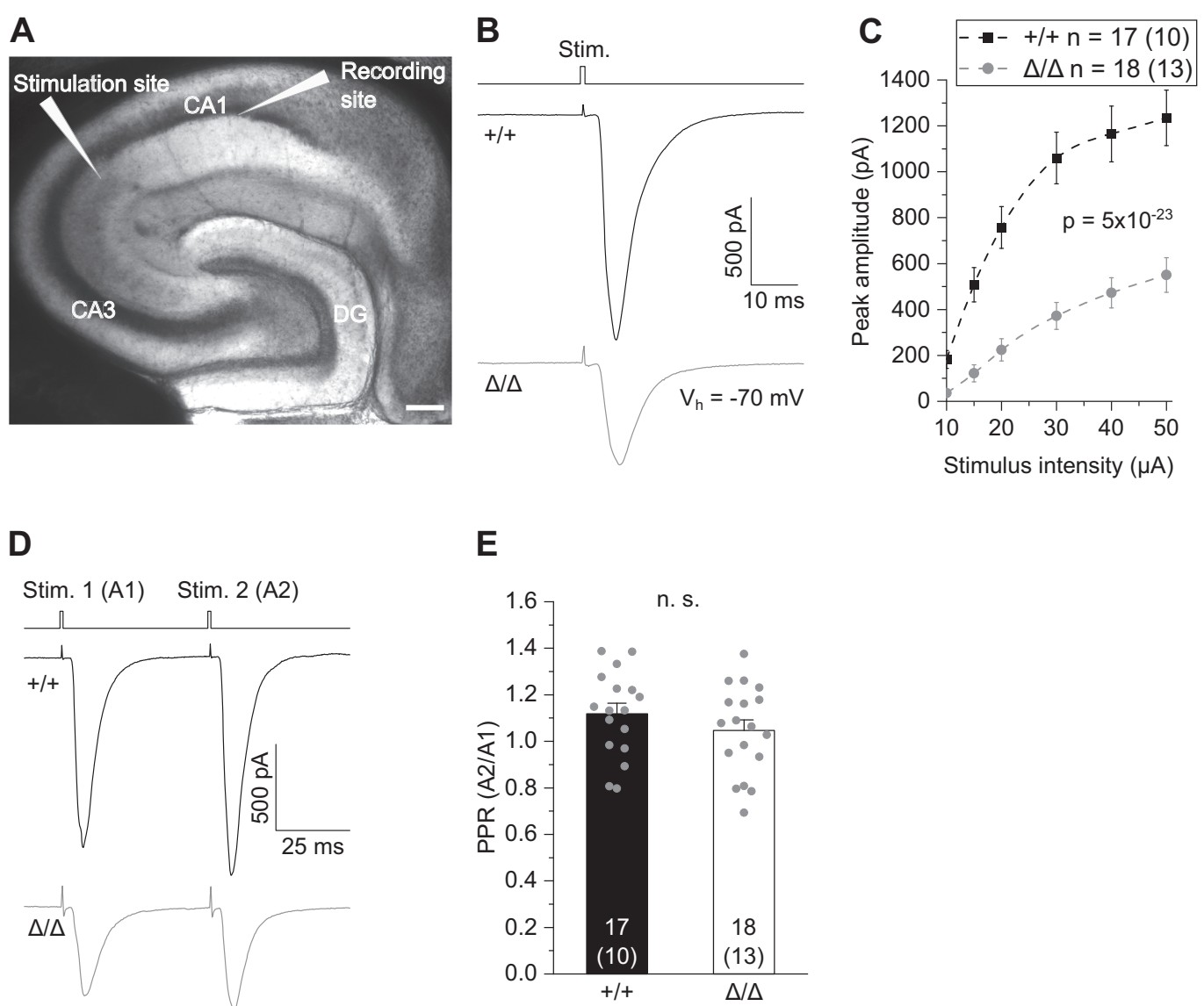

**Figure 3. Reduced CA3–CA1 glutamatergic transmission in *Csf1r*^ΔFIRE/ΔFIRE^ mice.**

(A) Differential interference contrast image showing the localization of CA3 stimulation and CA1 recording sites. Scale bar, 200 μm. (CA1, CA3: *cornu ammonis regions 1 and 3*; DG: *dentate gyrus*). (B) Specimen traces of excitatory postsynaptic currents (EPSC) in CA1 pyramidal cells in response to electrical stimulation for 0.2 ms. (C) Corresponding input–output relationship showing peak amplitudes of CA1 EPSCs with increasing stimulation strength. (D) Example traces (EPSC) in CA1 pyramidal cells after paired-pulse stimulation at 50 ms inter-stimulus intervals. (E) Comparison of paired-pulse ratios (PPR) as the quotient of the second vs first EPSC amplitude (A2/A1) of CA1 pyramidal cells. Data information: Data show mean ± SEM. Numbers on bars indicate tested cells and (the number of animals). *P* values are from two-way ANOVA (C) and unpaired Student's *t* test (E). Source data are available online for this figure.

component (Fig. 5B). This revealed a lower NMDAR-mediated charge in *Csf1r*^ΔFIRE/ΔFIRE^ mice compared to WT littermates (Fig. 5C), while, consistent with our findings above, no changes were seen for the AMPAR component (Fig. 5D, cf. Fig. 4E,F), resulting in a higher AMPAR/NMDAR charge ratio (Fig. 5E).

Analysis of average mEPSCs assumes a homogeneous expression of glutamate receptors at individual postsynaptic sites, but there may be differences in the proportion of synapses in *Csf1r*^ΔFIRE/ΔFIRE^ mice with increased or decreased AMPAR and NMDAR function that may have been lost by averaging. To address this issue, we compared the AMPAR-mediated peak current of individual

mEPSCs with its NMDAR current component (see "Methods" for details). Fitting of the data revealed a comparable degree of correlation of AMPAR to NMDAR currents in WT and *Csf1r*^ΔFIRE/ΔFIRE^ mice (Fig. 5F,G). This suggests a similar distribution of synapses with different fractions of AMPAR versus NMDAR expression. In addition, analysis of covariance revealed a downward shifted regression line in *Csf1r*^ΔFIRE/ΔFIRE^ mice (Fig. 5G), corroborating our finding of a reduced NMDAR component at the level of individual synapses.

Due to the observed differences in synaptic NMDAR function in *Csf1r*^ΔFIRE/ΔFIRE^ mice, we extended our analysis to investigate tonic

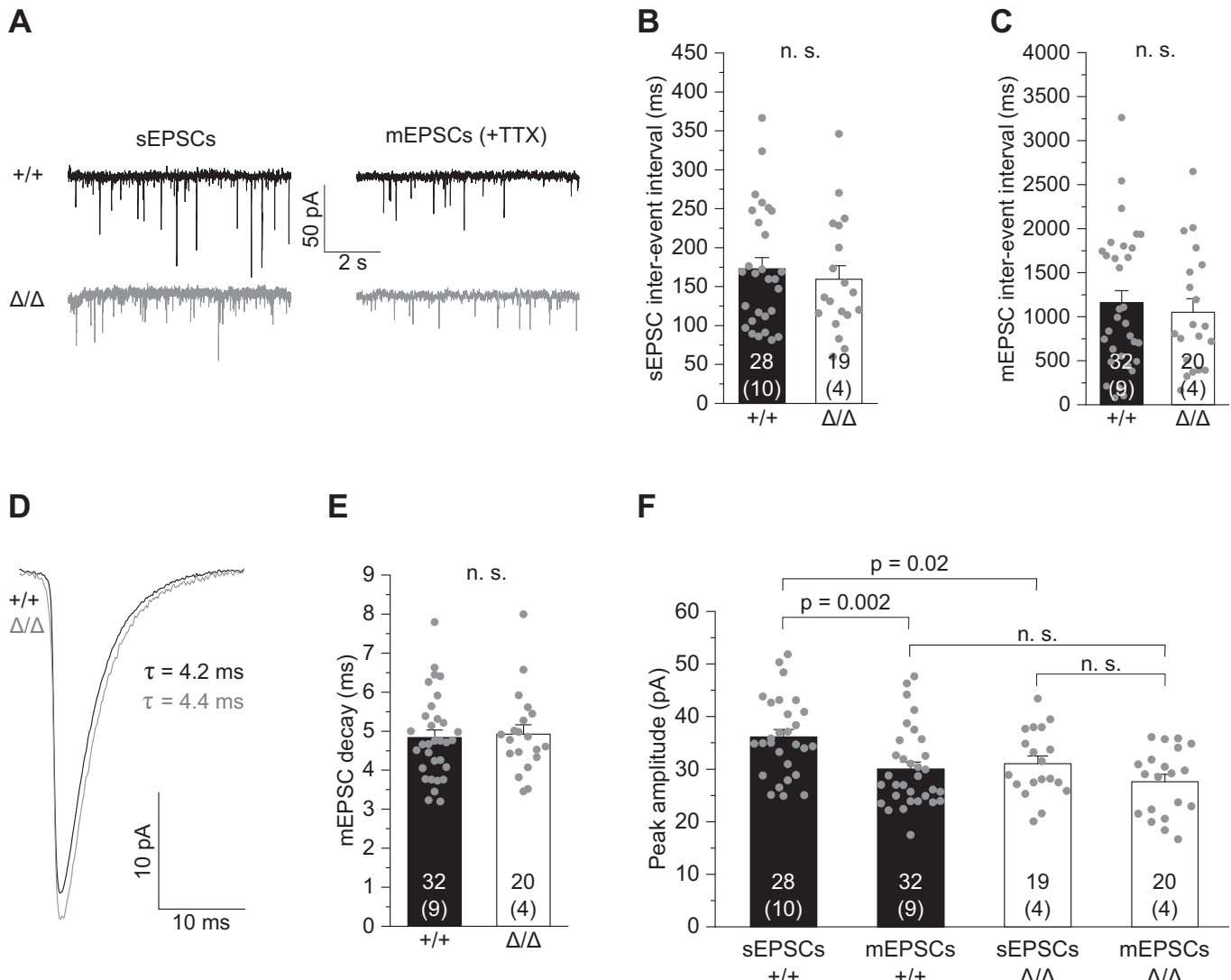

**Figure 4. Absence of microglia causes deficits in excitatory synaptic transmission.**

(A) Specimen traces showing AMPAR-mediated spontaneous EPSCs (sEPSCs) and miniature EPSCs (mEPSCs) in the presence of 300 nM TTX of CA1 pyramidal cells. (B, C) Comparison of inter-event intervals of sEPSCs (B) and mEPSCs (C). (D) Specimen traces showing average mEPSCs kinetics of CA1 pyramidal cells. (E) Analysis of decay times of CA1 mEPSCs (AMPA receptor-evoked currents). (F) Comparison of peak amplitudes of sEPSCs and mEPSCs. Note that mEPSC amplitudes are not different amongst genotypes, suggesting no change in functional synaptic contacts, whereas action potential-dependent sEPSC amplitudes are larger in +/+ but not in Δ/Δ mice, indicating impairment in synaptic multiplicity in the latter. Data information: Data indicate mean ± SEM. Numbers on bars show tested cells and (the number of animals). *P* values are from unpaired Student's *t* [**C**, **E**, **F** (for Δ/Δ and sEPSC comparison)] or Mann–Whitney tests [**B**, **F** (for +/+ and mEPSC comparison)]. Source data are available online for this figure.

activation of NMDAR in CA1 pyramidal cells activated by ambient glutamate in the extracellular space. Tonic D-AP5-sensitive inward currents were measured in $Mg^{2+}$-free, TTX-containing extracellular solution with glycine added as NMDAR co-agonist to unblock NMDAR and facilitate their activation (Fig. EV4A). Under these conditions, all existing NMDAR-mediated currents contribute to the tonic current, comprising receptors located at mixed AMPAR/ NMDAR synapses, silent synapses and extrasynaptically. Analysis of the D-AP5-sensitive current and RMS noise revealed an increase in both parameters in *Csf1r*$^{\Delta FIRE/\Delta FIRE}$ (but not *Csf1r*$^{+/\Delta FIRE}$) mice compared to WT littermates (Fig. EV4B–F), further indicating altered NMDAR function in mice deficient in microglia.

In summary, these data demonstrate the developmental impact of microglia in regulating postsynaptic function by a reduction of the NMDAR component.

## Absence of microglia changes cytokine and ApoE levels in the brain and results in reactive astrocytes

As the main immune cell type in the brain, microglia can affect neuronal function by releasing immune-modulatory signaling molecules, collectively referred to as cytokines. Notably, microglial cytokines play essential roles in synaptogenesis and synaptic transmission (Werneburg et al, 2017). To examine the consequences of permanent

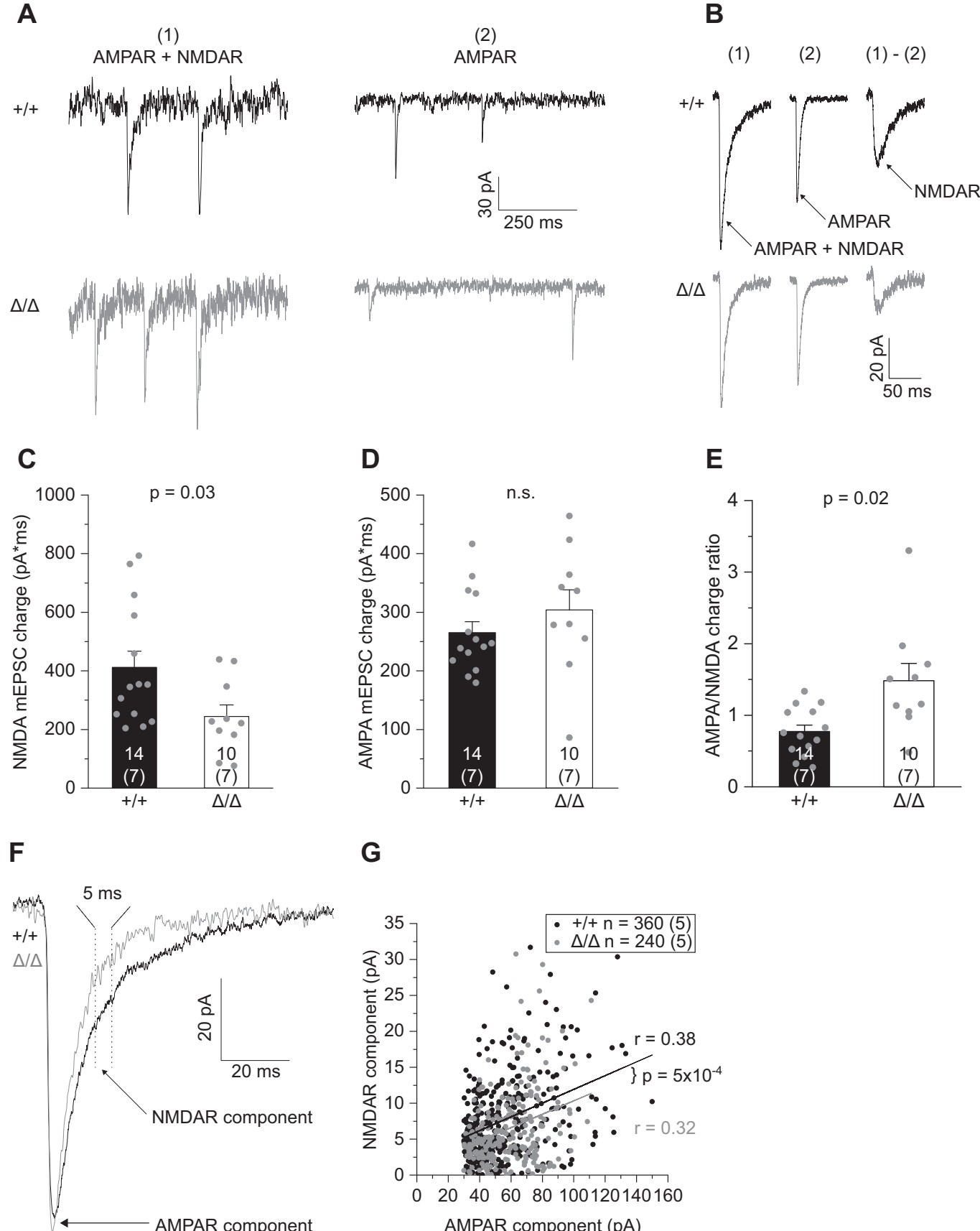

◄

**Figure 5. Absence of microglia results in reduced synaptic NMDA receptor component in CA1 pyramidal cells.**

(A) Specimen traces showing dual-component mEPSCs comprising AMPAR- and NMDAR-evoked currents measured in nominally Mg$^{2+}$-free extracellular solution in the presence of 300 nM TTX, 10 µM gabazine and 10 µM glycine (left), and pharmacologically isolated AMPAR-only mEPSCs after blockade of NMDARs with 50 µM D-AP5 (right). (B) Average dual-component (1) and AMPAR-only (2) mEPSCs from which the synaptic NMDAR component was calculated by subtracting (1) – (2). (C–E) Comparison of the NMDAR- (C) and AMPAR-mediated mEPSC charge (D) and the resulting AMPA/NMDA charge ratio (E). (F) Specimen traces showing individual dual-component mEPSCs containing both AMPAR and NMDAR components. Arrows indicate where respective currents were measured. (G). Correlation between the AMPA and NMDA measurements of individual mEPSCs comprising 40 events per cell. Data information: Data are represented as mean ± SEM. Numbers on bars indicate tested cells and (the number of animals). P values are from unpaired Student's t test (D, E), Mann–Whitney test (C) or ANCOVA (G, co-variant: AMPA peak amplitude). R values refer to Pearson correlation coefficients. Source data are available online for this figure.

absence of microglia on cytokine levels in the forebrain (excluding olfactory bulb and cerebellum), we determined the levels of anti- and pro-inflammatory cytokines in $Csf1r^{\Delta FIRE/\Delta FIRE}$ mice and WT littermates by high-sensitivity multiplex-ELISA in soluble and membrane-bound brain fractions (Fig. 6A). $Csf1r^{\Delta FIRE/\Delta FIRE}$ mice showed subtle changes in cytokine levels with increases in anti-inflammatory IL-10 and IL-4, while pro-inflammatory IFNγ and IL-2 were reduced (Fig. 6B and Table 1). Notably, TNFα and IL-1ß levels, two key pro-inflammatory microglial cytokines, were unchanged in the brains of $Csf1r^{\Delta FIRE/\Delta FIRE}$ mice compared to $Csf1r^{+/\Delta FIRE}$ and WT littermates (Fig. 6B and Table 1). In addition, $Csf1r^{+/\Delta FIRE}$ mice revealed subtle reductions in INFγ as well as CXCL1 (Table 1).

Apolipoprotein E (ApoE) is mainly produced by astrocytes in the murine brain (Boyles et al, 1985; Pitas et al, 1987; Zhang et al, 2014; Zhang et al, 2016). Interestingly, soluble and membrane-bound ApoE levels were strongly increased in the brains of $Csf1r^{\Delta FIRE/\Delta FIRE}$ mice compared to $Csf1r^{+/\Delta FIRE}$ and WT littermates (Fig. 6C and Table 1), suggesting that changes in ApoE are due to altered astrocyte function in the absence of microglia. Morphological analysis of astrocytes in the CA1 stratum radiatum of 6–10-week-old $Csf1r^{\Delta FIRE/\Delta FIRE}$ mice revealed an increase in both intensity and area covered by GFAP immunor-eactivity, while astrocyte numbers were unchanged compared to WT littermates (Fig. 7A–D). In line with this, we observed increased complexity of astrocyte morphology, as evidenced by Sholl analysis revealing increases in the number of intersections and total process length (Fig. 7E–H). Notably, the intensity and area covered by GFAP immunoreactivity was even more pronounced in $Csf1r^{\Delta FIRE/\Delta FIRE}$ mice at P22–P23 (Fig. 7I–K), a time when engulfment of synaptic material by hippocampal microglia would normally have reached its peak (Jawaid et al, 2018). This was accompanied by marked increases in the expression of the key phagocytic receptor mediating synapse elimination in astrocytes, Multiple EGF-like domains (MEGF)10 (Fig. 7I,L,M; Chung et al, 2013), and enhanced incorporation of the synaptic markers VGluT1 and Homer1 in hippocampal astrocytes of $Csf1r^{\Delta FIRE/\Delta FIRE}$ mice at P22–P23 compared to WT littermates (Fig. 7N–P).

Taken together, these data indicate a trend toward an anti-inflammatory state in the brains of $Csf1r^{\Delta FIRE/\Delta FIRE}$ mice along with the presence of reactive astrocytes in the hippocampus with increased uptake of synaptic material. These changes are most prominent during hippocampal development, suggesting that astrocytes may contribute to pruning in the absence of microglia.

### Absence of microglia does not alter object recognition memory

Finally, we asked whether the observed functional changes are reflected at the behavioral level by affecting memory. Novel object recognition

(NOR) is a widely accepted task for assessing nonspatial memory in rodents with an important contribution of the hippocampus and perirhinal cortex (Cohen and Stackman, 2015). Here, after habituation to a familiar object, mice are exposed to a novel object to interact with. We detected no differences in the performance in the novel object recognition test of 6–10-week-old male and female $Csf1r^{\Delta FIRE/\Delta FIRE}$ as well as $Csf1r^{+/\Delta FIRE}$ mice compared to WT littermates (Figs. 8A,B and EV5A,B), suggesting preserved nonspatial memory in the absence of microglia.

## Discussion

In this study, we investigated the role of microglia in embryonic and postnatal development with regard to their capacity to sculpt glutamatergic circuitry in the hippocampus and contribute to nonspatial memory. We examined synapse density and dendritic spine morphology along with the electrophysiological profile of hippocampal pyramidal neurons on the synaptic and cellular level, and cognitive function in the NOR task in young adult $Csf1r^{\Delta FIRE/\Delta FIRE}$ mice that are deficient in microglia throughout life. Our motivation was further fueled by the fact that these mice develop fairly normally (Rojo et al, 2019; McNamara et al, 2023), which appears to contradict findings attributing a crucial role to microglia in neural development and function.

Our study has disclosed the following main results: (1) Number and size of excitatory synapses in the CA1 stratum radiatum, as well as spine density of apical dendrites of CA1 pyramidal cells are unchanged in $Csf1r^{\Delta FIRE/\Delta FIRE}$ mice, suggesting that microglia are dispensable for synaptic pruning. (2) CA1 pyramidal cells are less excitable and receive less excitatory input from CA3 Schaffer collaterals in $Csf1r^{\Delta FIRE/\Delta FIRE}$ mice, resulting in a weakened glutamatergic transmission in the absence of microglia. At the synaptic level, this is accompanied by impairment of multivesicular release and a reduction in the postsynaptic NMDA receptor component, without changes in inhibitory GABAergic transmission. (3) Nonspatial memory, to which hippocampus and perirhinal cortex contribute, is not impaired in $Csf1r^{\Delta FIRE/\Delta FIRE}$ mice. (4) Absence of microglia results in a cerebral environment with mild increases in anti-inflammatory cytokines, along with strong increases in ApoE levels and reactive astrocytes with enhanced expression of GFAP and MEGF10 containing synaptic markers.

### Dispensability of microglia to eliminate the surplus of synapses of developing hippocampal neurons

A major outcome of our work is that the developmental and postnatal absence of microglia in young adult $Csf1r^{\Delta FIRE/\Delta FIRE}$ mice

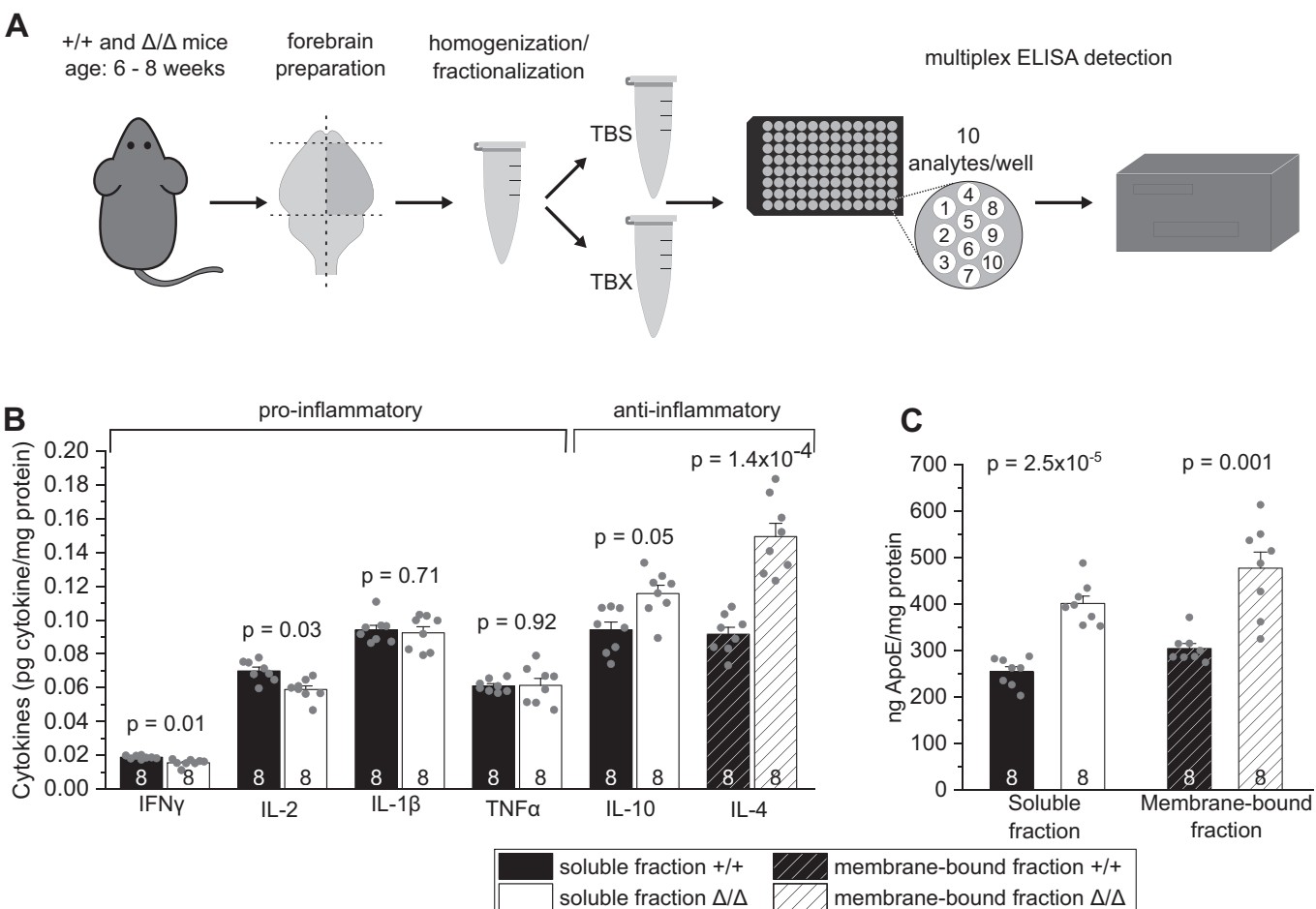

**Figure 6. Absence of microglia changes cytokine and ApoE levels in the brain.**

(**A**) Workflow illustrating the ELISA-based detection of brain cytokines and ApoE in soluble (TBS) and membrane-bound (TBX) fractions from cortical brain homogenates (see "Methods"). (**B, C**) Comparison of basal levels of anti- and pro-inflammatory cytokines (**B**) and apolipoprotein E (ApoE) (**C**) in soluble and membrane-bound fractions in $\Delta/\Delta$ mice and $+/+$ mice. Data information: Data are represented as mean ± SEM. Numbers on bars indicate the number of animals. *P* values are from unpaired Student's *t* tests corrected for multiple comparisons. Source data are available online for this figure.

does not affect synaptic markers and spine density in apical dendritic regions of CA1 pyramidal cells. This is a surprising finding given that microglia have been suggested to prune the majority of excess synapses during the first postnatal weeks in order to form a properly wired and adaptive brain (Schafer and Stevens, 2015). Spine dynamics of hippocampal CA1 neurons remain extremely high also beyond this pruning period with a turnover of spines every 1–2 weeks in 10–12-week-old mice (Attardo et al, 2015; Pfeiffer et al, 2018), a time still captured in our analysis. This implies that even minor microglial influences on CA1 synapse elimination, remodeling or formation would have been expected to produce a visible effect in the number of spines in $Csf1r^{\Delta FIRE/\Delta FIRE}$ mice. Nevertheless, we observed no morphological changes in synapse area and spine morphology, suggesting normal physical interactions between pre- and postsynaptic sites.

None of the previous studies providing evidence for microglial-dependent synapse elimination during postnatal development in the hippocampus, thalamus and visual cortex (Paolicelli et al, 2011; Schafer et al, 2012; Zhan et al, 2014; Sipe et al, 2016; Filipello et al, 2018; Liu et al 2021) investigated the consequences of permanent

absence of microglia during embryonic development and postnatal life on neuronal development and function. Although there is a wealth of evidence suggesting that microglia engulf synaptic structures (Faust et al, 2021), with CX3CR1 playing a major role in the hippocampus (Paolicelli et al, 2011; Zhan et al, 2014; Basilico et al, 2019), it is still unclear to what extent microglia or macroglia contribute to the elimination of excess synapses. Our findings in $Csf1r^{\Delta FIRE/\Delta FIRE}$ mice, which are free of potential confounding factors such as cell ablation, suggest that microglia are dispensable for the regulation of the number of synapses post-developmentally, consistent with recent findings showing that microglia in the developing hippocampus neither phagocytose postsynaptic material nor entire synapses (Weinhard et al, 2018). So far, no direct evidence, e.g., by real-time imaging, for the phagocytosis of excessive synapses by microglia during development has been provided (Eyo and Molofsky, 2023). Supporting our findings, the marked reduction of microglial numbers in mice lacking expression of TGF-ß or IL-34 did not result in obvious CNS developmental deficits under nondisease conditions (Wang et al, 2012; Butovsky et al, 2014). However, it is important to consider that the

**Table 1. Basal cytokine levels in the brains of *Csf1r*⁺/⁺, *Csf1r*⁺/ᐞ and *Csf1r*ᐞ/ᐞ mice.**

| | | +/+ | +/Δ | Δ/Δ |
|---|---|---|---|---|
| **Soluble fraction (pg/mg protein)** | **IFNγ** | 0.0188 ± 0.0003 | **0.0159 ± 0.0007\*** | **0.0156 ± 0.0007\*\*** |
| | IL-1β | 0.0942 ± 0.0027 | 0.0876 ± 0.0049 | 0.0924 ± 0.0036 |
| | TNFα | 0.0610 ± 0.0012 | 0.0538 ± 0.0030 | 0.0614 ± 0.0040 |
| | **IL-2** | 0.0700 ± 0.0021 | 0.0588 ± 0.0029 | **0.0589 ± 0.0022\*** |
| | IL-4 | 0.0277 ± 0.0023 | 0.0302 ± 0.0038 | 0.0205 ± 0.0035 |
| | IL-5 | 0.3477 ± 0.0079 | 0.3086 ± 0.0116 | 0.3118 ± 0.0115 |
| | IL-6 | 1.5144 ± 0.0359 | 1.3791 ± 0.0514 | 1.4355 ± 0.0496 |
| | **IL-10** | 0.0942 ± 0.0047 | 0.0934 ± 0.0095 | **0.1157 ± 0.0048\*** |
| | IL-12 | 2.9003 ± 0.0809 | 2.8181 ± 0.1824 | 3.1850 ± 0.1702 |
| | **CXCL1** | 0.8466 ± 0.1342 | **0.5605 ± 0.0125\*** | 0.7075 ± 0.0443 |
| | **ApoE** | 254980 ± 10723 | 282430 ± 10769 | **401370 ± 10279\*\*** |
| **Membrane-bound fraction (pg/mg protein)** | IFNγ | 0.0090 ± 0.0006 | 0.0079 ± 0.0003 | 0.0087 ± 0.0008 |
| | IL-1β | 0.0436 ± 0.0049 | 0.0349 ± 0.0018 | 0.0392 ± 0.0024 |
| | TNFα | 0.0714 ± 0.0045 | 0.0671 ± 0.0026 | 0.0704 ± 0.0061 |
| | IL-2 | 0.1009 ± 0.0094 | 0.0858 ± 0.0074 | 0.0793 ± 0.0036 |
| | **IL-4** | 0.0916 ± 0.0041 | 0.1049 ± 0.0113 | **0.1494 ± 0.0079\*\*** |
| | IL-5 | 0.3235 ± 0.1062 | 0.1914 ± 0.0050 | 0.2175 ± 0.0140 |
| | IL-6 | 0.9599 ± 0.0267 | 0.9502 ± 0.0228 | 1.0357 ± 0.0582 |
| | IL-10 | 0.2608 ± 0.0070 | 0.2542 ± 0.0093 | 0.2488 ± 0.0073 |
| | IL-12 | 2.1615 ± 0.0959 | 2.0723 ± 0.0787 | 2.1398 ± 0.0883 |
| | CXCL1 | 0.6317 ± 0.1193 | 0.4662 ± 0.0227 | 0.5892 ± 0.0403 |
| | **ApoE** | 309160 ± 10811 | 321460 ± 6277 | **499150 ± 7876\*\*** |

Significance levels indicate \*\*$P \leq 0.01$ and \*$P \leq 0.05$ (highlighted in bold). $P$ values are from unpaired Student's $t$ test with Bonferroni correction for multiple comparisons. $N = 16$ technical replicates from 8 mice for +/+ and Δ/Δ mice and 14 replicates from 7 mice for +/Δ, apart from IL-10 in Δ/Δ (15 replicates from 8 mice, soluble fraction), IL-4 in Δ/Δ (14 replicates from 8 mice, membrane fraction). ApoE values are from 8 replicates from 8 mice for +/+ and Δ/Δ, and 7 replicates from 7 mice for +/Δ.

mechanisms of microglia-synapse interactions critically depend on age, context and brain region (Faust et al, 2021; Mordelt and de Witte 2023), in line with a pronounced genetic heterogeneity of microglia in the developing brain (Li et al, 2019).

It has been proposed that targeting of synapses by microglia is not a random process but steered by synaptic activity (Tremblay et al, 2010; Schafer et al, 2012; Gunner et al, 2019), leading to the preferential elimination of weaker connections while sparing the stronger ones. Indeed, we find weaker and more immature synaptic connections in the hippocampal glutamatergic network in *Csf1r*^ΔFIRE/ΔFIRE^ mice. However, this is unlikely to result from a deficit in the elimination of weaker synaptic contacts, which would have caused an increase in spine density. Instead, the absence of microglia impairs glutamatergic transmission (further discussed below). These changes seem to persist into adulthood, as observed in adult mice with genetically deleted microglial CR3, CX3CR1 and TREM2 signaling (Paolicelli et al, 2011; Zhan et al, 2014; Filipello et al, 2018).

## Role of microglia-independent mechanisms in synaptic pruning

Which alternative cell types or mechanisms may regulate the pruning of synapses in brains devoid of microglia? Several types of macroglia, including astrocytes and oligodendrocyte precursor cells (OPCs), were identified as critical regulators of the pruning of synapses in the developing and adult brain (Chung et al, 2013; Buchanan et al, 2022). Notably, OPCs contain significantly more phagolysosomes filled with synaptic material than microglia in the developing mouse visual cortex (Buchanan et al, 2022). Astrocytes are more abundant than microglia and participate in activity-dependent synapse elimination and neural circuit formation in the developing and adult CNS by phagocytosing synapses via the MEGF10 and MERTK pathways (Chung et al, 2013; Lee et al, 2021). We find reactive astrocytes with increased expression of GFAP and MEGF10 in *Csf1r*^ΔFIRE/ΔFIRE^ mice at postnatal developmental stages when synaptic engulfment by microglia would have been highest (Jawaid et al, 2018). We also find elevated levels of ApoE and IL-10 in brains of *Csf1r*^ΔFIRE/ΔFIRE^ mice. While IL-10 plays a role in synapse formation (Lim et al, 2013), ApoE controls the rate of synaptic pruning by astrocytes (Chung et al, 2016) and can affect glutamatergic transmission in multiple ways via signaling through synaptic ApoE receptors (Lane-Donovan and Herz, 2017). It is tempting to speculate that astrocytes may at least in part compensate for the loss of microglia in *Csf1r*^ΔFIRE/ΔFIRE^ mice, consistent with reports showing an up-regulation of their phagocytic capacity in mice with dysfunctional or ablated microglia (Konishi et al, 2020; Berdowski et al, 2022).

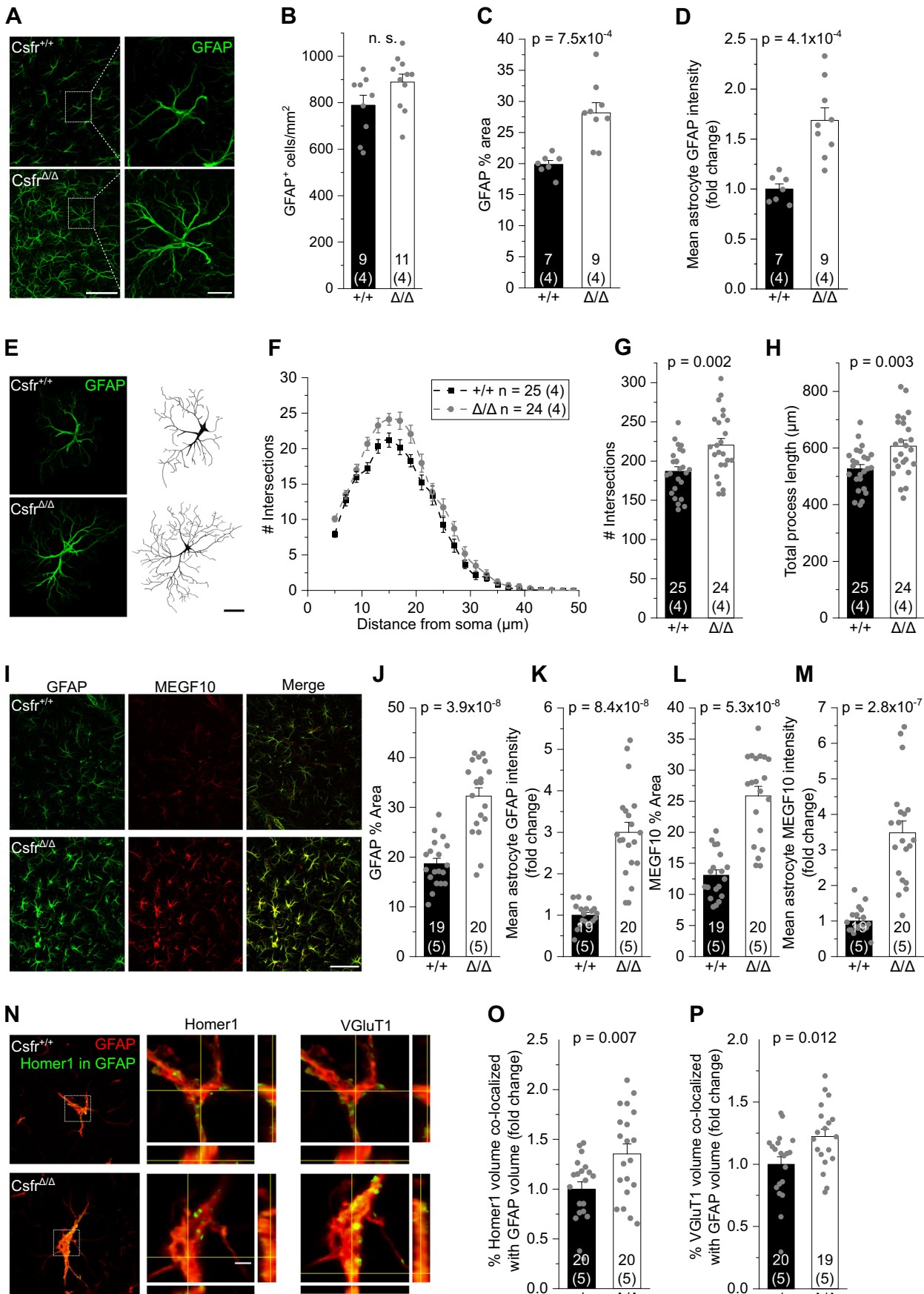

**Figure 7. Absence of microglia results in reactive astrocytes.**

(A) Specimen confocal images illustrating astrocytes in 6–10-week-old mice by GFAP immunoreactivity (green) in the CA1 *stratum radiatum*. Scale bars, 50 μm and 10 μm (for expanded view). (B–D) Analyses of GFAP$^+$ cell density (B), percentage of area covered by GFAP$^+$ astrocytes (C) and mean GFAP intensity (D). (E) Specimen confocal images illustrating astrocyte morphology by GFAP immunoreactivity (left) and their 3D reconstruction (right). Scale bar, 10 μm. (F–H) Sholl analysis-derived values of the number of intersections with Sholl radii at increasing distance from the soma (F) and the resulting total number of intersections (G) and total process length (H) of astrocytes. (I) Specimen confocal images illustrating astrocytes by GFAP (green) and MEGF10 (red) immunoreactivity in the CA1 *stratum radiatum* of mice aged P22-23. Scale bar, 50 μm. (J–M) Analyses of the percentage of area covered by GFAP$^+$ astrocytes (J), mean GFAP intensity (K), area covered by MEGF10$^+$ astrocytes (L) and mean MEGF10 intensity (M). (N) Left: Specimen confocal images showing Homer1 signal (green) within GFAP$^+$ astrocytes (red). Middle and right: Close-up images of astrocytes with orthogonal projections at the level of the crosshairs showing Homer1 (middle) or VGluT1 (right) signal within GFAP$^+$ astrocytes. Scale bars, 20 μm and 2 μm (for orthogonal projections). (O, P) Quantification of volume (%) of Homer1 (O) and VGluT1 (P) signal co-localizing with GFAP volume. Data information: Age of mice was 6–10 weeks (A–H) and P22–P23 (I–P) Data are represented as mean ± SEM. Numbers on bars indicate number of slices and (number of animals). *P* values are from unpaired Student's *t* tests. Source data are available online for this figure.

Future studies, ideally by high-resolution longitudinal live imaging of entire synapses, will help to determine what proportion of excess or unwanted synapses in development are subject to glial elimination as opposed to other processes, whether and how these processes differ between brain regions, and which specific functional properties of synapses determine their fate. As previously suggested, the simplified view that only weaker synaptic contacts are removed falls short of reflecting the high degree of structural and functional heterogeneity of synapses in neural networks (Wichmann and Kuner, 2022).

## Microglia affect glutamatergic network maturation in the hippocampus

The absence of microglia during embryonic development and postnatal life resulted in a distinct electrophysiological phenotype in *Csf1r*$^{\Delta FIRE/\Delta FIRE}$ mice, with weakened and immature glutamatergic hippocampal transmission compared to WT littermates. CA1 pyramidal cells in *Csf1r*$^{\Delta FIRE/\Delta FIRE}$ mice received less excitatory input from the CA3 region and are less capable of generating excitatory output. While presynaptic glutamate release probability was preserved, we identified changes in pre- and postsynaptic function, as evidenced by impairment of multivesicular release and reduction of NMDAR-mediated postsynaptic charge transfer. In contrast, we found no changes in inhibitory GABAergic synaptic input into CA1 pyramidal neurons in *Csf1r*$^{\Delta FIRE/\Delta FIRE}$ mice.

Interestingly, the electrophysiological profile in *Csf1r*$^{\Delta FIRE/\Delta FIRE}$ mice closely mirrors the changes in the hippocampal transmission that emerged after transient depletion of microglia in adult mice that had undergone normal development (cf. Table 2 for a detailed comparison). This suggests a role for microglia in supporting glutamatergic transmission in the adult brain but not in the developing brain. Along these lines, postnatal depletion of microglia selectively impaired excitatory transmission, with CA1 pyramidal cells receiving less glutamatergic input from CA3 Schaffer collaterals, whereas inhibition was unaffected (Parkhurst et al, 2013; Basilico et al, 2022; Ma et al, 2020). Congruent with our findings in *Csf1r*$^{\Delta FIRE/\Delta FIRE}$ mice, this was related to impairment in multivesicular release and changes in postsynaptic NMDA receptors without affecting the AMPAR component (Parkhurst et al, 2013; Basilico et al, 2022; Ma et al, 2020). The observed reduction in the postsynaptic NMDAR component may be related to an increase in ambient glutamate levels, which is mainly regulated by astrocytes (Le Meur et al, 2007) and could reflect the altered astrocytic properties in *Csf1r*$^{\Delta FIRE/\Delta FIRE}$ mice. In consequence,

NMDAR desensitization would be increased, leaving fewer NMDARs available for synaptic activation (Cavelier et al, 2005). Since no desensitization of AMPARs was observed in *Csf1r*$^{\Delta FIRE/\Delta FIRE}$ mice, this would apply to glutamate levels below the AMPAR desensitization threshold (≤2 μM; Colquhoun et al, 1992). Apart from the synapse, one-third of NMDARs in pyramidal cells are also extrasynaptically localized and contribute significantly to tonic NMDAR-mediated currents (Harris and Pettit, 2007; Le Meur et al, 2007). Notably, microglia depletion also seems to affect glutamatergic and GABAergic transmission differentially in the visual cortex compared to the hippocampus and motor cortex (Table 2).

A noticeable feature in both *Csf1r*$^{\Delta FIRE/\Delta FIRE}$ and microglia-depleted WT mice is the impairment of multivesicular release (Basilico et al, 2022; Table 2), a hallmark of mature hippocampal synaptic transmission (Hsia et al, 1998; Rigby et al, 2023). This may be due to a preferential interaction of microglia with presynaptic structures, contributing to the formation of multi-synaptic boutons in the developing hippocampus (Zhan et al, 2014; Weinhard et al, 2018). Related to this, activated microglia have been suggested to displace presynaptic terminals in a process called synaptic stripping (Trapp et al, 2007; Chen et al, 2014).

Despite the lack of overt behavioral abnormalities in *Csf1r*$^{\Delta FIRE/\Delta FIRE}$ mice (Rojo et al, 2019), an obvious assumption related to the altered electrophysiological profile of CA1 pyramidal cells are alterations in hippocampal-related behavior. However, we observed no differences in nonspatial memory in *Csf1r*$^{\Delta FIRE/\Delta FIRE}$ mice. Importantly, spatial learning and memory encoding were also normal in *Csf1r*$^{\Delta FIRE/\Delta FIRE}$ mice (McNamara et al, 2023). These findings are reminiscent of a previous study of depleting microglia in young WT mice, which did not result in behavioral deficits, but in fact even showed slight improvements in the Barnes maze and contextual fear conditioning (Elmore et al, 2014).

Finally, our data suggest changes in immunomodulatory molecules as a result of the absence of microglia in *Csf1r*$^{\Delta FIRE/\Delta FIRE}$ mice. These include increases in ApoE levels in both soluble and membrane-bound brain fractions of *Csf1r*$^{\Delta FIRE/\Delta FIRE}$ mice, suggesting increased release from astrocytes as the main producers of ApoE in the brain (Boyles et al, 1985; Pitas et al, 1987; Zhang et al, 2014; Zhang et al, 2016). Due to its paramount role in brain lipid metabolism and the ability to suppress the release of pro-inflammatory cytokines (Laskowitz et al, 1997; Laskowitz et al, 1998; Flowers and Rebeck 2020), the increase in ApoE is expected to promote anti-inflammatory processes and neural homeostasis. Indeed, we observed increases in IL-4 and IL-10 along with decreases in INF-γ and IL-2 in the brains of *Csf1r*$^{\Delta FIRE/\Delta FIRE}$ mice. Related to neural function, ApoE has been implicated in controlling

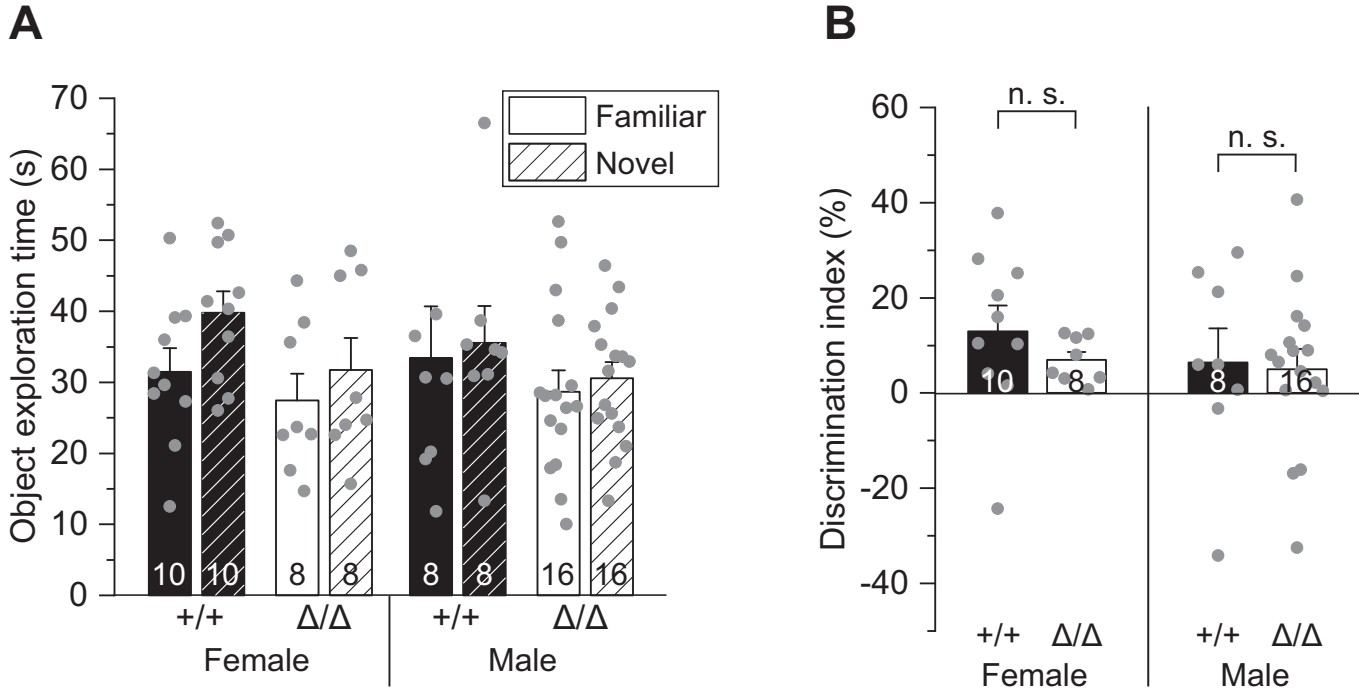

**Figure 8.  Absence of microglia does not alter object recognition memory.**

(A) Comparison of exploration times of the familiar and novel object after habituation for male and female Δ/Δ and +/+ mice (see "Methods"). (B) Comparison of the respective discrimination indices (see "Methods"). Data information: Data are represented as mean ± SEM. Numbers on bars indicate number of tested animals. *P* values are from two-way ANOVA (B). Source data are available online for this figure.

the phagocytosis of astrocytes (Chung et al, 2016) and glutamate signaling through ApoE receptors, affecting pre- and postsynaptic function (Lane-Donovan and Herz, 2017). For example, LDLR-related receptor 1 (Lrp1) controls the surface distribution and internalization of NMDARs (Maier et al, 2013), while cholesterol complexed to ApoE acts as a regulator of presynaptic function (Mauch et al, 2001). It is important to consider that the observed changes reflect the situation in the healthy brain and are not complicated by the engraftment of peripheral immune cells in $Csf1r^{\Delta FIRE/\Delta FIRE}$ mice (Parkhurst et al, 2013; Spangenberg et al, 2019; Konishi et al, 2020). This is likely to be different in the presence of brain damage or disease, in which the permanent absence of microglia has been associated with accelerated pathology in a dementia model and in prion disease (Spangenberg et al, 2019; Bradford et al, 2022; Shabestari et al, 2022).

Collectively, our findings provide evidence that microglia are not strictly required for synaptic pruning in the developing hippocampus and suggest that microglia-independent mechanisms, which may involve macroglia, contribute to a basic, albeit not fully mature, network connectivity without causing behavioral disturbances.

## Methods

### Mice

Experiments used transgenic $Csf1r^{\Delta FIRE/\Delta FIRE}$ mice (Rojo et al, 2019) which were on a mixed C57BL/6J x CBA/J background with

littermate heterozygous $Csf1r^{\Delta FIRE/+}$ and littermate wild-type $Csf1r^{+/+}$ controls of both sexes aged 6–10 weeks, or P9–P10 for experiments in Fig. 1M,N and P22–P23 for Fig. 7I–P. Mice were housed in groups in individually ventilated cages under specific-pathogen-free conditions on a 12-h light/dark cycle with food and water *ad libitum*. All procedures involving handling of living animals were carried out in accordance with the UK and German animal protection law and approved by the local authorities for health and social services (LaGeSo T-CH 0043/20).

### Novel object recognition test

The novel object recognition (NOR) test was performed in a dedicated behavioral testing room using mice of both sexes, with females and males always being tested on different days and analyzed separately. This test was performed on mice at 1–2 months of age to assess nonspatial memory. A total of 24 $Csf1r^{\Delta FIRE/\Delta FIRE}$ mice (16 male, 8 female), 29 $Csf1r^{\Delta FIRE/+}$ littermates (14 male, 15 female), and 18 $Csf1r^{+/+}$ littermates (8 male, 10 female) were tested. Mice were housed under standard conditions, with 12-h light/dark cycles, in temperature and humidity-controlled rooms. Researchers were blinded to experimental groups and remained blinded during data analysis. Handling was carried out 3–4 days prior to testing to reduce handling-induced stress on testing days. The equipment was cleaned with 70% ethanol between tests. Before starting this test, mice were run on an open field (47 cm × 47 cm), which habituated them to the arena. Mice were then placed back in the open field on a separate day and habituated to two identical objects, Lego towers.

**Table 2. Similar electrophysiological profile in the hippocampus of Csf1r$^{\Delta FIRE/\Delta FIRE}$ mice compared to data from wild-type mice with microglial depletion at comparable ages.**

| | Present study | Basilico et al, 2022 | Du et al, 2022 | Parkhurst et al, 2013 | Liu et al, 2021 |
|---|---|---|---|---|---|
| Brain region | Hippocampus CA1/3 | Hippocampus CA1/3 | Hippocampus CA1/3 | Motor cortex L5 | Visual cortex L5 |
| Age | 6–10 weeks | 6–10 weeks | 4–5 weeks | 3–6 weeks | 9–10 weeks |
| Microglial depletion | Throughout life, 100% | Transient (≈ 85–90%), postnatally/adolescence | | | |
| Glutamatergic transmission | ↓ | ↓ | ↓ | ↓ | ↑ |
| ▪ I/O (CA3–CA1) | ↓ | ↓ | ↓ | n.d. | n.d. |
| ▪ release probability | – | ↑ | – | n.d. | n.d. |
| ▪ mEPSC (AMPAR) | Freq – amp – | Freq – amp – | n.d. | Freq ↓ amp – | n.d. |
| ▪ multivesicular release | ↓ | ↓ | n.d. | n.d. | n.d. |
| ▪ NMDAR (postsyn.) | ↓ | ↓ | n.d. | ↓ | n.d. |
| GABAergic transmission | — | — | n.d. | n.d. | ↑ |
| ▪ m/sIPSCs | – | – | n.d. | n.d. | n.d. |

Key to symbols and abbreviations:
↓ / ↑ / –, increase/decrease/no change, *n.d.* not determined.
*mEPSC* miniature excitatory postsynaptic current, *m/sIPSC* miniature/spontaneous inhibitory postsynaptic curren, *freq* frequency, *amp* amplitude, *I/O* Input/Output relationship referring to CA1 compound EPSCs in response to CA3 electrical stimulation, *AMPA* alpha-amino-3-hydroxy-5-methyl-4-isoxazolpropionic acid, *NMDAR* N-methyl-ᴅ-aspartate receptor, *GABA* gamma-aminobutyric acid.

The following day (the testing day), one Lego tower was replaced by a light bulb as a novel object and mice were left to freely explore the novel and familiar objects for 10 min. Analyses of the acquired behavior videos were performed using Any-maze software. Exploration of an object was timed when the head of the mouse was within 1 cm of the object. Exploration time was not counted when the mouse was grooming near the object (it was only counted when the mouse was sniffing, examining, touching, or climbing on the object). No outliers were removed. Discrimination indices (DI) were calculated according to the formula: [DI = (Novel Object Exploration Time/Total Exploration Time) - (Familiar Object Exploration Time/Total Exploration Time) × 100].

### Brain slice preparation

Mice were decapitated under isoflurane anesthesia. Whole brains were rapidly removed from the skull and immediately immersed in ice-cold slicing solution. Acute horizontal brain slices (300 μm) of the ventral hippocampus were prepared according to the protocol of Bischofberger et al (2006) using a sucrose-based slicing solution containing (mM): 87 NaCl, 25 NaHCO$_3$, 2.5 KCl, 0.5 CaCl$_2$, 3 MgCl$_2$, 1.25 NaH$_2$PO$_4$, 10 glucose, 75 sucrose, pH 7.4, bubbled with 95%O$_2$/5%CO$_2$ at <4 °C. After cutting, slices were allowed to recover for 30 min in warmed (34–36 °C) slicing solution and then kept at room temperature until experimental use.

### External and intracellular solutions

For all experiments, slices were superfused with bicarbonate-buffered artificial cerebrospinal fluid (ASCF) at 34–36 °C containing (mM): 124 NaCl, 2.5 KCl, 26 NaHCO$_3$, 1 NaH$_2$PO$_4$, 2 CaCl$_2$, 1 MgCl$_2$, 10 glucose, bubbled with 95%O$_2$/5%CO$_2$.

For excitatory currents, cells were recorded with a potassium gluconate-based intracellular solution containing (mM): 120 K-gluconate, 10 HEPES, 10 EGTA, 2 MgCl$_2$, 5 Na$_2$-phosphocreatine, 2 Na$_2$ATP, 0.5 Na$_2$GTP, 5 QX-314 Cl or KCl (for excitability

measurements). In some experiments, 0.1% Biocytin was added for post-hoc anatomical visualization and reconstruction of neuronal morphology.

For inhibitory currents, a KCl-based intracellular solution was used containing (mM): 125 KCl, 4 NaCl, 10 HEPES, 10 EGTA, 1 CaCl$_2$, 4 MgATP, 0.5 Na$_2$GTP. Intracellular solutions were adjusted to a final osmolarity of 285 ± 5 mOsmol/L and a pH of 7.2.

### Electrophysiology

Hippocampal neurons in slices were visualized by IR-DIC optics using an upright Scientifica SliceScope equipped with a ×63 water immersion objective (N.A. 1.0) and Olympus XM10 camera.

Whole-cell recordings from pyramidal neurons were obtained at a depth of >40 μm below the slice surface using borosilicate glass pipettes with a tip resistance of 2.5–3.5 MΩ, resulting in access (series) resistances of <20 MΩ that were compensated by ~60%. Cells with changes in series resistance >20% were excluded from the analysis. Voltage- and current-clamp recordings were performed using a Multiclamp 700B amplifier (Molecular Devices). Currents were filtered at 2 kHz (10 kHz for membrane test), digitized (20 kHz), and analyzed off-line using pClamp10 software. Resting membrane potential was measured in current-clamp mode immediately after breaking into the cell. Electrode junction potentials were not compensated and were 14.5 mV and 3 mV for K-gluconate- and KCl-based intracellular solutions, respectively (determined using the *LJP* tool in pClamp10).

Input (Rt) and series resistances (Rs) were analyzed from voltage-clamped cells held at −70 mV by applying 10 mV hyperpolarizing voltage steps using Ohm's law (R = V/I). Total (input) resistances (Rt) and (Rs) were calculated as follows: Rt = V (applied voltage step)/Is (steady-state current), Rs = V (applied voltage step)/Ip (transient peak capacitive current). Cell capacitance (Cm) was calculated from the time constant (tau) of the current transient decay using a mono-exponential fit according to Cm = tau/(Rs*Rm/Rs+Rm) = tau*(Is+Ip)$^2$/(V*Is).

For determining neuronal excitability, CA1 pyramidal cells were held in current-clamp mode and voltage responses filtered and digitized at 10 and 20 kHz, respectively. Action potentials were elicited by injecting increasing steps of current (0.5 s) from $-25$ pA to 250 pA with increments of 25 pA at the cells' resting potential. The threshold potential was extracted from the first action potential elicited in response to a series of depolarizing current injections and was measured on the ascending phase where the slope exceeded 20 mV/ms. The first current stimulus step that elicited at least one action potential was considered the rheobase (Planert et al, 2023, bioRxiv).

Excitatory postsynaptic currents (EPSCs) in CA1 pyramidal cells were triggered by electrical stimulation of CA3 Schaffer collaterals at increasing current injections ranging from 10 to 100 μA for 0.2 ms using Tungsten concentric electrodes (WE3CEA3-200, Science Products) connected to a stimulus isolation unit (WPI A320). CA1 pyramidal cells were voltage-clamped at the estimated equilibrium potential for [Cl⁻] at $-70$ mV to largely suppress inhibitory GABAergic currents.

To determine paired-pulse ratios, half-maximal CA1 EPSCs were evoked by paired stimuli at an inter-stimulus interval of 50 ms. Pulses were triggered at 10 s intervals and six individual traces were collected and averaged to determine the EPSC amplitudes and their ratio.

Spontaneous EPSCs (sEPSCs; action potential-dependent and -independent events) and miniature EPSCs (mEPSCs; action potential-independent) were recorded in CA1 pyramidal cells voltage-clamped at $-70$ mV. For mEPSCs, 0.3 μM tetrodotoxin (TTX) was added to the ACSF to block action potentials. Synaptic currents were identified using the template search routine in Clampfit (pClamp10). AMPA current decay was determined using a mono-exponential fit.

AMPA/NMDA ratios were determined from mEPSCs by first measuring mixed AMPA- and NMDA receptor-mediated currents in Mg²⁺-free ACSF containing 0.3 μM TTX, 10 μM glycine and 10 μM gabazine, followed by application of 50 μM D-AP5 to isolate the AMPA current component. Average mEPSCs determined from 50 individual events per condition were superimposed and time-matched for current onset to calculate the NMDA receptor component by subtracting the average AMPA mEPSC from the average mixed AMPA + NMDA mEPSC. Individual AMPA and NMDA receptor-mediated components were analyzed as charge entering the cell by calculating respective areas under the mEPSC using the *Area* function in Clampfit (pClamp10).

For single event analysis of mixed AMPA/NMDA synapses, the peak of every mEPSC (AMPA peak) was measured as the mean current over a 1 ms window. NMDA current amplitude (NMDA measurement) was measured over a 5 ms window from 22 to 27 ms after the AMPA peak, a time at which isolated AMPA-mediated mEPSCs had almost completely terminated and declined to $1.08 \pm 0.42$ pA ($n = 15$) (Myme et al, 2003). Only events with peak amplitudes of $\geq -30$ pA that were significantly larger than the root mean square (RMS) noise of baseline current under Mg²⁺-free conditions were included in this analysis.

To analyze tonic NMDA receptor currents, D-AP5 sensitive RMS noise was measured by averaging holding (baseline) current amplitudes in ten different regions of 100 ms duration lacking postsynaptic currents, before and after application of D-AP5. D-AP5-sensitive current was obtained by determining the change in mean holding current in a 10 s window before and after D-AP5 application.

## Immunohistochemistry and confocal microscopy

Slices were fixed overnight in 0.1 M phosphate-buffered saline (PBS, 0.9% NaCl) containing 4% paraformaldehyde and 4% sucrose at 4 °C. After washing with PBS ($3 \times 15$ min), slices were transferred into blocking solution (10% Normal Goat Serum (NGS) and 0.5% Triton-X in PBS) for 1 h. Thereafter, slices were incubated with primary antibody in PBS containing 5% NGS and 0.3% Triton-X for 48–72 h at 4 °C. Microglia were labeled using polyclonal rabbit anti-Iba-1 (cat. No. 234 003, 1:1000; Synaptic Systems). Astrocytes were labeled using either monoclonal guinea pig anti-GFAP (cat. No. 173 308, 1:1000; Synaptic Systems), or monoclonal rat anti-GFAP (2.2B10 (13-0300), 1:1000; Invitrogen) and polyclonal rabbit anti-MEGF10 (ABC10, 1:500; Sigma). Excitatory pre- and post-synapses were labeled using polyclonal guinea pig anti-VGluT1 (cat. No. 135 304, 1:500; Synaptic Systems) and polyclonal rabbit anti-Homer1 (cat. No. 160 002, 1:500; Synaptic Systems). Inhibitory pre- and post-synapses were labeled using polyclonal guinea pig anti-VGAT (cat. No. 131 004, 1:500; Synaptic Systems) and monoclonal mouse anti-Gephyrin (cat. No. 147 111, 1:500; Synaptic Systems). Subsequently, slices were rinsed several times with PBS and then incubated with secondary antibodies in PBS containing 3% NGS and 0.1% Triton-X overnight, using Alexa Fluor-488 goat anti-rabbit (A-11034, 1:1000; Invitrogen) for Microglia. Alexa Fluor-488 goat anti-rabbit (A-11034, 1:500; Invitrogen) and Alexa Fluor-647 goat anti-guinea pig (A21450, 1:500; Invitrogen) were used for labeling excitatory synapses. Alexa Fluor-488 goat anti-guinea pig (A11073, 1:500; Invitrogen) and Alexa Fluor-647 goat anti-mouse (A21235, 1:500; Invitrogen) were used for labeling inhibitory synapses. For labeling of astrocytes either Alexa Fluor-488 goat anti-guinea pig (A11073, 1:1000; Invitrogen) or Alexa Fluor-568 goat anti-rat (A11077, 1:1000; Invitrogen) for GFAP and Alexa Fluor-647 goat anti-rabbit (A21245, 1:500; Invitrogen) for MEGF10 was used. After washing in PBS, slices were mounted onto 300 μm-thick metal spacers (Bolduan et al, 2020) using aqueous mounting medium Fluoromount-G (Southern Biotech, Birmingham, USA) or Fluor-oshield mounting medium with DAPI (Abcam, Cambridge, UK) and stored at 4 °C.

Confocal images were collected using Olympus ×30 (silicon oil-immersion, 1.05 N.A., 0.8 mm W.D.) and ×60 objectives (silicon oil-immersion, 1.35 N.A., 0.15 mm W.D.) on an upright confocal microscope (BX61, Olympus, Tokyo, Japan) with FluoView FV1000 V4.2 acquisition software (Olympus), through separate channels and temporally non-overlapping excitation of the fluorophores, to prevent nonspecific signals. DAPI and Alexa Fluor-647 were excited with diode lasers at 405 nm and 635 nm, respectively, and Alexa Fluor-488 with a multi-line argon laser at 488 nm.

## 3D reconstruction and spine density analysis

Morphology of CA1 pyramidal cell dendrites was analyzed by filling patch-clamped cells with biocytin (0.1%, ≥20 min) added to the intracellular solution. After the recordings, slices were fixed and the neurons visualized using Streptavidin-Alexa Fluor-647 con-jugate (1:1000; Molecular Probes, Eugene, Oregon, USA) which was

included with the secondary antibodies during immuno-labeling as outlined above. High-resolution confocal image stacks using ×30 or ×60 objectives were taken to visualize pyramidal cell morphology with their apical dendrites and representative fractions of the dendritic shaft and oblique branches. Image stacks were stitched in Fiji and neurons 3D-reconstructed using neuTube software (https://neutracing.com). Per individual neuron, five representative regions of interest comprising 15–20 μm long segments of apical dendritic regions were analyzed. For each individual segment, a 3D skeleton was generated and analyzed for spine number and length using Fiji. Final values for spine density and length per individual cell reflect averages from all five segments.

## Analysis of microglial and astrocyte morphology in brain slices

Slices of every 3D image stack were filtered with a median filter, background subtracted and then transformed into 2D by maximum intensity projection. Microglial coverage was measured through binarization of the images using the Otsu algorithm, and the percentage of GFAP area per total area was calculated. Microglial ramification was determined as an index defined as the ratio of the perimeter to the area, normalized by that same ratio calculated for a circle of the same area (Madry et al, 2018). GFAP and MEGF10 intensity was determined as the mean intensity within the GFAP area in 2D-projected images. Microglial morphological parameters were obtained from 3D-Morph automatic image analysis (York et al, 2018). Cell counts were performed in 3D image stacks normalized to a given volume.

## Synapse quantification

Brain slices of 300 μm thickness were cryoprotected in 30% sucrose/PBS solution and re-sectioned at 70 μm using a cryotome (Microm 340E + Microm KS 34, Thermo Scientific™). The outer slice regions that were most strongly affected by the initial slicing procedure were discarded, and only the inner sections were kept for further use. Image stacks consisting of 12 image planes at 0.25 μm axial ($z$ axis) intervals were acquired from the CA1 stratum radiatum starting 3 μm below the slice surface using a Leica SP5 confocal microscope with a HCX PL APO lambda blue 63 × 1.4 Oil objective with additional 3.5× zoom (image field 67.2 μm × 67.2 μm at 1024 × 1024 resolution). Image processing was done using Fiji software. First, images were median filtered ("despeckled") to remove background noise and maximum intensity projected. Synapses were identified by the colocalization of fluorescent punctae for pre- and postsynaptic markers using the Synapse-Counter plugin. This plugin includes image pre-processing (subtraction of smooth continuous background, "Subtract Background"-Plugin; rolling ball radius = 10), binarization and creation of an additive mask for automatic counting of colocalized pre- and postsynaptic elements. Size exclusion ($>1.2\ \mu m^2$ and $<0.1\ \mu m^2$) was applied to exclude any objects unlikely to represent synapses (McLeod et al, 2017).

## Analysis of astrocytic synapse uptake

Image stacks consisting of 9 image planes at 0.25 μm axial ($z$ axis) intervals were acquired from the CA1 stratum radiatum starting

5 μm below the slice surface using a Leica SP5 confocal microscope with a HCX PL APO lambda blue 63 × 1.4 Oil objective with additional 3.5× zoom (image field 67.2 μm × 67.2 μm at 1024 × 1024 resolution). Image analysis was done using Imaris. First, GFAP signal was manually thresholded using the surface tool to include primarily the astrocytes somata without including the finer processes to exclude VGlut1 and Homer1 punctae, respectively, merely touched by these processes. Then, signals for VGlut1 and Homer1 where masked to remove all pre- and postsynaptic punctae not co-localizing with the thresholded GFAP surface. The remaining signals for VGlut1 and Homer1 where then manually thresholded using the surface tool. The data points show the respective volumes of VGlut1 and Homer1, per volume of GFAP in the field of view relative to $Csf1r^{+/+}$ littermates.

## Sholl analysis

CA1 pyramidal cell morphology was assessed by Sholl analysis (Sholl, 1953) using the built in function of the Simple Neurite Tracer plugin in Fiji. Brain slices with biocytin-filled cells were prepared and PFA-fixed as described above. Confocal image stacks of individual CA1 pyramidal cells were acquired at 0.8- μm depth intervals. Image stacks were converted into 8-bit in Fiji and saved as TIF files.

## ELISA-based analysis of basal cytokine and ApoE levels

After removal from the skull, separated brain hemispheres were immediately snap-frozen in liquid nitrogen and kept at −80 °C until experimental use. Proteins were extracted from one hemisphere (excluding the cerebellum and olfactory bulb) using a multi-step protocol as described by Kawarabayashi et al (2001). Briefly, hemispheres were mechanically dissociated with a tissue homogenizer and 1 ml syringe with G26 cannulas. Brain homogenates were sequentially processed in Tris-buffered saline (TBS) followed by Triton-X-containing Tris buffer (TBX) for extraction of soluble (TBS) and membrane-bound proteins (TBX). Proteins dissolved in each buffer were extracted by ultracentrifugation at $100,000 \times g$ for 45 min. Supernatants of respective TBS and TBX fractions were collected and stored at −80 °C for further analyses. Cytokine and ApoE concentrations of TBS and TBX fractions were determined without further dilution (cytokines) or diluted 1:1000 (ApoE), using V-PLEX Pro-inflammatory Panel 1 (Meso Scale Discovery, catalog no. K15048D1) and mouse apolipoprotein E ELISA kit (abcam, catalog no. ab215086) according to the manufacturer's instructions. Cytokines and ApoE levels were normalized to the total amount of protein determined by BCA protein assay (Thermo Scientific, Pierce, catalog no. 23225).

## Statistics

Data are presented as means + s.e.m. Experiments on mice of different genotype were interleaved and randomized. Statistical tests were performed using OriginPro 9.8 and IBM SPSS statistics (Version: 29.0.0.0-241). $P$ values are from unpaired Student's $t$ tests (for normally distributed data) or Mann–Whitney $U$ tests (for non-normally distributed data). Normality of data was determined using Anderson–Darling and Shapiro–Wilk tests and equality of variance confirmed using the $F$-test. Two-way ANOVA followed by

Bonferroni post-hoc analysis was used for analysis of the input–output relationship and excitability measurements. For multiple comparisons, *P* values were corrected using an equivalent to the Holm-Bonferroni method (for N comparisons, the most significant p value is multiplied by N, the 2nd most significant by N-1, the 3rd most significant by N-2, etc.). For analysis of single event mixed AMPA/NMDA mEPSC, Pearson correlation and analysis of covariance (ANCOVA) was used. Estimation of an appropriate sample size for a standard experiment was as follows: For a control response of 100%, a typical response standard deviation of 25%, an effect size of 50%, a power of 80%, and a significance level of $P < 0.05$, at least six cells are required in both groups (http://www.biomath.info/power/ttest.htm). Actual numbers may vary for different experiments depending on effect size and standard deviation. *P* values were considered significant at ≤0.05. Where appropriate, analyses were performed with the researcher blind to experimental condition.

## Data availability

This study has not generated data that requires deposition in a public database.

## Peer review information

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

## Acknowledgements

The authors wish to thank Henrik Alle, David Attwell, Jörg Geiger, and David Hume for comments on the manuscript, Clare Pridans for donating *Csf1r*^ΔFIRE/ΔFIRE mice and Christian Böttcher, Mehreen Mohammad and Andrea Wilke for expert technical assistance. This work was supported in part by grants from the German Research Foundation (SFB/TRR 167 A10N to CM and B07 to JP), the BMBF/DLR (FZ: 01EE2303B to JP), the BIH (fellowship to AR) and the UK DRI (Programme Award to JP).

## Author contributions

**Michael Surala**: Formal analysis; Investigation; Visualization; Writing—review and editing. **Luna Soso-Zdravkovic**: Formal analysis; Investigation; Visualization. **David Munro**: Formal analysis; Investigation; Visualization. **Ali Rifat**: Formal analysis; Investigation; Visualization. **Koliane Ouk**: Formal

analysis; Investigation. **Imre Vida**: Validation; Methodology; Writing—review and editing. **Josef Priller**: Conceptualization; Supervision; Funding acquisition; Project administration; Writing—review and editing. **Christian Madry**: Conceptualization; Data curation; Supervision; Funding acquisition; Validation; Writing—original draft; Project administration; Writing—review and editing.

## Funding

## Disclosure and competing interests statement
The authors declare no competing interests.

# Expanded View Figures

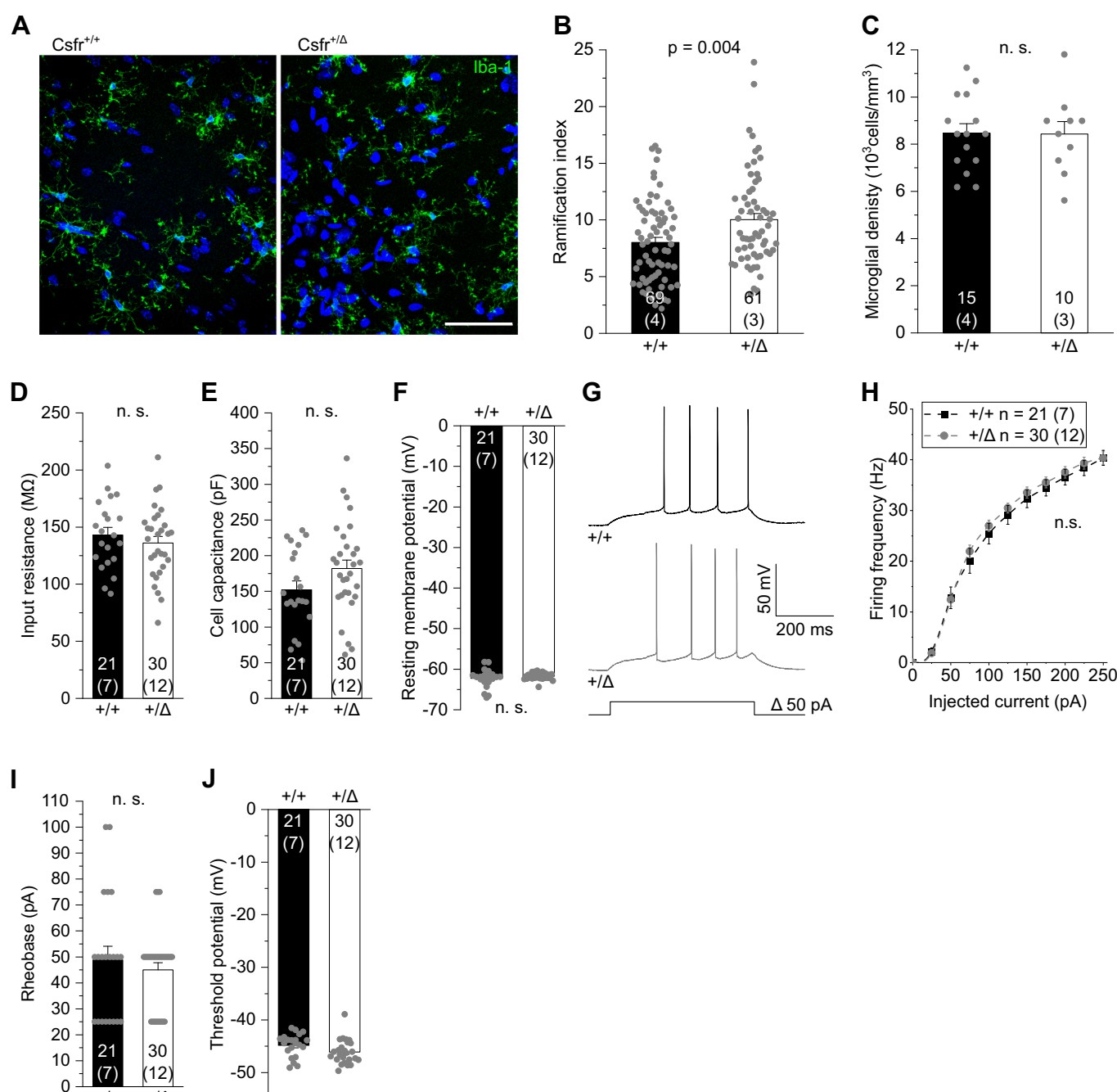

**Figure EV1.  Normal excitability of CA1 pyramidal cells in heterozygous *Csf1r*$^{+/\Delta FIRE}$ mice.**

(**A**) Specimen confocal images illustrating microglia by Iba1 immunoreactivity (green) in acute hippocampal slices of *Csf1r*$^{+/\Delta FIRE}$ (+/Δ) and WT littermates (+/+). DAPI labeling of cellular nuclei in blue. Scale bar, 50 μm. (**B**, **C**) Quantification of microglial ramification (**B**) and cell density (**C**) in the CA1 *stratum radiatum*. (**D**–**F**) Analysis of input resistance (**D**), cell capacitance (**E**) and resting membrane potential (**F**) of CA1 pyramidal cells. (**G**) Specimen traces showing action potential firing patterns of CA1 pyramidal cells in response to 500 ms depolarizing current injections. (**H**). Corresponding course of action potential firing frequencies of CA1 pyramidal cells on increasing depolarizations. (**I**, **J**) Values for rheobase (**I**) and action potential threshold voltage (**J**). Data information: Data indicate mean ± SEM. Numbers on bars show tested cells or number of slices (**C**) and (number of animals). *P* values are from unpaired Student's *t* (**B**–**F**, **J**) or Mann–Whitney tests (**I**) and two-way ANOVA (**H**).

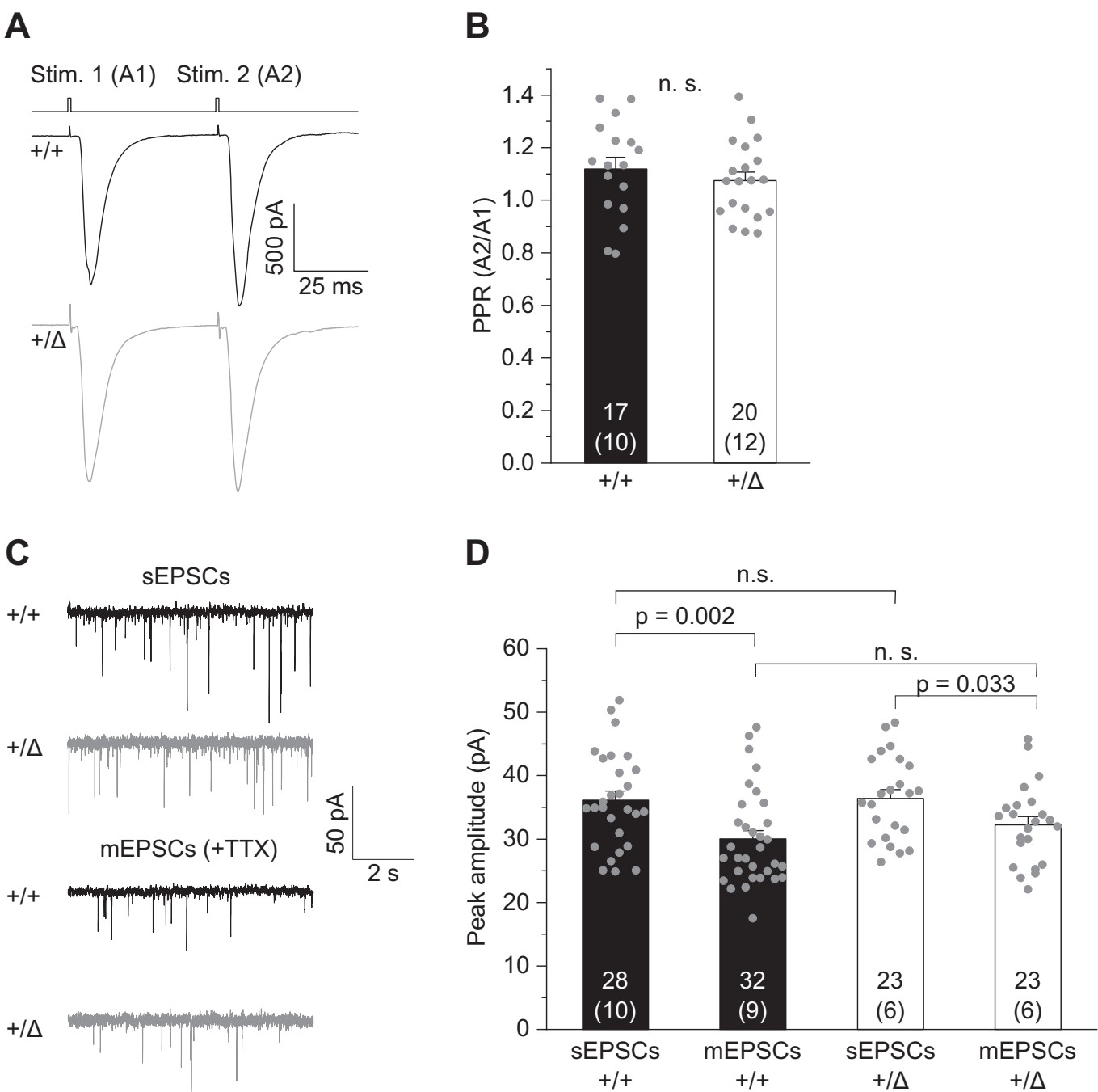

**Figure EV2. Normal glutamatergic transmission in heterozygous *Csf1r*<sup>+/ΔFIRE</sup> mice.**

(A) Example traces (EPSCs) of CA1 pyramidal cells after paired-pulse stimulation at 50 ms inter-stimulus intervals. (B) Comparison of paired-pulse ratios (PPR) as the quotient of the second vs first EPSC amplitude (A2/A1) of CA1 pyramidal cells. (C) Specimen traces showing AMPAR-mediated spontaneous EPSCs (sEPSCs) and miniature EPSCs (mEPSCs) in the presence of 300 nM TTX of CA1 pyramidal cells. (D) Comparison of peak amplitudes of sEPSCs and mEPSCs. Data information: Data are represented as mean ± SEM. Numbers on bars indicate tested cells and (number of animals). *P* values are from unpaired Student's *t* (B, D, for +/Δ and sEPSC comparison) or Mann–Whitney tests (D, for +/+ and mEPSC comparison).

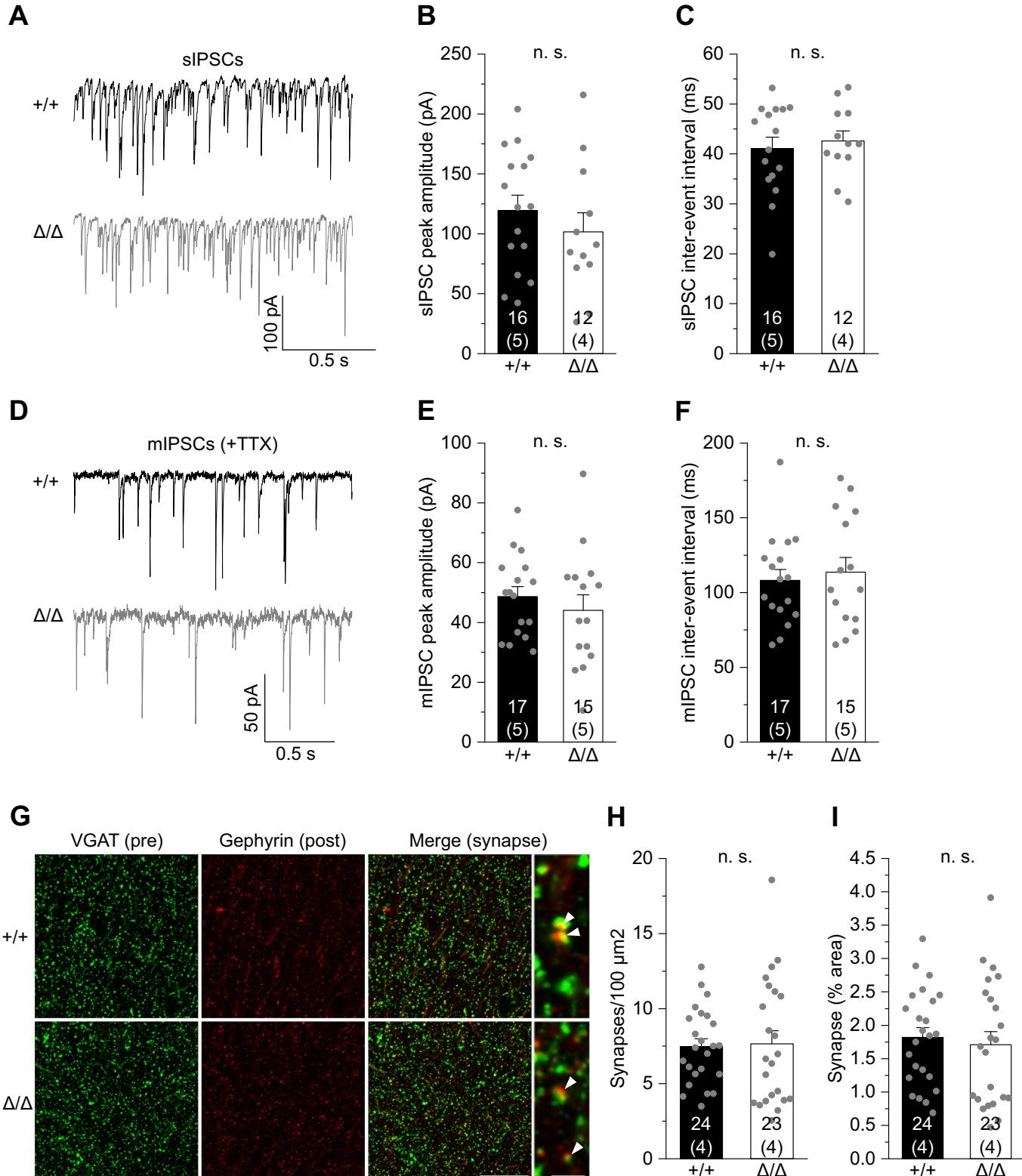

◀ **Figure EV3.   Unaltered inhibitory synaptic transmission of CA1 pyramidal cells in *Csf1r*[ΔFIRE/ΔFIRE] mice.**

(A) Specimen traces showing GABA$_A$ receptor-evoked spontaneous inhibitory postsynaptic currents (sIPSCs) of CA1 pyramidal cells in $+/+$ and Δ/Δ mice. (B, C) Comparison of sIPSC peak amplitudes (B) and inter-event intervals (C). (D) Specimen traces showing miniature IPSCs (mIPSCs) of CA1 pyramidal cells. (E, F) Comparison of peak amplitudes (E) and inter-event intervals (F). (G) Confocal images showing VGAT-labeled presynaptic puncta (green) and Gephyrin-labeled postsynaptic puncta (red) in the CA1 *stratum radiatum* (left and middle) and colocalization of puncta (right, arrowheads). Scale bars, 20 μm and 2 μm (for expanded view). (H, I) Quantification of colocalized puncta (inhibitory synapses) per 100 μm$^2$ (H) and their area covered (I). Data information: Data indicate mean ± SEM. Numbers on bars show tested cells and (number of animals). *P* values are from unpaired Student's *t* (B, C, E, F, I) and Mann–Whitney (H) tests.

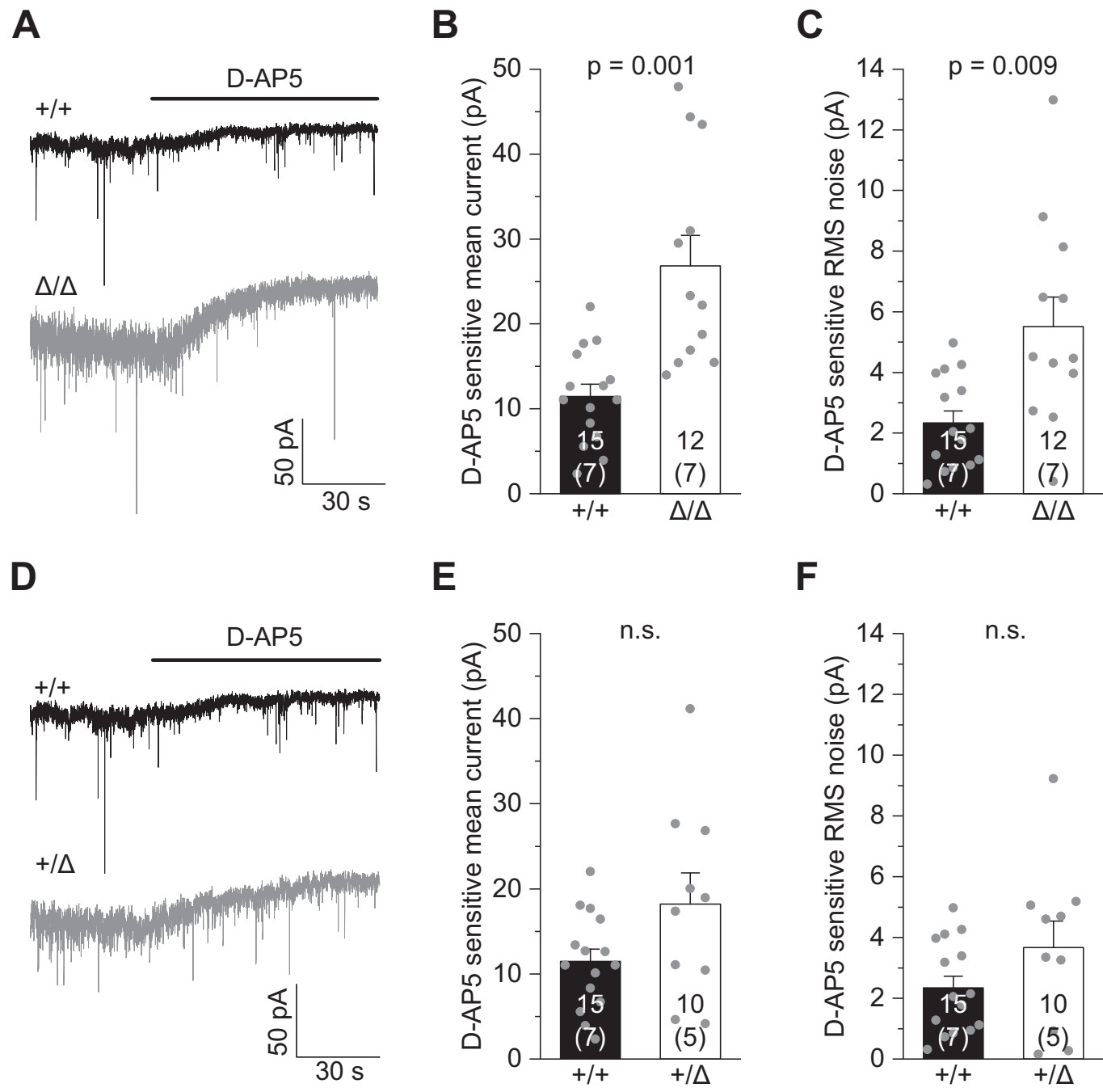

**Figure EV4. Increased tonic NMDA current in *Csf1r*^ΔFIRE/ΔFIRE mice.**

(A) Example traces showing changes in holding current after application of 50 µM D-AP5 for Δ/Δ compared with +/+ mice, reflecting blockade of all tonic NMDAR-mediated currents in CA1 pyramidal cells. Measurements were done in nominally $Mg^{2+}$-free extracellular solution in the presence of 300 nM TTX and 10 µM glycine. (B, C) Comparison of the D-AP5-sensitive mean holding current (B) and root mean square (RMS) noise (C) for the different genotypes. Note the contribution of both synaptic and extrasynaptic NMDA receptors to these parameters. (D–F) Same as for (A–C), but comparison of wild-type (+/+) and heterozygous phenotype (+/Δ). Data information: Data are represented as mean ± SEM. Numbers on bars indicate tested cells and (number of animals). *P* values are from unpaired Student's *t* tests.

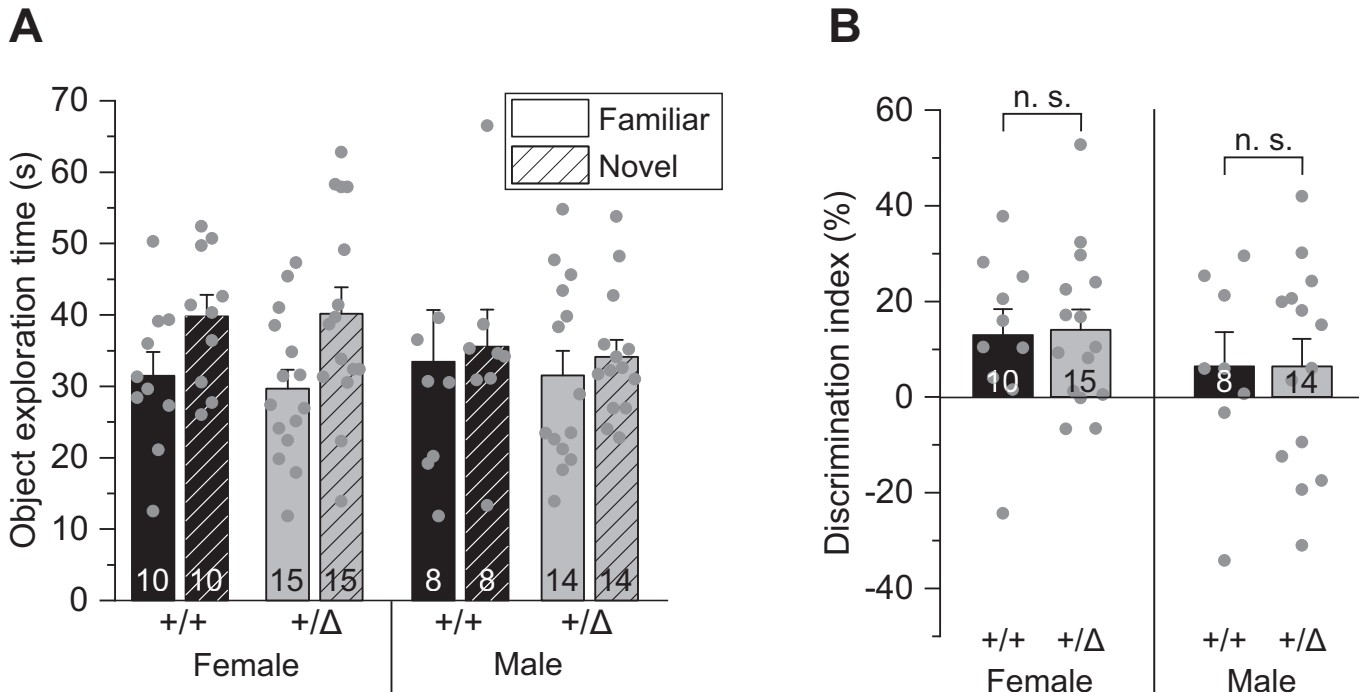

**Figure EV5. Unaltered object recognition memory in heterozygous *Csf1r*$^{+/\Delta FIRE}$ mice.**

(A) Comparison of exploration times of the familiar and novel object after habituation for male and female +/Δ and +/+ mice. (B) Comparison of the respective discrimination indices (see "Methods"). Data information: Data are represented as mean ± SEM. Numbers on bars indicate number of tested animals. *P* values are from two-way ANOVA (B).

