## [Peer Review File · EMBO Reports]

Lifelong absence of microglia alters hippocampal glutamatergic networks but not synapse and spine density

Michael Surala, Luna Šošo Zdravkovic, David Munro, Ali Rifat, Koliane Ouk, Imre Vida, Josef Priller, and Christian Madry

Corresponding author(s): Christian Madry (christian.madry@charite.de)

Review Timeline:

Transfer Date:	27th Feb 24
Editorial Decision:	11th Mar 24
Revision Received:	13th Mar 24
Accepted:	20th Mar 24

Editor: Esther Schnapp

Transaction Report: A revised version of this manuscript was transferred to EMBO reports following peer review at the EMBO Journal.

Date: 15th Sep 23 02:28:47

Last Sent: 15th Sep 23 02:28:47

Triggered By: Karin Dumstrei

From: k.dumstrei@embojournal.org

To: christian.madry@charite.de

BCC: h.sonntag@source-data.org

Subject: Manuscript EMBOJ-2023-114813 - Decision

Message: Dear Christian,

Thank you for sending me the point-by-point response and your proposal for how to address the raised concerns. I have now had a chance to look at it and I appreciate the suggested experiments. I would therefore like to invite you to submit a revised manuscript for our consideration.

Thank you for the opportunity to consider your work for publication. I look forward to your revision.

with best wishes

Karin

Karin Dumstrei, PhD
Senior Editor
The EMBO Journal

See also guidelines for figure legends:

<https://www.embopress.org/page/journal/14602075/authorguide#figureformat>

Further information is available in our Guide For Authors:

We realize that it is difficult to revise to a specific deadline. In the interest of protecting the conceptual advance provided by the work, we recommend a revision within 3 months (14th Dec 2023). As a matter of policy, competing manuscripts published during this period will not negatively impact on our assessment of the conceptual advance presented by your study. Please discuss the revision progress ahead of this time with the editor if you require more time to complete the revisions.

Use the link below to submit your revision:

Link Unavailable

Referee #1:

In their manuscript, Surala et al. study the impact of permanent microglial deficiency on the developmental wiring of hippocampal neuronal networks. The authors take advantage of a recently developed mouse model with the germ-line deletion of the *fms*-intronic regulatory element (FIRE) in the *Csf1r* locus, which is required for the expression of the CSF1 receptor in bone marrow progenitors and blood monocytes and leads to a complete absence of microglia throughout life. They find that the depletion of microglia does not cause a gross change in either synapse density or dendritic arborization. Instead, the absence of microglia weakened glutamatergic transmission at the CA3 to CA1 synapse, accompanied by impairment of multivesicular release and a reduction in the postsynaptic NMDA receptor component, without changes in inhibitory GABAergic transmission. Besides, the authors observed no impairment in the novel object recognition test, a mild increase in the level of anti-inflammatory cytokines, and a strong increase in ApoE levels as well as increased hippocampal GFAP immunoreactivity. These results are interesting and challenge the current view that microglia are key for proper CNS development due to their pivotal role in synaptic pruning.

The manuscript is clearly structured and well-written, however, several points should be addressed before the manuscript can be recommended for publication in the EMBO Journal.

1) Throughout the manuscript, the numbers of cells and mice are low e.g. in the experiments shown in Figures 1D-I, 3E, 5, etc. only 1-2 cells per mouse were studied. This is problematic given the rather high variability of the results. Moreover, at this level of data variability such low numbers might cause the misinterpretation of the data (e.g. no effect on CA1 cell morphology or synaptic density (Fig. 1J-L, obtained from a very small cohort of 3 mice/group)).

2) "...these results demonstrate that the absence of microglia in *Csf1r* Δ FIRE/ Δ FIRE mice leads to a reduced ability of CA1 pyramidal cells to generate action potentials, thus limiting glutamatergic transmission downstream. These differences appear to be associated with microglial CSF1R signaling, as suggested by the qualitatively similar findings in heterozygous *Csf1r*⁺/ Δ FIRE mice with normal microglial density".

The authors have to at least discuss and better test how microglial CSF1R signaling leads to a "lower excitatory input into CA1 pyramidal cells from the hippocampal CA3 region" and a "reduced ability of CA1 pyramidal cells to generate action potentials".

3) "...The observed reduction in the postsynaptic NMDAR component may be related to an increase in ambient glutamate levels, as suggested by the increase in tonic D-AP5-sensitive inward current in *Csf1r* Δ FIRE/ Δ FIRE mice. In consequence, NMDAR desensitization would be increased, leaving fewer NMDARs available for synaptic activation...".

This reviewer has difficulty following this explanation, as no desensitization of AMPA receptors, known to desensitize strongly, was observed in the very same experiments.

4) "...these data indicate a trend toward an anti-inflammatory state in brains of *Csf1r* Δ FIRE/ Δ FIRE mice. This is accompanied by reactive astrocyte morphology...".

This is also a contradiction, isn't it, as the reactive astrocyte morphology is usually observed in the pro-inflammatory state.

5) The authors have to provide a more precise discussion of their data in the context of previous findings leading to the "microglia-dependent synapse elimination hypothesis". Where the discrepancies might originate from? Which pro-/contra-arguments exist? How to test which hypothesis is right?

6) The authors state that "... The absence of microglia during embryonic development and postnatal life resulted in a distinct electrophysiological phenotype in *Csf1r* Δ FIRE/ Δ FIRE mice, with weakened and immature glutamatergic hippocampal transmission compared to WT littermates." However, this phenotype was partially also seen in *Csf1r*+/ Δ FIRE mice (e.g. Fig. EV2), which have normal microglial density, making it unlikely that the absence of microglia is the causative factor for synapse immaturity. Therefore, the authors should make an effort to figure out which deficits can be attributed to the absence of microglia and which ones to a deficit in CSFR signaling.

7) The potential capability of astrocytes "taking over" the role of microglia during synaptic pruning is very interesting but highly speculative based on data shown in the manuscript. It would be highly informative to see whether astrocytes are indeed capable of taking up synaptic material during the pruning phase in this particular mouse model and the authors have all techniques in place to address this question.

8) It is unclear to this reviewer how the "ramification index" shown in Fig. EV1 B was determined. The authors should add an explanation to the methods section.

9) This reviewer finds it extremely difficult to read a paper without page numbers and strongly recommends the authors to use them in the future. Moreover, the absence of page numbers strongly impeded and slowed down the review process, as it was extremely difficult to refer to a given test passage.

Referee #2:

In this study, Surala and colleagues show that microglial cells have a fundamental role in correctly shaping the hippocampal network, but they are not pivotal for pruning of synapses during development. To state this, the Authors took advantage of a mouse model that lacks microglial cells throughout the entire life, thus avoiding confounding effect due to microglial deprivation through pharmacological treatments. Even though the electrophysiological data investigating excitatory and inhibitory hippocampal

circuitry are convincing, the conclusion that microglia are dispensable for pruning of synapses during development is not supported by data. As also shown by the Authors, the brain environment is heavily affected in Δ FIRE / Δ FIRE mice, and this could contribute to modulate synapse homeostasis. For example, if Δ FIRE / Δ FIRE mice are defective in pruning but also in synaptogenesis, a balance leading to an equivalent synapse number is expected, and this doesn't mean that microglia pruning is dispensable. Also, the neuronal density could be altered in these mice, leading to a similar scenario.

More experiments addressing maturation/proliferation of neurons and synapse density at different developmental stages (before the starting and after the end of the pruning process) would be needed before drawing this conclusion. Also, astrocytic phenotype and functions and immune cells infiltrating in brain parenchyma should be addressed. Finally, authors should be more consistent with the number and sex identity of animals analyzed in each experiment and condition.

Specific comments:

1) Fig.1 C/D/E: The representative figures do not seem to correctly represent the data: there is a clear difference in spine number between WT and Δ FIRE / Δ FIRE mice, with Δ FIRE / Δ FIRE mice showing a higher density of spine. In Fig. 1E, high data variability in Δ FIRE / Δ FIRE animals is detectable. More neurons need to be analysed. Moreover, authors need to specify whether and how data were normalized in fig. 1 D/E.

2) Fig.1 J: The authors should stain hippocampal slices for inhibitory synaptic markers (GPHN/vGAT) to evaluate whether alterations in synapse density are present.

3) Fig.1 K/L: Due to the high variability of data, authors should increase the number of analysed animals.

4) Figure EV1 D/E/F/G/H/I/J: the authors compare very different numbers of animals and cells in different experiments. Higher consistency is required (see also below).

5) Heterozygous $Csf1r+/\Delta$ FIRE mice show the same amount of microglia as WT mice, but display reduced excitability of CA1 pyramidal cells, phenotype which is closer to the effect detected in $Csf1r \Delta$ FIRE / Δ FIRE. Therefore, $Csf1r+/\Delta$ FIRE may help dissecting the contribution of a defective CSF1R signaling (lack of FIRE) in the presence of a normal amount of microglia. How is the density and morphology of CA1 apical dendritic spines and excitatory synapses in heterozygous $Csf1r+/\Delta$ FIRE?

6) Fig.4 B/C/E: The authors tested only three Δ FIRE / Δ FIRE animals (half of WT), but the number of analyzed cells is similar. This leads to oversampling of the animals and might hide differences that could be evident by using the same number of mice. Along this line, it can be noted that in all figures illustrating the excitatory synapse density, by either microscopy (Fig 1C, Fig. 1E, Fig. 1L) or electrophysiology (4B, 4C), a high variability and generally higher average values are detectable in Δ FIRE / Δ FIRE mice.

7) Is synaptic multiplicity altered in *Csf1r*^{+/ΔFIRE} mice?

8) Fig. 5 C/D/E: While the number of animals analyzed is convincing, the number of cells sampled is too low. Authors should record at least 3 cells per mouse.

9) Fig. 6B: the fact that in *Csf1r*^{ΔFIRE/ΔFIRE} mice the brain environment seems to convert to a more anti-inflammatory one, raise the possibility that the process of synaptogenesis may be impacted in these mice. This is not even considered by the Authors.

10) Fig.6C: The authors state that "Apolipoprotein E (ApoE) is mainly produced by astrocytes in the murine brain (Boyles et al., 1985; Pitas et al., 1987; Zhang et al., 2014 and 2016). Interestingly, soluble and membrane bound ApoE levels were strongly increased in the brains of *Csf1r*^{ΔFIRE/ΔFIRE} mice compared to *Csf1r*^{+/ΔFIRE} and WT littermates (Fig 6C and Table 1), suggesting that changes in ApoE result from the absence of microglia rather than alterations in CSF1R function". I disagree with the authors regarding this statement. As they mentioned, ApoE is mainly produced by astrocytes in the brain. In fig. 7, authors show an increase in astrocyte density and "activation" profile in FIRE animals. Thus, the increase in ApoE is likely related to the increase in number and activation of astrocyte, rather than "alterations in CSF1R function". *Csf1r*^{+/ΔFIRE} could help in addressing more clearly this issue.

11) Fig.6A: Why did authors analyze a whole hemisphere? Focusing on the hippocampus, as in the rest of the paper, would have been more accurate in describing the phenotypes observed in previous figures.

12) Fig.6B: Authors sampled different cytokines by multiplex ELISA. However, they did not analyze what could be the origin of this dysregulation. The lack of microglial cells might have a role in Blood-Brain Barrier (BBB) formation (Ronaldson & Davis 2020), thus causing differential invasion of immune cells in the brain parenchyma. This needs to be addressed or at least discussed. Also, the macrophage colony-stimulating factor (CSF1) receptor (CSF1R) has two ligands, CSF1 and IL34. Did the authors test the levels of CSF1 and IL34 in the hippocampus and in the whole brain of *Csf1r*^{+/ΔFIRE} and *Csf1r*^{ΔFIRE /ΔFIRE} mice?

13) Fig.6C: What is the functional meaning of ApoE increase in *ΔFIRE /ΔFIRE* animals? Authors should at least discuss it in text.

14) Fig.7: The Authors show an increase in astrocyte density and ramification, which can be related to their activation. Since the role of astrocyte in eliminating adult hippocampal excitatory synapses is recognized (Lee et al., 2020), the Authors should test whether astrocytic synapse elimination is enhanced in the *ΔFIRE /ΔFIRE* model. Should this not occur and given that a similar synapse density is apparently present in

WT and Δ FIRE / Δ FIRE mice, the possibility that the process of synaptogenesis is impaired in Δ FIRE / Δ FIRE mice would become even more likely, seriously challenging the Authors' conclusions.

15) Fig.8: Before concluding that memory is not impaired in FIRE animals, additional behavioural tests should be performed (Morris water maze, T maze). Also, a different way of showing the results of the test should be adopted, in order to facilitate the readers in appreciating that WT mice are able to discriminate the novel object.

16) Authors should state clearly for every experiment whether they are analyzing the same number of male and female mice in CTRL and FIRE animals, since they are evaluating phenotypes that might be related to sex.

17) Authors should specify in all figures to what the number in the bars they refers to.

Referee #3:

In this remarkably well-written manuscript, Surala and colleagues explore the functional consequences of developmental absence of microglia in the FIRE CSF1R mouse. This mouse will likely find widespread application in teasing out microglia-specific function in neurodevelopment and disease and thus careful analyses are of great interest amongst neuroscientists and neuroimmunologists. The authors demonstrate that, while not of consequence in one non-spatial learning paradigm, congenital absence of microglia due to CSF1R fire element deletion influences functional but not structural synapse formation and leads to decreased excitatory synapse function in hippocampal circuits. Astrocytes show increased GFAP reactivity as a potential clue into their function. The studies are technically excellent and the analyses complete. The potential limitation of this work for the audience is that it characterizes one rodent model of congenital absence of microglia (as very few if any equivalent models exist) and the mechanism remains uncertain. To further strengthen the impact of this work, the authors could determine whether known astrocyte synaptogenic proteins are differentially regulated in microglia absence (especially those which mediate functional but not necessarily structural synapse formation). That said, this work is of broad interest and is likely to be highly regarded as there is increasing excitement in studying microglial function using new genetic systems.

Referee #1:

In their manuscript, Surala et al. study the impact of permanent microglial deficiency on the developmental wiring of hippocampal neuronal networks. The authors take advantage of a recently developed mouse model with the germ-line deletion of the *fms*-intronic regulatory element (FIRE) in the *Csf1r* locus, which is required for the expression of the CSF1 receptor in bone marrow progenitors and blood monocytes and leads to a complete absence of microglia throughout life. They find that the depletion of microglia does not cause a gross change in either synapse density or dendritic arborization. Instead, the absence of microglia weakened glutamatergic transmission at the CA3 to CA1 synapse, accompanied by impairment of multivesicular release and a reduction in the postsynaptic NMDA receptor component, without changes in inhibitory GABAergic transmission. Besides, the authors observed no impairment in the novel object recognition test, a mild increase in the level of anti-inflammatory cytokines, and a strong increase in ApoE levels as well as increased hippocampal GFAP immunoreactivity. These results are interesting and challenge the current view that microglia are key for proper CNS development due to their pivotal role in synaptic pruning.

The manuscript is clearly structured and well-written, however, several points should be addressed before the manuscript can be recommended for publication in the EMBO Journal.

We thank the referee for their positive judgement of our manuscript.

1) Throughout the manuscript, the numbers of cells and mice are low e.g. in the experiments shown in Figures 1D-I, 3E, 5, etc. only 1-2 cells per mouse were studied. This is problematic given the rather high variability of the results. Moreover, at this level of data variability such low numbers might cause the misinterpretation of the data (e.g. no effect on CA1 cell morphology or synaptic density (Fig. 1J-L, obtained from a very small cohort of 3 mice/group)).

We acknowledge the referee's criticism and have significantly increased the number of cells and animals for all morphological and electrophysiological experiments. For clarity, the attached table (see below) lists all previous and newly added experiments for each subfigure with details of animal number, genotype and sex. Specifically, we nearly doubled the number of cells and mice for Fig. 1B-I (from n=22/9 cells/mice to n=35/14), Fig. 1J, L (from 28/6 to 55/13) and Fig 5 (from n=16/10 to n=24/14). Importantly, this also improved the sex ratio of the animals while increasing statistical robustness. In total, we included 164 additional data points (cells/slices) from 48 additional mice (WT, *Csf1r*^{+/ \$\Delta\$ FIRE}, *Csf1r* ^{\$\Delta\$ FIRE/ \$\Delta\$ FIRE}) in the analyses of the revised manuscript (Figs. 1-5 and Figs. EV 1-4).

2) "...these results demonstrate that the absence of microglia in *Csf1r* Δ FIRE/ Δ FIRE mice leads to a reduced ability of CA1 pyramidal cells to generate action potentials, thus limiting glutamatergic transmission downstream. These differences appear to be associated with microglial CSF1R signaling, as suggested by the qualitatively similar findings in heterozygous *Csf1r*^{+/ Δ FIRE} mice with normal microglial density".

The authors have to at least discuss and better test how microglial CSF1R signaling leads to a "lower excitatory input into CA1 pyramidal cells from the hippocampal CA3 region" and a "reduced ability of CA1 pyramidal cells to generate action potentials".

By adding further experiments, the differences in CA1 excitability between $Csf1r^{+/+}$ (WT) and $Csf1r^{\Delta FIRE/\Delta FIRE}$ mice have now become smaller (but remained significantly different). This was mainly due to slightly lower values for the WT, possibly related to the now nearly equalised sex balance (while the data initially was from females, it now comes from 4 females and 3 males). As a result, the difference in neuronal excitability between WT and $Csf1r^{+/+/\Delta FIRE}$ mice was no longer significant (cf. Fig. EV1), which we have adapted in the text as follows (changes are highlighted in bold):

- However, CA1 pyramidal cells produced a **slightly** lower rate of action potentials on injection of depolarizing current in $Csf1r^{\Delta FIRE/\Delta FIRE}$ mice compared to WT littermates... (page 7, line 175)
- In contrast, in heterozygous $Csf1r^{+/\Delta FIRE}$ mice, in which microglia were present at normal densities but were morphologically more ramified compared to WT littermates (Figs EV1A-C), **no changes were seen for excitability of CA1 pyramidal cells, input resistance, cell capacitance, and resting potential (Figs EV1D-J)**. (page 7, lines 185-186)
- Therefore, we also omitted the last sentence of the 3rd paragraph on page 7 (“*These differences appear...*”).

However, amplitudes of CA1 EPSCs upon Schaffer collateral stimulation remained significantly lower in $Csf1r^{\Delta FIRE/\Delta FIRE}$ compared to WT mice (Fig. 3C). A similar decrease in CA3-CA1 transmission was also seen in adult mice after acute PLX-induced microglia depletion (Basilico et al., 2022, PMID 34661306; Table 2 of our manuscript), implying that the effect is due to microglia supporting glutamatergic transmission in the adult, but not in the developing brain. (considered in the Discussion on page 16, lines 461-462)

3) "...The observed reduction in the postsynaptic NMDAR component may be related to an increase in ambient glutamate levels, as suggested by the increase in tonic D-AP5-sensitive inward current in $Csf1r^{\Delta FIRE/\Delta FIRE}$ mice. In consequence, NMDAR desensitization would be increased, leaving fewer NMDARs available for synaptic activation..."

This reviewer has difficulty following this explanation, as no desensitization of AMPA receptors, known to desensitize strongly, was observed in the very same experiments.

This is a fair point raised, and the referee is right in pointing out that no desensitisation of AMPAR-mediated currents was observed in $Csf1r^{\Delta FIRE/\Delta FIRE}$ mice. While the extent of desensitisation is far greater for AMPARs than NMDARs, the concentration dependence of desensitisation is the more relevant parameter in this context. Our assumption that NMDARs may be desensitised due to a higher ambient glutamate concentration would be feasible up to a glutamate concentration at which AMPARs do not yet desensitise significantly ($\leq 2 \mu\text{M}$; Colquhoun et al., 1992, PMID 1338788). Because of the ~ 4-fold higher affinity for glutamate-dependent desensitisation of NMDARs (Zorumski et al., 1996, PMID 8842005; Cavalier et al., 2005, PMID 15471587), an increase in ambient glutamate levels within this range would lead to an increase in desensitised NMDARs unavailable for synaptic activation. To make this point clearer and to emphasise the implications of increased tonic NMDAR currents in $Csf1r^{\Delta FIRE/\Delta FIRE}$ mice, we have modified our statement on page 16, lines 469 ff, as follows:

- The observed reduction in the postsynaptic NMDAR component may be related to an increase in ambient glutamate levels, **which is mainly regulated by astrocytes (Le**

Meur et al., 2007) and could reflect the altered astrocytic properties in $Csf1r^{\Delta FIRE/\Delta FIRE}$ mice. In consequence, NMDAR desensitization would be increased, leaving fewer NMDARs available for synaptic activation (Cavelier et al., 2005). Since no desensitisation of AMPARs was observed in $Csf1r^{\Delta FIRE/\Delta FIRE}$ mice, this would apply to glutamate levels below the AMPAR desensitisation threshold (~ 2 μM ; Colquhoun et al., 1992). Apart from the synapse, one third of NMDARs in pyramidal cells are also extrasynaptically localised and contribute significantly to tonic NMDAR-mediated currents (Harris and Pettit, 2007; Le Meur et al., 2007).

4) "...these data indicate a trend toward an anti-inflammatory state in brains of $Csf1r^{\Delta FIRE/\Delta FIRE}$ mice. This is accompanied by reactive astrocyte morphology..."

This is also a contradiction, isn't it, as the reactive astrocyte morphology is usually observed in the pro-inflammatory state.

Indeed, this point may appear contradictory at first glance. However, recent findings indicate two functionally opposite classes of GFAP+ reactive astrocytes, i.e. proinflammatory neurotoxic A1 and antiinflammatory neuroprotective A2 types (Liddlelow et al., 2017, PMID 28099414). The existence of a proinflammatory state and presence of A1-type astrocytes is not compatible with the cytokine profile we obtained in brains of $Csf1r^{\Delta FIRE/\Delta FIRE}$ mice (Fig. 6). Moreover, A1 astrocytes, which require activation by microglia, have a strongly deramified shape, whereas in our case the increase in GFAP intensity was accompanied by a more complex morphology (Fig. 7). Similar to our findings, an increase in astrocytic GFAP intensity without emergence of neuroinflammation was observed upon acute depletion of microglia in adult mice by CSF1R blockade (Basilico et al., 2022, PMID 34661306).

5) The authors have to provide a more precise discussion of their data in the context of previous findings leading to the "microglia-dependent synapse elimination hypothesis". Where the discrepancies might originate from? Which pro-/contra- arguments exist? How to test which hypothesis is right?

Our findings indicate the clear absence of a pruning deficit in the hippocampus of $Csf1r^{\Delta FIRE/\Delta FIRE}$ mice, a brain region extensively studied in this context (e.g. Paolicelli et al., 2011, PMID 21778362; Zhan et al., 2014, PMID 24487234; Filipello et al., 2018, PMID 29752066; Basilico et al., 2018, PMID 30417584; Jay et al., 2019, PMID 31265185; Konishi et al., 2020, PMID 32959911). In support of this, the morphological and functional changes we observed in $Csf1r^{\Delta FIRE/\Delta FIRE}$ mice closely resembled those resulting from acute microglial depletion in adult animals that have undergone normal development (cf. Table 2). A major difference between our work and previous studies is the complete absence of microglia throughout life in $Csf1r^{\Delta FIRE/\Delta FIRE}$ mice, whereas in studies examining mechanisms of microglial pruning they were present, albeit functionally impaired. These works have undisputably identified many important mechanisms by which microglia may interact with and engulf synaptic structures, including CX3CR1 (which plays a central role in the hippocampus in particular), TREM2, CR3 and, more recently, GABA_B receptors (Favuzzi et al., 2021, PMID 34233165).

Nevertheless, it is still unclear what precise roles microglia, astroglia and OPCs play in the elimination of excess synapses (Chung et al., 2013, PMID 24270812; Faust et al., 2021, PMID 34545240; Buchanan et al., 2022, PMID 36417438; Mordelt & de Witte 2023, PMID

36657237). A study by Weinhard et al., 2018 (PMID 29581545) found no evidence for phagocytic uptake of entire synapses by microglia in the developing hippocampus using high-resolution ultrastructural analyses. Instead, microglia interacted with presynaptic structures, entirely ignoring postsynaptic spines. Until now, direct phagocytosis of synapses by microglia has not been reported (Eyo & Molofsky, 2023, PMID 37708287). The mere presence of postsynaptic material inside microglia would not necessarily indicate specific synaptic pruning, as this could also result from the uptake of neuronally released vesicles containing postsynaptic material (Eyo & Molofsky, 2023; Mordelt & de Witte, 2023).

Apart from the general question whether or to what extent microglia are required for synaptic pruning, it is important to consider that the mechanisms of microglia-synapse interactions critically depend on age, brain region and context. This is consistent with the marked genetic heterogeneity of microglia throughout development as revealed by recent transcriptomic studies (Li et al., 2019, PMID 30606613; Masuda et al., 2020, PMID 32023447), which suggest specialization of microglial functions. For example, CX3CR1 is critical for synaptic engulfment in the hippocampus but is not required for the pruning of climbing fibers in the cerebellum (Paolicelli et al., 2011, PMID 21778362; Kaiser et al., 2020, PMID 32488990), whereas complement-dependent mechanisms via CR3 regulate synaptic elimination in the developing retinogeniculate system but not in somatosensory cortex (Schafer et al., 2012, PMID 22632727; Gunner et al., 2019, PMID 31209379).

Given our findings and the previous literature, how to resolve these apparent discrepancies about the role of microglia in neurodevelopment? Earlier studies already indicated that a substantial reduction in microglia number (in mice lacking expression of TGF- β or CSF1R agonist IL-34) did not result in obvious developmental CNS deficits under non-disease conditions (Wang et al., 2012, PMID 32029629; Butovsky et al., 2014, PMID 24316888), supporting our and recent findings in *Csf1r^{AFIRE/AFIRE}* mice (Rojo et al., 2019, PMID 31324781). Given the growing evidence that microglia, OPCs and astrocytes jointly orchestrate synaptic pruning (Chung et al., 2013, PMID 24270812; Bialas et al., 2013, PMID 24162655; Vainchtein et al., 2018, PMID 29420261; Lee et al., 2020, PMID 33361813; Damisah et al., 2020, PMID 32637606; Buchanan et al., 2022 PMID 36417438), it is plausible that astrocytes are able to (at least partially) compensate for the loss of microglia in *Csf1r^{AFIRE/AFIRE}* mice. Indeed, our newly generated data demonstrate a > 3-fold upregulation of the phagocytic receptor MEGF10 and 30% increase in synaptic markers VGlut1 and Homer1 in hippocampal astrocytes of *Csf1r^{AFIRE/AFIRE}* mice (see below).

Importantly, due to the division of labour between microglia and astrocytes, different consequences may arise if (i) microglia are present but pruning-relevant mechanisms are impaired, or (ii) if they are entirely absent. In the first case, astrocytes would continue to do "their" part, receiving signals from microglia with which they also coordinate their phagocytic capacity such as TGF- β , IL-33 or via TREM2-mediated signaling (Bialas et al., 2013; Vainchtein et al., 2018; Jay et al., 2019). In the second case, reflecting *Csf1r^{AFIRE/AFIRE}* mice, astrocytes no longer receive a "confirmatory" signal indicating the presence of microglia and were then shown to take over the phagocytic part of microglia, albeit with some delay (Damisah et al., 2020). Consequently, in the complete absence of microglia, changes may manifest differently than if these are present but functionally compromised. Although technically challenging, high-resolution longitudinal live imaging of microglia-astrocyte-synaptic interactions, e.g. in the developing hippocampus, would help to improve our understanding of the modalities of how glial cells are engaged in synaptic pruning.

Following the referee's advice, we have substantially extended our discussion, taking these considerations into account. For reasons of space, we also refer to excellent recent reviews in which the role of microglial pruning for neurodevelopment is discussed in greater detail (Faust et al., 2021) and the current concept of microglial pruning is challenged by proposing an updated view (Mordelt and de Witte, 2023; Eyo & Molofsky, 2023).

The corresponding sections of the discussion (page 14, 2nd paragraph ff) have been amended as follows (changes highlighted in bold):

“Dispensability of microglia to eliminate the surplus of synapses of developing hippocampal neurons” (page 14, 2nd paragraph ff)

- *None of the previous studies providing evidence for microglial-dependent synapse elimination during postnatal development in hippocampus, thalamus and visual cortex (Paolicelli et al., 2011; Schafer et al., 2012; Zhan et al., 2014; Sipe et al., 2016; Filipello et al., 2018; Liu et al. 2021) investigated the consequences of permanent absence of microglia during embryonic development and postnatal life on neuronal development and function. **Although there is a wealth of evidence suggesting that microglia engulf synaptic structures (Faust et al., 2021), with CX3CR1 playing a major role in the hippocampus (Paolicelli et al., 2011; Zhan et al., 2014; Basilico et al., 2018), it is still unclear to what extent microglia or macroglia contribute to the elimination of excess synapses.** Our findings in $Csf1r^{\Delta FIRE/\Delta FIRE}$ mice, which are free of potential confounding factors such as cell ablation, suggest that microglia are dispensable for the regulation of the number of synapses post-developmentally, **consistent with recent findings showing that microglia in the developing hippocampus neither phagocytose postsynaptic material nor entire synapses (Weinhard et al., 2018). So far, no direct evidence, e.g. by real-time imaging, for the phagocytosis of excessive synapses by microglia during development has been provided (Eyo & Molofsky, 2023). Supporting our findings, the marked reduction of microglial numbers in mice lacking expression of TGF- β or IL-34 did not result in obvious CNS developmental deficits under non-disease conditions (Wang et al., 2012; Butovsky et al., 2013). However, it is important to consider that the mechanisms of microglia-synapse interactions critically depend on age, context and brain region (Faust et al., 2021; Mordelt & de Witte 2023), in line with a pronounced genetic heterogeneity of microglia in the developing brain (Li et al., 2019).***

“Role of microglia-independent mechanisms in synaptic pruning” (page 15, 2nd paragraph ff)

- *Which alternative cell types or mechanisms may regulate the pruning of synapses in brains devoid of microglia? **Several types of macroglia, including astrocytes and oligodendrocyte precursor cells (OPCs), were identified as critical regulators of the pruning of synapses in the developing and adult brain (Chung et al., 2013; Buchanan et al., 2022). Notably, OPCs contain significantly more phagolysosomes filled with synaptic material than microglia in the developing mouse visual cortex (Buchanan et al., 2022). Astrocytes are more abundant than microglia and participate in activity-dependent synapse elimination and neural circuit formation in the developing and adult CNS by phagocytosing synapses via the MEGF10 and MERTK pathways (Chung et al., 2013; Lee et al., 2021). We find reactive astrocytes***

*with increased expression of GFAP and MEGF10 in $Csf1r^{\Delta FIRE/\Delta FIRE}$ mice at postnatal developmental stages when synaptic engulfment by microglia would have been highest (Jawaid et al., 2018). We also find elevated levels of ApoE and IL-10 in brains of $Csf1r^{\Delta FIRE/\Delta FIRE}$ mice. While IL-10 plays a role in synapse formation (Lim et al., 2013), ApoE controls the rate of synaptic pruning by astrocytes (Chung et al., 2016) and can affect glutamatergic transmission in multiple ways via signaling through synaptic ApoE receptors (Lane-Donovan & Herz, 2017). It is tempting to speculate that astrocytes may at least in part compensate for the loss of microglia in $Csf1r^{\Delta FIRE/\Delta FIRE}$ mice, consistent with reports showing an up-regulation of their phagocytic capacity in mice with dysfunctional or ablated microglia (Konishi et al., 2020; Berdowski et al., 2022). Future studies, **ideally by high-resolution longitudinal live imaging of entire synapses**, will help to determine what proportion of excess or unwanted synapses in development are subject to glial elimination as opposed to other processes, whether and how these processes differ between brain regions, and which specific functional properties of synapses determine their fate. As previously suggested, the simplified view that only weaker synaptic contacts are removed falls short of reflecting the high degree of structural and functional heterogeneity of synapses in neural networks (Wichmann and Kuner, 2022).*

6) The authors state that "... The absence of microglia during embryonic development and postnatal life resulted in a distinct electrophysiological phenotype in $Csf1r^{\Delta FIRE/\Delta FIRE}$ mice, with weakened and immature glutamatergic hippocampal transmission compared to WT littermates." However, this phenotype was partially also seen in $Csf1r^{+/\Delta FIRE}$ mice (e.g. Fig. EV2), which have normal microglial density, making it unlikely that the absence of microglia is the causative factor for synapse immaturity. Therefore, the authors should make an effort to figure out which deficits can be attributed to the absence of microglia and which ones to a deficit in CSFR signaling.

The question about the role of CSF1R signalling has already been touched on above (see point 2). By increasing the number of experiments (to achieve a more balanced sex ratio), CA1 pyramidal cell excitability no longer differed significantly between $Csf1r^{+/\Delta FIRE}$ and WT mice. To get a more complete picture of the functional changes in hetero- and homozygous mice, as suggested by the referee, we additionally determined the amplitude ratio for m- vs sEPSCs (as a measure of synaptic multiplicity) for $Csf1r^{+/\Delta FIRE}$ mice, and observed no difference between $Csf1r^{+/\Delta FIRE}$ and WT mice (see new Figs EV2 C and D). However, m- vs sEPSCs were significantly different between WT and $Csf1r^{\Delta FIRE/\Delta FIRE}$ mice (see updated Fig. 4F). In addition, we analysed the D-AP5-sensitive current and RMS noise also in heterozygous $Csf1r^{+/\Delta FIRE}$ mice (both parameters were significantly increased in $Csf1r^{\Delta FIRE/\Delta FIRE}$ vs. WT mice) but found no difference compared to WT (see new Figs EV4 D-F).

We have considered these additional findings on page 7 lines 175 and 183 ff, page 8 lines 223-224, and page 10 lines 285-286. As an altered EPSC amplitude between WT and heterozygous $Csf1r$ mice occurred only at very high stimulus intensities, we questioned the validity and removed these data from the manuscript (former Fig EV2A, B).

7) The potential capability of astrocytes "taking over" the role of microglia during synaptic pruning is very interesting but highly speculative based on data shown in the manuscript. It would be highly informative to see whether astrocytes are indeed capable of taking up

synaptic material during the pruning phase in this particular mouse model and the authors have all techniques in place to address this question.

We thank the referee for suggesting these additional experiments. The role of astrocytes in $Csf1r^{AFIRE/\Delta FIRE}$ mice is indeed of great importance to better understand the lack of a deficit in the number of dendritic spines and synapses in this model. As suggested, we have greatly increased the number of experiments and animals in the revised manuscript, corroborating our key finding of an unchanged number of spines and excitatory synapses in $Csf1r^{AFIRE/\Delta FIRE}$ vs WT mice post development. Interestingly, there was even a non-significant trend towards a slight decrease in the number of synapses in microglia-deficient mice, which is incompatible with a pruning deficit (Fig. 1K, L). To follow the referee's advice and shed more light on the role of astrocytes, we first examined GFAP expression at developmental stages P22-23, when the phagocytic capacity of microglia to engulf synaptic material has been reported to be highest in the hippocampal CA1 region (Jawaid et al., 2018, PMID 29274095). This showed an even greater increase in GFAP intensity and area compared to postdevelopmental stages, suggesting a change in astrocyte function especially in this critical phase of synapse remodelling in the developing hippocampus when microglia are otherwise active. To test whether this was associated with an increase in phagocytosis activity of astrocytes, we analysed expression of MEGF10 as a key astrocytic receptor regulating phagocytosis and synapse elimination (Chung et al., 2013, PMID 24270812). This revealed a strong increase (> 3-fold) in MEGF10 intensity in astrocytes of $Csf1r^{AFIRE/\Delta FIRE}$ mice at P22/23 compared to littermate controls. Consistent with this, we found significantly increased (~25-30%) amounts of pre- and postsynaptic markers (VGluT1, Homer1) in astrocytes of $Csf1r^{AFIRE/\Delta FIRE}$ mice.

We incorporated these new findings into the revised manuscript to the following sections of the Results and Discussion (please also note related amendments made in response to point 5):

- *Apolipoprotein E (ApoE) is mainly produced by astrocytes in the murine brain (Boyles et al., 1985; Pitas et al., 1987; Zhang et al., 2014 and 2016). Interestingly, soluble and membrane bound ApoE levels were strongly increased in the brains of $Csf1r^{AFIRE/\Delta FIRE}$ mice compared to $Csf1r^{+/AFIRE}$ and WT littermates (Fig 6C and Table 1), suggesting that changes in ApoE are due to altered astrocyte function in the absence of microglia. Morphological analysis of astrocytes in the CA1 stratum radiatum of 6-10-week-old $Csf1r^{AFIRE/\Delta FIRE}$ mice revealed an increase in both intensity and area covered by GFAP immunoreactivity, while astrocyte numbers were unchanged compared to WT littermates (Figs 7A-D). In line with this, we observed increased complexity of astrocyte morphology, as evidenced by Sholl analysis revealing increases in the number of intersections and total process length (Figs 7E-H). **Notably, the intensity and area covered by GFAP immunoreactivity was even more pronounced in $Csf1r^{AFIRE/\Delta FIRE}$ mice at P22-P23 (Fig. 7I-K), a time when engulfment of synaptic material by hippocampal microglia would normally have reached its peak (Jawaid et al., 2018). This was accompanied by marked increases in the expression of the key phagocytic receptor mediating synapse elimination in astrocytes, Multiple EGF-like domains (MEGF)10 (Fig. 7I, L, M; Chung et al., 2013), and enhanced incorporation of the synaptic markers VGluT1 and Homer1 in hippocampal astrocytes of $Csf1r^{AFIRE/\Delta FIRE}$ mice at P22-P23 compared to WT littermates (Fig. 7N-P).** Taken together, these data indicate a trend toward an anti-inflammatory state in brains of $Csf1r^{AFIRE/\Delta FIRE}$ mice **along with the presence of reactive astrocytes in the hippocampus with increased uptake of synaptic material. These changes are most prominent during hippocampal development, suggesting that astrocytes may***

contribute to pruning in the absence of microglia. (Results, page 11, 1st and 2nd paragraph)

- *Which alternative cell types or mechanisms may regulate the pruning of synapses in brains devoid of microglia? **Several types of macroglia, including astrocytes and oligodendrocyte precursor cells (OPCs), were identified as critical regulators of the pruning of synapses in the developing and adult brain (Chung et al., 2013; Buchanan et al., 2022). Notably, OPCs contain significantly more phagolysosomes filled with synaptic material than microglia in the developing mouse visual cortex (Buchanan et al., 2022). Astrocytes are more abundant than microglia and participate in activity-dependent synapse elimination and neural circuit formation in the developing and adult CNS by phagocytosing synapses via the MEGF10 and MERTK pathways (Chung et al., 2013; Lee et al., 2021). We find reactive astrocytes with increased expression of GFAP and MEGF10 in $Csf1r^{\Delta F1RE/\Delta F1RE}$ mice at postnatal developmental stages when synaptic engulfment by microglia would have been highest (Jawaid et al., 2018). We also find elevated levels of ApoE and IL-10 in brains of $Csf1r^{\Delta F1RE/\Delta F1RE}$ mice. While IL-10 plays a role in synapse formation (Lim et al., 2013), ApoE controls the rate of synaptic pruning by astrocytes (Chung et al., 2016) and can affect glutamatergic transmission in multiple ways via signaling through synaptic ApoE receptors (Lane-Donovan & Herz, 2017). It is tempting to speculate that astrocytes may at least in part compensate for the loss of microglia in $Csf1r^{\Delta F1RE/\Delta F1RE}$ mice, consistent with reports showing an up-regulation of their phagocytic capacity in mice with dysfunctional or ablated microglia (Konishi et al., 2020; Berdowski et al., 2022). Future studies, **ideally by high-resolution longitudinal live imaging of entire synapses**, will help to determine what proportion of excess or unwanted synapses in development are subject to glial elimination as opposed to other processes, whether and how these processes differ between brain regions, and which specific functional properties of synapses determine their fate. As previously suggested, the simplified view that only weaker synaptic contacts are removed falls short of reflecting the high degree of structural and functional heterogeneity of synapses in neural networks (Wichmann and Kuner, 2022).*** (Discussion, page 15, 2nd paragraph ff)
- *Collectively, our findings provide evidence that microglia are dispensable for synaptic pruning in the developing hippocampus and suggest that microglia-independent mechanisms, **which may involve macroglia**, contribute to a basic, albeit not fully mature, network connectivity without causing behavioral disturbances.* (end of Discussion, page 18, 2nd paragraph)

8) It is unclear to this reviewer how the "ramification index" shown in Fig. EV1 B was determined. The authors should add an explanation to the methods section.

We apologise for the lack of information and have added the description of the ramification index in the revised Methods section, page 24, line 726 ff.

9) This reviewer finds it extremely difficult to read a paper without page numbers and strongly recommends the authors to use them in the future. Moreover, the absence of page numbers strongly impeded and slowed down the review process, as it was extremely difficult to refer to a given text passage.

We apologise for this inconvenience, and we have now added line and page numbers in the revised manuscript.

Referee #2:

In this study, Surala and colleagues show that microglial cells have a fundamental role in correctly shaping the hippocampal network, but they are not pivotal for pruning of synapses during development. To state this, the Authors took advantage of a mouse model that lacks microglial cells throughout the entire life, thus avoiding confounding effect due to microglial deprivation through pharmacological treatments. Even though the electrophysiological data investigating excitatory and inhibitory hippocampal circuitry are convincing, the conclusion that microglia are dispensable for pruning of synapses during development is not supported by data. As also shown by the Authors, the brain environment is heavily affected in Δ FIRE / Δ FIRE mice, and this could contribute to modulate synapse homeostasis. For example, if Δ FIRE / Δ FIRE mice are defective in pruning but also in synaptogenesis, a balance leading to an equivalent synapse number is expected, and this doesn't mean that microglia pruning is dispensable. Also, the neuronal density could be altered in these mice, leading to a similar scenario.

More experiments addressing maturation/proliferation of neurons and synapse density at different developmental stages (before the starting and after the end of the pruning process) would be needed before drawing this conclusion. Also, astrocytic phenotype and functions and immune cells infiltrating in brain parenchyma should be addressed. Finally, authors should be more consistent with the number and sex identity of animals analyzed in each experiment and condition.

We thank the referee for their assessment of our work, which we have substantially revised considering their comments and suggestions. Our overarching goal was to investigate the influence of microglia on pruning, i.e. the adaptation of the number of synapses and their maturation after completion of brain development, using $Csf1^{\Delta$ FIRE/ Δ FIRE mice deficient of microglia throughout life.

Following the referee's advice, we have added a pre-pruning time point (P9-P10) and analyzed the number of excitatory synapses. As was the case at post-developmental stages, there were no differences in the number and size of colocalized synaptic puncta labeled for VGluT1 and Homer1 in WT in $Csf1^{\Delta$ FIRE/ Δ FIRE mice at this earlier stage. We have added these new findings to Figure 1M, N and considered them in the manuscript on page 6, line 164 ff, as follows (changes highlighted in bold):

- *Consistent with the spine data, the number and size of excitatory synapses, as defined by the colocalization of both markers, were unchanged in the CA1 region in $Csf1^{\Delta$ FIRE/ Δ FIRE mice **during development at postnatal days (P)9-P10 and in young adulthood compared with WT mice (Figs 1J-N).***

We have further investigated the role of astrocytes in $Csf1^{\Delta$ FIRE/ Δ FIRE mice in greater detail and included another time point within the developmental period at P22/P23 (see point 14 below). Overall, we have significantly increased the number of cells and animals for all morphological and electrophysiological experiments by including 164 additional experiments (data points) from 48 additional mice (WT, $Csf1^{+/\Delta$ FIRE}, $Csf1^{\Delta$ FIRE/ Δ FIRE) in the analyses of the revised manuscript (Figs. 1-5 and Figs. EV 1-4). For clarity, the attached table (see below) lists all previous and newly added experiments for each subfigure with details of animal number, genotype and sex.

Specific comments:

1) Fig.1 C/D/E: The representative figures do not seem to correctly represent the data: there is a clear difference in spine number between WT and Δ FIRE / Δ FIRE mice, with Δ FIRE / Δ FIRE mice showing a higher density of spine. In Fig. 1E, high data variability in Δ FIRE / Δ FIRE animals is detectable. More neurons need to be analysed. Moreover, authors need to specify whether and how data were normalized in fig. 1 D/E.

As suggested, we have increased the number of neurons examined for spine density (Fig. 1 C-E) and selected more representative specimen images for Fig. 1C. Specifically, we nearly doubled the number of cells and mice for Fig. 1B-I (from n=22/9 cells/mice to n=35/14). These further experiments have corroborated our main finding that spine density and number of excitatory synapses do not differ between $Csf1r^{\Delta$ FIRE/ Δ FIRE and WT mice.

Data for spine density and length (Fig. 1D, E) were not normalized. As stated in the Methods (page 24, 1st paragraph, line 706 ff), ***“Per individual neuron, five representative regions of interest comprising 15-20 μ m long segments of apical dendritic regions were analyzed. For each individual segment, a 3D skeleton was generated and analyzed for spine number and length using Fiji. Final values for spine density and length per individual cell reflect averages from all five segments.”***

2) Fig.1 J: The authors should stain hippocampal slices for inhibitory synaptic markers (GPHN/vGAT) to evaluate whether alterations in synapse density are present.

Following the referee’s suggestion, we analyzed inhibitory synapses by immunohistochemistry using VGAT and Gephyrin as pre- and postsynaptic markers. Consistent with our functional data on inhibitory synaptic currents, no differences in the number of inhibitory synapses were found in the hippocampus of young adult $Csf1r^{\Delta$ FIRE/ Δ FIRE compared with WT mice. These new data are now added to Figs EV3G-I, with amendments in the Results section (page 8, 3rd paragraph) made as follows:

- ***Absence of microglia does not change inhibitory synaptic transmission***
- ***No changes were observed in GABAergic synaptic transmission since inter-event intervals and amplitudes of either spontaneous inhibitory postsynaptic currents (sIPSCs) or miniature inhibitory postsynaptic currents (mIPSCs) were unaltered in CA1 neurons of $Csf1r^{\Delta$ FIRE/ Δ FIRE mice (Figs EV3A-F). The number and size of inhibitory synapses, as defined by the colocalization of presynaptic VGAT and postsynaptic Gephyrin immunoreactivities, were unchanged in the CA1 region in 6-10-week-old $Csf1r^{\Delta$ FIRE/ Δ FIRE mice compared with WT littermates (Figs EV3G-I).***

3) Fig.1 K/L: Due to the high variability of data, authors should increase the number of analysed animals.

The new data set for Fig. 1K, L now includes twice the number of animals and analyzed slices, from 28/6 to 55/13 (slices/animals), with the result being unchanged.

4) Figure EV1 D/E/F/G/H/I/J: the authors compare very different numbers of animals and cells in different experiments. Higher consistency is required (see also below).

We have significantly increased the number of animals and analyzed neurons for this set of experiments from 36/15 (cells/animals) to 51/19. This led to an abolishment of the initially observed minor change in firing frequency between WT and *Csf1r^{+ΔFIRE}* mice, and was mainly due to slightly lower values for firing frequencies of CA1 pyramidal cells in the WT. The latter is possibly due to the now balanced sex ratio (while the data initially was from females, it now comes from 4 females and 3 males).

5) Heterozygous *Csf1r^{+ΔFIRE}* mice show the same amount of microglia as WT mice, but display reduced excitability of CA1 pyramidal cells, phenotype which is closer to the effect detected in *Csf1r ΔFIRE /ΔFIRE*. Therefore, *Csf1r^{+ΔFIRE}* may help dissecting the contribution of a defective CSF1R signaling (lack of FIRE) in the presence of a normal amount of microglia. How is the density and morphology of CA1 apical dendritic spines and excitatory synapses in heterozygous *Csf1r^{+ΔFIRE}*?

As mentioned above, by adding further experiments the initially observed mild difference in excitability of CA1 pyramidal cells between WT and *Csf1r^{+ΔFIRE}* mice is no longer seen and the difference in CA1 excitability between *Csf1r^{+/+}* (WT) and *Csf1r^{ΔFIRE/ΔFIRE}* mice has now become smaller (but remained significantly different). In fact, this updated result further corroborates our main finding that the absence of microglia does not affect the number of synapses yet influences the maturation of glutamatergic transmission (but the change of which is even smaller in the course of this new data).

As suggested by the referee, we further analyzed spine density in heterozygous mice, but found no changes compared to the WT [spine density: WT, 2.54 ± 0.12 N°/μm (n = 21/7) vs *Csf1r^{+ΔFIRE}*, 2.88 ± 0.19 N°/μm (n = 10/5); p = 0.123 (Student's test) and spine length: WT, 1.26 ± 0.02 μm (n = 21/7) vs *Csf1r^{+ΔFIRE}*, 1.22 ± 0.02 μm (n = 10/5); p = 0.204 (Mann-Whitney test)]. Since no changes were seen, and we find that this data would rather distract from the main message of an unchanged number of spines and synapses in the absence of microglia, we have not included this finding in the manuscript. In general, *Csf1r* haploinsufficiency (*Csf1r^{+ΔFIRE}* mice) results in a very complex phenotype (see e.g. Chitu et al., 2015, PMID 25497733; Biundu et al., 2020, PMID 33079443) and should not be understood as a "half" knock-out, specifically because the number of microglia is unchanged in these mice.

6) Fig.4 B/C/E: The authors tested only three *ΔFIRE /ΔFIRE* animals (half of WT), but the number of analyzed cells is similar. This leads to oversampling of the animals and might hide differences that could be evident by using the same number of mice. Along this line, it can be noted that in all figures illustrating the excitatory synapse density, by either microscopy (Fig 1C, Fig. 1E, Fig. 1L) or electrophysiology (4B, 4C), a high variability and generally higher average values are detectable in *ΔFIRE /ΔFIRE* mice.

To improve statistical robustness, we greatly increased the number of experiments and animals as follows: Fig. 4B from n=31/9 (cells/mice) to n=47/14 and Fig. 4C, E from n=33/8 to 52/13. The additional experiments further corroborated our central finding of an unchanged number of synaptic markers in *Csf1r^{ΔFIRE/ΔFIRE}* and WT mice with similar variability of data between the two groups.

7) Is synaptic multiplicity altered in *Csf1r^{+ΔFIRE}* mice?

We analyzed synaptic multiplicity also in heterozygous $Csf1r^{+/\Delta FIRE}$ mice and found that it was not different from WT mice. In light of the altered synaptic multiplicity in $Csf1r^{\Delta FIRE/\Delta FIRE}$ mice, we have added this data for the heterozygous genotype to Fig EV2 and amended the main text as follows:

- *In contrast, these changes were not seen in $Csf1r^{+/\Delta FIRE}$ heterozygous mice (Figs EV2C and D).* (page 8 lines 223-224)

8) Fig. 5 C/D/E: While the number of animals analyzed is convincing, the number of cells sampled is too low. Authors should record at least 3 cells per mouse.

We carried out additional experiments and increased the number of cells and animals by 40-50%, resulting in an average of ~2 recorded neurons per mouse (pls note the attached table). Since each run of this experiment takes about 45-60 minutes (due to the pharmacology applied including wash-in and wash-out times), in the best case 2-3 such experiments can be performed per individual slice preparation per mouse without the slices becoming too old (which would lead to an additional bias). These extra experiments have substantiated our initial finding and further increased statistical significance of a higher AMPA/NMDA charge ratio in $Csf1r^{\Delta FIRE/\Delta FIRE}$ than WT mice.

9) Fig. 6B: the fact that in $Csf1r^{\Delta FIRE/\Delta FIRE}$ mice the brain environment seems to convert to a more anti-inflammatory one, raise the possibility that the process of synaptogenesis may be impacted in these mice. This is not even considered by the Authors.

The focus of our study is on the influence of pruning by microglia, i.e. the net elimination of synapses produced in excess during development, the resulting number of which is also influenced by synaptogenesis. Our central finding of an unchanged number of synapses and spine density after completion of development in $Csf1r^{\Delta FIRE/\Delta FIRE}$ mice (in addition to a lack of pruning) also suggests unaltered synaptogenesis. To consider this aspect, we amended the respective section in the Discussion to read as:

- *"Spine dynamics of hippocampal CA1 neurons remain extremely high also beyond this pruning period with a turnover of spines every 1-2 weeks in 10-12-week-old mice (Attardo et al., 2015; Pfeiffer et al., 2018), a time still captured in our analysis. This implies that even minor microglial influences on CA1 synapse elimination, remodeling **or formation** would have been expected to produce a visible effect in the number of spines in $Csf1r^{\Delta FIRE/\Delta FIRE}$ mice."* (page 14, line 380)

10) Fig.6C: The authors state that "Apolipoprotein E (ApoE) is mainly produced by astrocytes in the murine brain (Boyles et al., 1985; Pitas et al., 1987; Zhang et al., 2014 and 2016). Interestingly, soluble and membrane bound ApoE levels were strongly increased in the brains of $Csf1r^{\Delta FIRE/\Delta FIRE}$ mice compared to $Csf1r^{+/\Delta FIRE}$ and WT littermates (Fig 6C and Table 1), suggesting that changes in ApoE result from the absence of microglia rather than alterations in CSF1R function". I disagree with the authors regarding this statement. As they mentioned, ApoE is mainly produced by astrocytes in the brain. In fig. 7, authors show an increase in astrocyte density and "activation" profile in FIRE animals. Thus, the increase in ApoE is likely related to the increase in number and activation of astrocyte, rather than "alterations in CSF1R function". $Csf1r^{+/\Delta FIRE}$ could help in addressing more clearly this

issue.

We thank the referee for pointing out a possible misunderstanding. Indeed, what the referee suggests is congruent with our interpretation of the increased ApoE values. As this effect was only observed in homozygous but not heterozygous mice, this suggests that it is due to altered astrocyte function (as a result of the absence of microglia) rather than reduced or abolished CSF1R signaling.

To make this point clearer, we changed this sentence on page 11, line 312, to read:

- *Interestingly, soluble and membrane bound ApoE levels were strongly increased in the brains of $Csf1r^{\Delta FIRE/\Delta FIRE}$ mice compared to $Csf1r^{+/\Delta FIRE}$ and WT littermates (Fig 6C and Table 1), suggesting that changes in ApoE **are due to altered astrocyte function in the absence of microglia.***

11) Fig.6A: Why did authors analyze a whole hemisphere? Focusing on the hippocampus, as in the rest of the paper, would have been more accurate in describing the phenotypes observed in previous figures.

It is possible that cytokine levels may vary locally, but since $Csf1r^{\Delta FIRE/\Delta FIRE}$ mice lack microglia globally, our main goal was to examine if and to what extent the inflammatory milieu of the whole brain is altered. Potential changes in soluble cytokines may, however, not be unique to the brain region of interest, but also spread to and affect other brain regions. Importantly, based on the changes in whole brain cytokine and ApoE levels indicating altered astrocyte function, we have analyzed astrocyte properties in more detail in the hippocampus by additional immunohistochemical experiments (pls. note new Figs 7I-P).

12) Fig.6B: Authors sampled different cytokines by multiplex ELISA. However, they did not analyze what could be the origin of this dysregulation. The lack of microglial cells might have a role in Blood-Brain Barrier (BBB) formation (Ronaldson & Davis 2020), thus causing differential invasion of immune cells in the brain parenchyma. This needs to be addressed or at least discussed. Also, the macrophage colony-stimulating factor (CSF1) receptor (CSF1R) has two ligands, CSF1 and IL34. Did the authors test the levels of CSF1 and IL34 in the hippocampus and in the whole brain of $Csf1r^{+/\Delta FIRE}$ and $Csf1r^{\Delta FIRE/\Delta FIRE}$ mice?

As outlined in the review article by Ronaldson & Davis 2020 (PMID 32928017), activation of microglia can affect BBB integrity. However, in the absence of brain pathology, ablation of microglia is not known to cause damage of the blood-brain barrier or infiltration of non-microglial monocytes into the brain parenchyma as shown by numerous studies (e.g. Parkhurst et al., 2013, PMID 24360280; Elmore et al., 2014, PMID 24742461; Spangenberg et al., 2019, PMID 31434879; Konishi et al., 2022, PMID 32959911). Consistently, analysis of the integrity of the BBB in $Csf1r^{\Delta FIRE/\Delta FIRE}$ mice showed no evidence for disruption compared to WT littermates (Munro et al., manuscript submitted).

We have considered this point in the discussion on page 18, line 513 ff, stating that:

- *It is important to consider that the observed changes reflect the situation in the healthy brain **and are not complicated by the engraftment of peripheral immune cells in $Csf1r^{\Delta FIRE/\Delta FIRE}$ mice (Parkhurst et al., 2013; Spangenberg et al., 2019; Konishi et al.,***

2022). *This is likely to be different in the presence of brain damage or disease, in which the permanent absence of microglia has been associated with accelerated pathology in a dementia model and in prion disease (Spangenberg et al., 2019; Bradford et al., 2022; Shabestari et al., 2022).*

We did not investigate possible changes in CSF1 or IL34, as the loss of the CSF1R is decisive for the absence of microglia, on which its ligands have no influence. The complete absence of microglia in this mouse model is evident, i.e. there is no alternative signaling pathway for CSF1 or IL34 acting in place of CSF1R to maintain microglia. For the objectives of our study, only the absence of microglia is crucial, and potential direct effects of CSF1 or IL34 on neurons can be excluded as they do not express CSF1R.

13) Fig.6C: What is the functional meaning of ApoE increase in Δ FIRE / Δ FIRE animals? Authors should at least discuss it in text.

We thank the referee for raising this point. Indeed, apart from promoting anti-inflammatory effects, ApoE has been implicated in controlling phagocytosis of astrocytes (Chung et al., 2016, PMID 27559087) and glutamate signaling through ApoE receptors, affecting pre- and postsynaptic function (Lane-Donovan and Herz, 2017, PMID 28057414). For example, LDLR-related receptor 1 (Lrp1) controls the surface distribution and internalisation of NMDARs, while cholesterol complexed to ApoE acts as a positive regulator of presynaptic function (Mauch et al., 2001, PMID 11701931). These aspects are functionally related to the changes we observed in $Csf1^{\Delta$ FIRE/ Δ FIRE mice and suggest ApoE-mediated interactions between microglia and astrocytes that could be addressed by future studies.

We have now included potential functional implications of elevated ApoE levels in the discussion on pages 17/18, line 507 ff, as follows.

- ***Related to neural function, ApoE has been implicated in controlling phagocytosis of astrocytes (Chung et al., 2016) and glutamate signaling through ApoE receptors, affecting pre- and postsynaptic function (Lane-Donovan and Herz, 2017). For example, LDLR-related receptor 1 (Lrp1) controls the surface distribution and internalisation of NMDARs (Maier et al., 2013), while cholesterol complexed to ApoE acts as a regulator of presynaptic function (Mauch et al., 2001).***

14) Fig.7: The Authors show an increase in astrocyte density and ramification, which can be related to their activation. Since the role of astrocyte in eliminating adult hippocampal excitatory synapses is recognized (Lee et al., 2020), the Authors should test whether astrocytic synapse elimination is enhanced in the Δ FIRE / Δ FIRE model. Should this not occur and given that a similar synapse density is apparently present in WT and Δ FIRE / Δ FIRE mice, the possibility that the process of synaptogenesis is impaired in Δ FIRE / Δ FIRE mice would become even more likely, seriously challenging the Authors' conclusions.

The role of astrocytes in $Csf1^{\Delta$ FIRE/ Δ FIRE mice is indeed of great importance to better understand the lack of a deficit in the number of dendritic spines and synapses in this model. As suggested, we have greatly increased the number of experiments and animals in the revised manuscript, corroborating our key finding of an unchanged number of spines and excitatory synapses in $Csf1^{\Delta$ FIRE/ Δ FIRE vs WT mice post development. To follow the referee's advice and shed more light on the role of astrocytes, we first examined GFAP expression at

developmental stages P22-23, when the phagocytic capacity of microglia to engulf synaptic material has been reported to be highest in the hippocampal CA1 region (Jawaid et al., 2018, PMID 29274095). This showed an even greater increase in GFAP intensity and area compared to postdevelopmental stages, suggesting a change in astrocyte function especially in this critical phase of synapse remodelling in the developing hippocampus when microglia are otherwise active. To test whether this was associated with an increase in phagocytosis activity of astrocytes, we analysed expression of MEGF10 as a key astrocytic receptor regulating phagocytosis and synapse elimination (Chung et al., 2013, PMID 27559087). This revealed a strong increase (> 3-fold) in MEGF10 intensity in astrocytes of $Csf1r^{\Delta F1RE/\Delta F1RE}$ mice at P22/23 compared to littermate controls. Consistent with this, we found significantly increased (~25-30%) amounts of pre- and postsynaptic markers (VGlut1, Homer1) in astrocytes of $Csf1r^{\Delta F1RE/\Delta F1RE}$ mice.

We incorporated these new findings into the revised manuscript to the following sections of the Results and Discussion:

- *Apolipoprotein E (ApoE) is mainly produced by astrocytes in the murine brain (Boyles et al., 1985; Pitas et al., 1987; Zhang et al., 2014 and 2016). Interestingly, soluble and membrane bound ApoE levels were strongly increased in the brains of $Csf1r^{\Delta F1RE/\Delta F1RE}$ mice compared to $Csf1r^{+/+\Delta F1RE}$ and WT littermates (Fig 6C and Table 1), suggesting that changes in ApoE are due to altered astrocyte function in the absence of microglia. Morphological analysis of astrocytes in the CA1 stratum radiatum of 6-10-week-old $Csf1r^{\Delta F1RE/\Delta F1RE}$ mice revealed an increase in both intensity and area covered by GFAP immunoreactivity, while astrocyte numbers were unchanged compared to WT littermates (Figs 7A-D). In line with this, we observed increased complexity of astrocyte morphology, as evidenced by Sholl analysis revealing increases in the number of intersections and total process length (Figs 7E-H). **Notably, the intensity and area covered by GFAP immunoreactivity was even more pronounced in $Csf1r^{\Delta F1RE/\Delta F1RE}$ mice at P22-P23 (Fig. 7I-K), a time when engulfment of synaptic material by hippocampal microglia would normally have reached its peak (Jawaid et al., 2018). This was accompanied by marked increases in the expression of the key phagocytic receptor mediating synapse elimination in astrocytes, Multiple EGF-like domains (MEGF)10 (Fig. 7I, L, M; Chung et al., 2013), and enhanced incorporation of the synaptic markers VGlut1 and Homer1 in hippocampal astrocytes of $Csf1r^{\Delta F1RE/\Delta F1RE}$ mice at P22-P23 compared to WT littermates (Fig. 7N-P).***

Taken together, these data indicate a trend toward an anti-inflammatory state in brains of $Csf1r^{\Delta F1RE/\Delta F1RE}$ mice along with the presence of reactive astrocytes in the hippocampus with increased uptake of synaptic material. These changes are most prominent during hippocampal development, suggesting that astrocytes may contribute to pruning in the absence of microglia. (Results, page 11, 1st and 2nd paragraph)

- *Which alternative cell types or mechanisms may regulate the pruning of synapses in brains devoid of microglia? **Several types of macroglia, including astrocytes and oligodendrocyte precursor cells (OPCs), were identified as critical regulators of the pruning of synapses in the developing and adult brain (Chung et al., 2013; Buchanan et al., 2022). Notably, OPCs contain significantly more phagolysosomes filled with synaptic material than microglia in the developing mouse visual cortex (Buchanan et al., 2022). Astrocytes are more abundant than microglia and participate in activity-dependent synapse elimination and neural circuit formation in the developing and adult CNS by phagocytosing synapses via the MEGF10 and MERTK pathways (Chung et al., 2013; Lee et al., 2021). We find reactive astrocytes with increased expression of GFAP and MEGF10 in $Csf1r^{\Delta F1RE/\Delta F1RE}$ mice at postnatal developmental stages when synaptic engulfment by microglia would have been***

highest (Jawaid et al., 2018). We also find elevated levels of ApoE and IL-10 in brains of $Csf1^{\Delta FIRE/\Delta FIRE}$ mice. While IL-10 plays a role in synapse formation (Lim et al., 2013), ApoE controls the rate of synaptic pruning by astrocytes (Chung et al., 2016) and can affect glutamatergic transmission in multiple ways via signaling through synaptic ApoE receptors (Lane-Donovan & Herz, 2017). It is tempting to speculate that astrocytes may at least in part compensate for the loss of microglia in $Csf1^{\Delta FIRE/\Delta FIRE}$ mice, consistent with reports showing an up-regulation of their phagocytic capacity in mice with dysfunctional or ablated microglia (Konishi et al., 2020; Berdowski et al., 2022). Future studies, **ideally by high-resolution longitudinal live imaging of entire synapses**, will help to determine what proportion of excess or unwanted synapses in development are subject to glial elimination as opposed to other processes, whether and how these processes differ between brain regions, and which specific functional properties of synapses determine their fate. As previously suggested, the simplified view that only weaker synaptic contacts are removed falls short of reflecting the high degree of structural and functional heterogeneity of synapses in neural networks (Wichmann and Kuner, 2022). (Discussion, page 15, 2nd and 3rd paragraphs ff)

- Collectively, our findings provide evidence that microglia are dispensable for synaptic pruning in the developing hippocampus and suggest that microglia-independent mechanisms, **which may involve macroglia**, contribute to a basic, albeit not fully mature, network connectivity without causing behavioral disturbances. (end of Discussion, page 18, 2nd paragraph)

15) Fig.8: Before concluding that memory is not impaired in FIRE animals, additional behavioural tests should be performed (Morris water maze, T maze). Also, a different way of showing the results of the test should be adopted, in order to facilitate the readers in appreciating that WT mice are able to discriminate the novel object.

As we state in the discussion, spatial memory has already been examined in $Csf1^{\Delta FIRE/\Delta FIRE}$ mice at the same age as in our work using the Barnes maze (McNamara et al., 2023, PMID 36517604). This study showed that spatial learning and memory encoding were unaltered in $Csf1^{\Delta FIRE/\Delta FIRE}$ mice. Moreover, further behavioural analysis revealed that these mice showed no signs of anxiety or motor deficits but were less capable to adjust to a new situation. This deficit in cognitive flexibility is dependent on the structural integrity of myelin, which is impaired in the absence of microglia (McNamara et al., 2023).

As suggested by the referee, we have chosen a more intuitive way to represent the discrimination index in Fig. 8B using the formula: $[DI = (Novel\ Object\ Exploration\ Time / Total\ Exploration\ Time) - (Familiar\ Object\ Exploration\ Time / Total\ Exploration\ Time) \times 100]$, which we added in the methods, page 19/20, line 561 ff. This plot of the data shows that the mice preferentially interacted with the novel object (positive discrimination indices), but without differences seen between genotypes.

16) Authors should state clearly for every experiment whether they are analyzing the same number of male and female mice in CTRL and FIRE animals, since they are evaluating phenotypes that might be related to sex.

The attached table with information on all animals/cells/slices included in our study indicates that the sex ratio is now well balanced for all experiments. We would therefore refrain from specifying the individual sex per subfigure for reasons of readability and refer to this in the methods by stating that: “Experiments used transgenic $Csf1^{\Delta FIRE/\Delta FIRE}$ mice (Rojo et al., 2019, PMID 31324781) which were on a mixed C57BL/6J x CBA/J background with

littermate heterozygous $Csf1^{\Delta FIRE/+}$ and littermate wild-type $Csf1^{+/+}$ controls of both sexes aged 6-10 weeks, or P9-P10 for experiments in Figs 1M, N and P22-P23 for Figs 7I-P.” (page 19, 1st paragraph)

17) Authors should specify in all figures to what the number in the bars they refers to.

We have added this information to all figure legends.

Referee #3:

In this remarkably well-written manuscript, Surala and colleagues explore the functional consequences of developmental absence of microglia in the FIRE CSF1R mouse. This mouse will likely find widespread application in teasing out microglia-specific function in neurodevelopment and disease and thus careful analyses are of great interest amongst neuroscientists and neuroimmunologists. The authors demonstrate that, while not of consequence in one non-spatial learning paradigm, congenital absence of microglia due to CSF1R fire element deletion influences functional but not structural synapse formation and leads to decreased excitatory synapse function in hippocampal circuits. Astrocytes show increased GFAP reactivity as a potential clue into their function. The studies are technically excellent and the analyses complete. The potential limitation of this work for the audience is that it characterizes one rodent model of congenital absence of microglia (as very few if any equivalent models exist) and the mechanism remains uncertain. To further strengthen the impact of this work, the authors could determine whether known astrocyte synaptogenic proteins are differentially regulated in microglia absence (especially those which mediate functional but not necessarily structural synapse formation). That said, this work is of broad interest and is likely to be highly regarded as there is increasing excitement in studying microglial function using new genetic systems.

We thank the referee for the positive assessment of our study. The role of astrocytes in $Csf1^{\Delta FIRE/\Delta FIRE}$ mice is indeed of great importance to better understand the lack of a deficit in the number of dendritic spines and excitatory as well as inhibitory synapses in this model. In the revised manuscript, we extended our analyses of astrocytes by examining GFAP expression in addition at developmental stages P22-23, when the phagocytic capacity of microglia to engulf synaptic material has been reported to be highest in the hippocampal CA1 region (Jawaid et al., 2018, PMID 29274095). This showed an even greater increase in GFAP intensity and area compared to postdevelopmental stages (6-10 weeks), suggesting a change in astrocyte function especially in this critical phase of synapse remodelling in the developing hippocampus when microglia are otherwise active. To test whether this was associated with an increase in phagocytosis activity of astrocytes, we further analyzed expression of MEGF10 as a key astrocytic receptor regulating phagocytosis and synapse elimination (Chung et al., 2013, PMID 27559087). This revealed a strong increase (> 3-fold) in MEGF10 intensity in astrocytes of $Csf1^{\Delta FIRE/\Delta FIRE}$ mice at P22/23 compared to littermate controls, suggesting upregulation of phagocytosis in the absence of microglia. Consistent with this, we found significantly increased (~25-30%) amounts of pre- and postsynaptic markers (VGluT1, Homer1) incorporated into astrocytes of microglia-deficient mice.

We incorporated these new findings to the following sections of the revised manuscript (changes are highlighted in bold):

- *Apolipoprotein E (ApoE) is mainly produced by astrocytes in the murine brain (Boyles et al., 1985; Pitas et al., 1987; Zhang et al., 2014 and 2016). Interestingly, soluble and membrane bound ApoE levels were strongly increased in the brains of Csf1r^{ΔFIRE/ΔFIRE} mice compared to Csf1r^{+ΔFIRE} and WT littermates (Fig 6C and Table 1), suggesting that changes in ApoE are due to altered astrocyte function in the absence of microglia. Morphological analysis of astrocytes in the CA1 stratum radiatum of 6-10-week-old Csf1r^{ΔFIRE/ΔFIRE} mice revealed an increase in both intensity and area covered by GFAP immunoreactivity, while astrocyte numbers were unchanged compared to WT littermates (Figs 7A-D). In line with this, we observed increased complexity of astrocyte morphology, as evidenced by Sholl analysis revealing increases in the number of intersections and total process length (Figs 7E-H). **Notably, the intensity and area covered by GFAP immunoreactivity was even more pronounced in Csf1r^{ΔFIRE/ΔFIRE} mice at P22-P23 (Fig. 7I-K), a time when engulfment of synaptic material by hippocampal microglia would normally have reached its peak (Jawaid et al., 2018). This was accompanied by marked increases in the expression of the key phagocytic receptor mediating synapse elimination in astrocytes, Multiple EGF-like domains (MEGF)10 (Fig. 7I, L, M; Chung et al., 2013), and enhanced incorporation of the synaptic markers VGlut1 and Homer1 in hippocampal astrocytes of Csf1r^{ΔFIRE/ΔFIRE} mice at P22-P23 compared to WT littermates (Fig. 7N-P).** Taken together, these data indicate a trend toward an anti-inflammatory state in brains of Csf1r^{ΔFIRE/ΔFIRE} mice **along with the presence of reactive astrocytes in the hippocampus with increased uptake of synaptic material. These changes are most prominent during hippocampal development, suggesting that astrocytes may contribute to pruning in the absence of microglia.** (Results, page 11, 1st and 2nd paragraph)*
- *Which alternative cell types or mechanisms may regulate the pruning of synapses in brains devoid of microglia? **Several types of macroglia, including astrocytes and oligodendrocyte precursor cells (OPCs), were identified as critical regulators of the pruning of synapses in the developing and adult brain (Chung et al., 2013; Buchanan et al., 2022). Notably, OPCs contain significantly more phagolysosomes filled with synaptic material than microglia in the developing mouse visual cortex (Buchanan et al., 2022). Astrocytes are more abundant than microglia and participate in activity-dependent synapse elimination and neural circuit formation in the developing and adult CNS by phagocytosing synapses via the MEGF10 and MERTK pathways (Chung et al., 2013; Lee et al., 2021). We find reactive astrocytes with increased expression of GFAP and MEGF10 in Csf1r^{ΔFIRE/ΔFIRE} mice at postnatal developmental stages when synaptic engulfment by microglia would have been highest (Jawaid et al., 2018). We also find elevated levels of ApoE and IL-10 in brains of Csf1r^{ΔFIRE/ΔFIRE} mice. While IL-10 plays a role in synapse formation (Lim et al., 2013), ApoE controls the rate of synaptic pruning by astrocytes (Chung et al., 2016) and can affect glutamatergic transmission in multiple ways via signaling through synaptic ApoE receptors (Lane-Donovan & Herz, 2017). It is tempting to speculate that astrocytes may at least in part compensate for the loss of microglia in Csf1r^{ΔFIRE/ΔFIRE} mice, consistent with reports showing an up-regulation of their phagocytic capacity in mice with dysfunctional or ablated microglia (Konishi et al., 2020; Berdowski et al., 2022). Future studies, **ideally by high-resolution longitudinal live imaging of entire synapses**, will help to determine what proportion of excess or unwanted synapses in development are subject to glial elimination as opposed to other processes, whether and how these processes differ between brain regions, and which specific functional properties of synapses determine their fate. As previously suggested, the simplified view that only weaker synaptic contacts are removed falls short of reflecting the high degree of***

structural and functional heterogeneity of synapses in neural networks (Wichmann and Kuner, 2022). (Discussion, pages 15-16)

- *Collectively, our findings provide evidence that microglia are dispensable for synaptic pruning in the developing hippocampus and suggest that microglia-independent mechanisms, **which may involve macroglia**, contribute to a basic, albeit not fully mature, network connectivity without causing behavioral disturbances. (end of Discussion, page 18, 2nd paragraph)*

Date: 2nd Feb 24 15:04:04

Last Sent: 2nd Feb 24 15:04:04

Triggered By: Ioannis Papaioannou

From: i.papaioannou@embojournal.org

To: christian.madry@charite.de

CC: josef.priller@charite.de

Subject: Manuscript EMBOJ-2023-114813R - Decision

Message: Dear Christian, dear Josef,

Thank you for submitting your revised manuscript (EMBOJ-2023-114813R) for consideration by The EMBO Journal. It has now been seen by the three original referees, and we have received the full set of their comments, which you can find below.

As you will see, the referees recognize that the study was thoroughly revised and that the strengthened new version of the manuscript is now significantly improved with the addition of new data. However, they raise concerns regarding the interpretation of the results, and they point out that the new data in essence change the outcome of the study. They explain that the findings overall are more consistent with the notion that microglia are actually relevant to synapse pruning during development.

I have discussed your study and the referees' comments with the other members of our team, and I am very sorry to say that -in light of the new data and the referees' reports- we are all in agreement that the overall advance provided by the revised study is not sufficiently striking for publication in The EMBO Journal, and we have therefore decided that we unfortunately cannot offer publication of the study in our journal.

That said, I have discussed your manuscript along with the referees' reports with my colleague Dr. Esther Schnapp, senior editor of our sister journal EMBO reports. Esther would like to discuss the suitability of your work for EMBO reports with the referees should you be interested in the option of transferring your manuscript and its review history to EMBO reports. Please use the transfer link at the bottom of this message if you are interested (no re-formatting or re-upload of files will be necessary), or contact Esther directly (at e.schnapp@emboreports.org) if you have any questions.

For The EMBO Journal, I am sorry to have to disappoint you on this occasion, but I hope that you will view the possibility of a transfer favorably. I wish you every success in publishing your study in a more suitable journal.

Best regards,

Ioannis

Referee #1:

The revised version of the article is significantly improved compared to the original submission. As expected, the addition of new data has changed the outcome (e.g. Fig. EV1). The authors should, however, pay more attention to adjusting their text respectively (e.g., the title of Fig. EV1 still talks about the reduced excitability of CA1 pyramidal cells, which, according to Fig. EV1 itself, is completely gone when adding new data). The new astrocytic data obtained at P22-23 and showing a > 3-fold increase in MEGF10 intensity in astrocytes of *Csf1r Δ FIRE/ Δ FIRE* mice as well as 25-30% more pre- and postsynaptic markers within these astrocytes are also extremely valuable. To my opinion, however, these data completely change the meaning of the main findings of this study. Instead of stating that microglia are dispensable, i.e. non-essential, for synaptic pruning in the developing hippocampus (the main tenor of the current story), the data seem to show that under physiological conditions microglia are responsible for phagocytosing at least 25-30% of synapses but under pathological conditions (i.e. complete absence of microglia) this task can be taken over by activated astrocytes.

As correctly stated by the authors, OPCs also vividly phagocytose synaptic material and thus might have an even higher capacity to compensate for lacking microglia. Would the likely upregulation of the OPC-mediated synaptic phagocytosis also be analyzed in this study, it would turn out that with ~ 50% of pruned synapses, the microglia are, contrary to what is now stated by the authors, the main responsible players under physiological conditions. Unsurprisingly, however, their role can be taken over by other neuroglia under pathological conditions.

It is up to the editor to request the analyses of OPCs. In any case, however, the alternative scenario described above has to be thoroughly discussed in the manuscript including the abstract, not to mislead the readers.

Minor but still very important: when writing 19 pages-long response to the reviewer's comments, please be so kind as to include page numbers.

Referee #2:

The authors answered most questions and added more mice to the statistical analyzes as required.

However, I still have major concerns about the main take-home message of this manuscript.

The authors' main claim, also reported in the title of the manuscript, is that microglia are dispensable for synaptic pruning. This statement is misleading.

What the authors observe is that, in *Csf1r Δ FIRE/ Δ FIRE* mice, astrocytes compensate for microglial function, which supports the relevance of microglia in synapse pruning during development. The title and conclusions are not supported by the data and should be strongly moderated.

Furthermore:

The authors do not to state the sex of the mice in all conditions tested.

Analysis in the pruning window (p15-p20), which is key for the aim of this study, has not been performed.

The authors mention an attached table listing all previous and newly added experiments for each subfigure with details of animal number, genotype and sex. I was unable to find the table mentioned by the Authors in the point-to-point response.

Referee #3:

The authors sufficiently answered this reviewer's concerns. Of note, the authors take a heavy hand in the discussion to explain that microglia are dispensable for synapse elimination in the long term, discussing that there is not the confound of ablation methods. While true, there is also great literature looking at cell-specific gene knockouts, which do show deficits. Missing from the discussion, then, is relating some of the confounders of the fire model (i.e., perhaps not all macroglia are growing up as normal in this microglia-less system) and why there is a discrepancy. One hypothesis raised in the discussion is that other papers didn't look long enough; on the other hand, there may be consequences of lacking an important immune cell in the brain in the long term i.e. developmentally arming

macroglia to take on additional roles.

** As a service to authors, EMBO Press provides authors with the possibility to transfer a manuscript that one journal cannot offer to publish to another EMBO publication or the open access journal Life Science Alliance launched in partnership between EMBO Press, Rockefeller University Press and Cold Spring Harbor Laboratory Press. The full manuscript and if applicable, reviewers' reports, are automatically sent to the receiving journal to allow for fast handling and a prompt decision on your manuscript. For more details of this service, and to transfer your manuscript please click on ** Link Unavailable. **

Dear Ioannis,

Many thanks for your last email. We were astonished by your editorial decision to reject our manuscript after initial submission of our work in June 2023 and thorough revision addressing all of the referees' suggestions. Please allow us to comment on this.

There appears to be consensus among the three referees that our extensive revision experiments have significantly increased the validity of our study by generating important new findings. There also appears to be agreement on the quality and importance of our previous and new data. Our main statement that microglia are evidently not mandatory for synaptic pruning during brain development is novel and defies a dogma which the referees seem to try to uphold at any cost.

Referees 1 and 2 criticize the interpretation of our data. They should be aware that our statement about microglia being 'dispensable' for pruning does not mean that microglia are irrelevant under conditions when they are present. In fact, this is not the subject of our study and is not denied by us at any point. Instead, our model addresses the important question of the necessity of microglia for proper pruning, which we explicitly point out at the beginning of the abstract. Our data unequivocally demonstrate that microglia are not needed (either because their functions are taken over by other cells or because macroglia always contribute to pruning). This is a remarkable rewriting of current views on the development of the CNS.

It is completely unclear to us how referee 1 comes up with the conclusion that microglia are the main players in physiological synaptic pruning when this is unaffected by their absence throughout life. The fallacy of attributing macroglia only compensatory roles is not scientifically sound and represents a misleading inversion of the logic of our main finding of normal pruning in the absence of microglia.

Our results clearly demonstrate that microglia are not necessary for pruning, i.e. they are not essential and are dispensable (redundant) for this process, although we do find a significant influence of microglia on glutamatergic network maturation. Compared to our first submission, no new data have emerged that cast doubt on our central statements.

We ask you to reconsider your editorial decision in the light of these misunderstandings in the interpretation of our data. We are happy to reformulate in a way that you feel would avoid polarizing interpretation. We feel strongly that personal views of referees should not obstruct the publication of timely results. Perhaps we could have a short zoom call tomorrow? Perhaps Karin, who initially edited our submission and invited us to revise our work, could also share her views?

Many thanks for your consideration and looking forward to hearing from you,

Christian and Josef

Dear Christian, dear Josef,

Thank you for your message requesting us to re-consider our decision on your manuscript. I have read your comments carefully and discussed them again with the other members of our team. Since this manuscript was initially handled by Karin (who also invited the revision, as you correctly point out), I included her in our consultation, of course - she also participated in the previous round of discussion.

We are all in agreement that with the referee input available at this point, the convergent opinions of two of the expert referees -who rather strongly suggest that the results of the work are misinterpreted in a biased way- cannot be overruled at the editorial level. We do understand, however, that your conclusions go to a large extent against a well-established dogma, i.e. that of microglia necessity for physiological synaptic pruning, and we are aware that such cases can be challenging for various reasons.

All things considered, I would like to suggest consulting with an additional advisor/arbitrator, with whom I would share your revised manuscript as well as the last round of referee reports, and ask for their additional advice on whether they find the data strong and clear-cut enough to support your conclusions regarding the questions of dispensability and importance of microglia in physiological synaptic pruning. I think this would be the fairest way forward and would inform and facilitate our further discussion, but I would like to explain that this process would likely take a few days, and that there are no guarantees regarding its outcome.

Please let me know if you are interested in this option, or if you have any other comments.

Best wishes,

Ioannis

Author's Response to Editor's Feedback

Dear Ioannis,

Thank you very much for your feedback and explanation. We welcome your suggestion to obtain additional advice for clarification.

Please allow us one more comment: Referees 1 and 2 raise the question of the *relevance* of microglia for synaptic pruning. Our study gives a clear answer to this by providing comprehensive experimental evidence as to which function of microglia is non-essential, i.e. dispensable (synaptic pruning) and which is not (glutamatergic network maturation). Whether the dispensability of a process (the existence of which we do not deny) may nevertheless be considered as *relevant*, as the referees insinuate and which is core of the controversy, is subject to personal, but not scientific, interpretation and unfortunately distracts from the central findings of our study.

We would also appreciate the opportunity to have a zoom call with you after the consultation.

Thank you again and best wishes,

Christian and Josef

Date: 26th Feb 24 12:30:04

Last Sent: 26th Feb 24 12:30:04

Triggered By: Ioannis Papaioannou

From: i.papaioannou@embojournal.org

To: christian.madry@charite.de

CC: Josef.Priller@charite.de

Subject: Manuscript EMBOJ-2023-114813R1-Q - Decision

Message: Dear Christian, dear Josef,

Thank you for your message requesting us to re-consider our previous decision on your manuscript EMBOJ-2023-114813R1-Q. I have now received advice from an additional expert in the field with whom I shared the last version of your manuscript, the referee reports, and your arguments presented in your appeal letter. Please find their advice included below.

As you will see, the advisor thinks that the data presented in the manuscript do not sufficiently support the main conclusion regarding dispensability of microglia in the pruning process. They also point out that, in their view, the absence of microglia itself represents a pathological condition, which could trigger a compensatory response from astrocytes. Furthermore, they identify a limitation in the study, i.e. that the data shown are limited to a late developmental phase, which in their opinion further weakens the study and its conclusions.

In light of this advice, we have discussed again your manuscript, the input of the three referees and the additional advisor, as well as your appeal letter in our editorial team. I would like to note here that Karin Dumstrei, who handled the previous version of your manuscript, was included in these consultations. I am afraid that with this input from the reviewers and the advisor, we are all in agreement that we unfortunately cannot overturn our initial decision and cannot offer publication of the study in The EMBO Journal.

Having said that, I would like to encourage you to consider the -still-standing- offer of the EMBO reports editorial team to discuss the suitability of your manuscript for them with the referees, should you decide to transfer your manuscript to them.

I am sorry to have to disappoint you on this occasion, and I thank you once again for the opportunity to consider your manuscript. I hope that you will view the option of transferring your manuscript to EMBO reports favorably.

Best regards,

Ioannis

Referee #4 (additional advisor):

I completed the analysis of the MS by Surala and Coauthors, in the respect required. In general, I consider the study solid and valuable, although on this matter the work of the reviewers has been certainly deeper.

Considering the main point of controversy, I am prone to sharing the position expressed by the reviewers #1 and #2. Indeed, I think that the main claim of the dispensability of microglia in the pruning process is not fully substantiated by the results of the study. In particular, I share the motivation supported by the two reviewers, that the absence of microglia itself represents a pathological condition, which could trigger a compensatory response from astrocytes. This could compensate for the absence of microglia, hindering the estimation of the role of these cells in the physiological process.

In my view, more importantly, the data showed are limited to a late developmental phase while the pruning defect might be measurable only in a precise developmental window (i.e. 3rd - 4th week), being subsequently compensated by other mechanisms, but leaving permanent functional and structural defects (see Paolicelli et al., 2011), similar to those observed by the authors and, in some respects, to those induced by post developmental microglia removal. To demonstrate the absence of pruning in the microglia defective mice, the authors should check for the density of spines at key intermediate developmental stages. Otherwise, I would suggest moderating the main conclusions.

** As a service to authors, EMBO Press provides authors with the possibility to transfer a manuscript that one journal cannot offer to publish to another EMBO publication or the open access journal Life Science Alliance launched in partnership between EMBO Press, Rockefeller University Press and Cold Spring Harbor Laboratory Press. The full manuscript and if applicable, reviewers' reports, are automatically sent to the receiving journal to allow for fast handling and a prompt decision on your manuscript. For more details of this service, and to transfer your manuscript please click on **Link Unavailable. **

Dear Christian and Josef,

Thank you for the transfer of your revised manuscript to EMBO reports. I discussed it with the EMBO reports team, and we would like to publish it pending some text changes. I asked referee 2 for further input and s/he agreed with the following changes:

- Please remove the word pruning from the title, given that pruning has not been directly investigated. Pruning can be mentioned in the abstract, see below. I would like to suggest a title along these lines:
"Microglia shape hippocampal networks but are dispensable for excitatory synapse number and spine density"
Similar title suggestions from your side are very welcome.
- In the first sentence of the abstract, please replace "believed" with "have been shown". If you disagree, please let me know.
- Please replace the last sentence of the abstract with something like this:
Thus, our findings suggest/indicate that microglia are not strictly required/dispensable for synapse pruning, and that in their absence, pruning is likely taken over by other glia cells.
- The last sentence in the introduction is incorrect and needs to be deleted or modified.
- May be the last sentence in the Discussion could be modified to "Collectively, our findings provide evidence indicating/suggesting that ..."

A few other editorial requests will also need to be addressed:

- Some funding info is missing in the ms file but present in our online ms submission system, please correct. The information must be the same.
- Please add up to 5 keywords to the ms file.
- Please add a DATA AVAILABILITY SECTION (DAS) to your ms file. If no newly generated data have been deposited in public databases please mention this fact in the DAS.
- Please correct the conflict of interest subheading to "DISCLOSURE AND COMPETING INTERESTS STATEMENT"
- Please co-submit with your final ms a fully completed author checklist that can be found here:
<<https://www.embopress.org/page/journal/14693178/authorguide>>.
- All EV figures need to be uploaded as individual files.
- Our screening of revised ms images identified 2 image duplications in Figure 1A and EV1A, and in Figure 1J and EV3G. If these are the same images from the same experiments please mention this in the figure legend. If not, please explain.
- Please note that the legends for figures EV 4b-d are not provided in the sequential manner (legend for figure EV 4d is provided before the legend of figures EV 4b-c). Please correct.
- Please note that the figure panels 1c, f do not provide "n". Please rectify the statistics "n" related information in the legends appropriately.
- Information related to "n" is missing in the legends of figure 8b; EV 5b, please add.
- Please note that the scale bar needs to be defined for figure 7n.
- Please note that yellow arrowheads are defined in the legend of figure 1j, however no yellow arrowheads are present in the figure. This needs to be rectified.

EMBO press papers are accompanied online by A a short (1-2 sentences summary of the findings and their significance, B 2-3 bullet points highlighting key results and C a synopsis image that is exactly 550 pixels wide and 200-600 pixels high (the height is variable. You can either show a model or key data in the synopsis image. Please note that text needs to be readable at the final size. Please send us this information along with the final manuscript.

I look forward to seeing a final version of your manuscript as soon as possible

All editorial and formatting issues were resolved by the authors.

Prof. Christian Madry
Charite - Universitätsmedizin Berlin
Institute of Neurophysiology
Chariteplatz 1
Berlin 10117
Germany

Dear Christian,

I am very pleased to accept your manuscript for publication in the next available issue of EMBO reports. Thank you for your contribution to our journal.
